# A Task-Centric Theory for Iterative Self-Improvement with Easy-to-Hard Curricula

**Chenruo Liu** [1]  **Yijun Dong** [1]  **Yiqiu Shen** [1]  **Qi Lei** [1]

## Abstract

Iterative self-improvement fine-tunes an autoregressive large language model (LLM) on reward-verified outputs generated by the LLM itself. In contrast to the empirical success of self-improvement, the theoretical foundation of this generative, iterative procedure in a practical, finite-sample setting remains limited. We make progress toward this goal by modeling each round of self-improvement as maximum-likelihood fine-tuning on a reward-filtered distribution and deriving finite-sample guarantees for the expected reward. Our analysis reveals an explicit feedback loop where better models accept more data per iteration, supporting sustained self-improvement while explaining eventual saturation of such improvement. Adopting a task-centric view by considering reasoning tasks with multiple difficulty levels, we further prove quantifiable conditions on model initialization, task difficulty, and sample budget where easy-to-hard curricula provably achieve better guarantees than training on fixed mixtures of tasks. Our analyses are validated through Monte-Carlo simulations and experiments spanning a synthetic graph-based reasoning task and multiple standard mathematical reasoning benchmarks.

## 1. Introduction

Conditioned on strong pre-training, modern large language models (LLMs) increasingly acquire their unprecedented reasoning skills during post-training not only from human-annotated supervision but also via *iterative self-improvement*—a supervision-free loop where the model iteratively generates candidate answers for questions from the downstream task and then gets fine-tuned on curated question-answer pairs that pass certain external verifications (Xin et al., 2024; 2025; Zelikman et al., 2022; Lin et al., 2025; 2026; Ren et al., 2025; Zhang et al., 2025; Guan et al., 2025). A closely related practice is to schedule these self-improvement iterations using an *easy-to-hard curriculum*, where gradually shifting the downstream question distribution toward more challenging instances often improves the final performance (Ren et al., 2025; Koh et al., 2026; Lee et al., 2025).

From a theoretical perspective, the empirical success of self-improvement is arguably surprising because the training data is not exogenous, i.e., the candidate solutions are generated by the model itself. Recent theories on self-improvement have made progress on clarifying why learning such endogenous data succeeds without violating the data processing inequality (Shannon, 1948), mainly from the model evolution perspective. For example, Huang et al. (2025) casts self-improvement as a form of probability-mass "sharpening"; while Sun et al. (2026) takes a solver–verifier-gap view of the learning dynamics at the continuous limit. However, as a post-training strategy, the success of self-improvement is highly dependent on the interaction between the pre-trained model and the downstream task. In addition, the discrete, multi-step iterations are critical for the appealing empirical gain of self-improvement. What remains theoretically under-specified for such self-improvement pipelines is a finite-sample, task-centric account that answers two practical questions:

> *With a single task, when does (multi-step)*
> *self-improvement happen?*
> *With a mixture of tasks, when does task scheduling, like*
> *easy-to-hard curricula, provably help?*

**Our contributions.** We provide theory-grounded answers to the above questions that closely match self-improvement in practice, which can be summarized as follows:

- **A task-centric framework unveiling when does (multi-step) self-improvement happen on a single task.** Toward the first question, we model each self-improvement iteration as maximum-likelihood fine-tuning on a reward-filtered distribution induced by the model's own gener-

[1]New York University. Correspondence to: Chenruo Liu <cl7758@nyu.edu>, Qi Lei <ql518@nyu.edu>.

*Proceedings of the 43rd International Conference on Machine Learning*, Seoul, South Korea. PMLR 306, 2026. Copyright 2026 by the author(s).

ation and an external verifier (Section 3). On a single task, our expected reward lower bounds for single/multi-step self-improvement highlight a feedback loop in which the filtered distributions of better models have more data per iteration being accepted, effectively increasing the training set size. This lens further clarifies the effects of model initialization, task difficulty, and finite sample sizes on the sustained self-improvement and its eventual saturation (Section 4).

- **Across multiple tasks, moderate separation in task difficulties is essential for effective easy-to-hard curricula.** Toward the second question, we compare easy-to-hard scheduling across self-improvement iterations with training on a fixed-mixture baseline under the same budget. We derive quantifiable conditions under which easy-to-hard curricula enjoy strictly better expected reward lower bounds. The analysis unveils three levers for the effectiveness of easy-to-hard scheduling in self-improvement: moderate separation of task difficulties, a critical sample budget, and model initialization (Section 5).

- **Empirical validation across diverse mathematical reasoning tasks.** We validate the analysis for both questions through experiments on a synthetic graph-based reasoning task and multiple standard mathematical reasoning benchmarks, including GSM8K (Cobbe et al., 2021) and DeepMind Mathematics (Saxton et al., 2019) (Section 6), and complement these experiments with Monte-Carlo simulations that directly visualize the evolution of the expected reward lower bound across self-improvement iterations (Section 5).

## 2. Related Work

**Self-improvement for LLM mathematical reasoning.** In the realm of LLMs, self-improvement broadly refers to a family of procedures in which a LLM produces its own supervision signal and then leverages this signal to improve its capabilities. Empirically, many pipelines for reasoning tasks, especially mathematical reasoning, instantiate this idea via an iterative generate-and-filter loop: the model generates one or multiple candidate solutions for each problem, retains a subset of correct solutions, and then fine-tunes on the retained solutions to enhance performance (Zelikman et al., 2022; Xin et al., 2024; 2025; Guo et al., 2025; Lin et al., 2025; 2026; Ren et al., 2025; Zhang et al., 2025; Guan et al., 2025). While existing approaches vary in implementation details (e.g., incorporating long chain-of-thought (CoT) reasoning (Lin et al., 2026), tactic annotations (Xin et al., 2025), code-augmented CoT data (Guan et al., 2025), or reflection steps (Zhang et al., 2025)), their overall framework shares the same spirit of bootstrapping from model-generated attempts filtered by an explicit correctness signal. Additionally, a growing line of work (Ren et al., 2025; Lee et al., 2025; Koh et al., 2026) shows that combining

self-improvement with an explicit easy-to-hard curriculum across rounds can further strengthen model performance.

**Theoretical Understanding of LLM self-improvement.** While LLM self-improvement has shown strong empirical success, another line of work seeks to understand its underlying mechanisms from a theoretical perspective. Huang et al. (2025) formalizes self-improvement as a consequence of a sharpening mechanism, which encourages the model to place larger probability mass on higher-quality sequences. From a different perspective, Mohri et al. (2025) studies self-improvement through the lens of coherence. Yang et al. (2026) investigates the impact of budget allocation policies across self-improvement iterations. Sun et al. (2026) models the training dynamics of self-improvement via the solver-verifier gap. In contrast to these works, our theoretical modeling of self-improvement is more closely aligned with mathematical reasoning settings, i.e., binary and verifiable rewards together with reject sampling for data collection. More importantly, our analysis characterizes multi-step iterative self-improvement and further incorporates its interaction with easy-to-hard curricula.

**Self-distillation, self-consuming loops, and model collapse.** First, several theoretical works on self-distillation analyze training a model with supervision signals generated by the model itself (Mobahi et al., 2020; Das & Sanghavi, 2023; Pareek et al., 2024). In contrast, we study LLM self-improvement for generative reasoning, which is typically outside the scope of standard self-distillation analyses (e.g., linear predictors). Second, another related line of work concerns self-consuming loops in generative models, a failure mode where repeatedly training on generated data can degrade performance and even lead to model collapse (Fu et al., 2024; 2025). Recent theory shows that such degradation can be mitigated under suitable mechanisms (Gillman et al., 2024; Gerstgrasser et al., 2024; Ferbach et al., 2024; Feng et al., 2025; Fu et al., 2025; 2026). While the objective of preventing model collapse in this literature differs from LLM self-improvement, Ferbach et al. (2024) is particularly relevant to our work as it explicitly characterizes how curated data can optimize a reward signal. However, it is not tailored to mathematical reasoning, and thus the resulting trends do not align as closely with empirical practice. A primary reason is that it does not account for the finite-sample regime, which, as we argue in this paper, is important to understanding self-improvement in mathematical reasoning.

## 3. Problem Setup and Notation

**Problem setup.** We introduce a theoretical formulation of a practical single-iteration self-improvement procedure for mathematical reasoning (Zelikman et al., 2022; Xin et al., 2025; Guo et al., 2025; Lin et al., 2026). At iteration $t$

(with $t$ starting from 0), we sample questions $q \sim p_0$ and generate answers $a \sim \pi_{\theta_t}(\cdot \mid q)$ using the current model parameters $\theta_t$. Unless otherwise stated, we generate a single candidate answer per question. For each pair $(q, a)$ we define a reward (score) function $s(q, a) \in [0, 1]$, where a larger reward indicates a better answer. We retain only samples whose reward is at least a threshold $\tau \in (0, 1]$ and discard the rest.

At the population level, this filtering induces the distribution

$$ D'_{p_0, \theta_t}(q, a) = \frac{p_0(q)\, \pi_{\theta_t}(a \mid q)\, \mathbf{1}_{\{\, s(q,a) \geq \tau \,\}}}{Z_{p_0}(\theta_t)}, $$

where $\alpha(\theta, q)^1 := \Pr_{a \sim \pi_\theta(\cdot|q)}\big[s(q, a) \geq \tau\big]$ and $Z_p(\theta) := \mathbb{E}_{q \sim p}\big[\alpha(\theta, q)\big]$ denote the per-question and global acceptance rates, respectively. The idealized model update is then $\theta_{t+1} = \arg\max_\theta \mathbb{E}_{(q,a) \sim D'_{p_0, \theta_t}}\big[\log \pi_\theta(a \mid q)\big]$. In practice, with finite samples, given the current model $\hat{\theta}_t$ and a dataset of $n$ sampled questions, we obtain a random number $n_t$ (with $n_t \leq n$) of accepted samples $\{(q_i, a_i)\}_{i=1}^{n_t} \sim D'_{p_0, \hat{\theta}_t}$, and perform empirical maximum likelihood estimation:

$$ \hat{\theta}_{t+1} = \arg\max_\theta \ \frac{1}{n_t} \sum_{i=1}^{n_t} \log \pi_\theta(a_i \mid q_i). $$

Our goal is to relate this empirical self-improvement objective to the evaluation metric, namely the expected reward under $p_0$, $V_{p_0}(\hat{\theta}_{t+1}) := \mathbb{E}_{(q,a) \sim D_{p_0, \hat{\theta}_{t+1}}}\big[s(q, a)\big]$, where $D_{p, \theta}(q, a) := p(q)\pi_\theta(a \mid q)$.

**Notation.** Appendix Table 3 summarizes the notation used throughout the paper, together with brief descriptions and definitions.

# 4. Iterative Self-Improvement

This section develops basic theoretical tools for iterative self-improvement. We start with a single-step analysis under a general reward function $s(q, a)$ and threshold $\tau$. We then specialize to mathematical reasoning and study the resulting multi-step self-improvement, which will serve as a foundation for our analysis of the easy-to-hard curriculum in Section 5.

## 4.1. Single-Step Self-Improvement

We begin by analyzing a single round of self-improvement in the finite-sample regime.

**Theorem 4.1.** *Fix an iteration $t$ with current model $\hat{\theta}_t$. Let $\mathcal{Q}$ denote the question space and let $\Delta(\mathcal{A})$ be the set of probability measures on the answer space $\mathcal{A}$. Let $\Pi \subset (\mathcal{Q} \rightarrow$*

---

[1] Since an LLM can, in principle, assign nonzero probability to any reasonable text continuation, including a correct solution, we treat $\alpha(\theta, q) > 0$ throughout.

$\Delta(\mathcal{A}))$ *be a finite model class, and suppose that the conditional distribution over answers induced by $D'_{p_0, \hat{\theta}_t}$ belongs to $\Pi$. Suppose that for each $q \sim p_0$ we draw $m$ ($m \geq 1$) i.i.d. candidates $a_1, a_2, \ldots, a_m \sim \pi_{\hat{\theta}_t}(\cdot \mid q)$ sequentially and keep the first accepted candidate: we include $(q, a_j)$ in the training data where $j := \min\{i \in [m] : s(q, a_i) \geq \tau\}$, and discard $q$ if no such $j$ exists. Let $n_t^{(m)}$ be the resulting number of accepted training pairs. Assume that $\operatorname{ess\,inf}_q \alpha(\hat{\theta}_t, q) > 0$. Then, with probability at least $1 - \delta$,*

$$ V_{p_0}(\hat{\theta}_{t+1}) \ \geq \ \tau \left( 1 - \frac{Z_{p_0}^{(m)}(\hat{\theta}_t)}{\alpha^{(m)}(\hat{\theta}_t)} \sqrt{\frac{2 \log\big(|\Pi|\, \delta^{-1}\big)}{n_t^{(m)}}} \right), $$

*where $\alpha^{(m)}(\hat{\theta}_t, q) := 1 - \big(1 - \alpha(\hat{\theta}_t, q)\big)^m$, $Z_{p_0}^{(m)}(\hat{\theta}_t) := \mathbb{E}_{q \sim p_0}\big[\alpha^{(m)}(\hat{\theta}_t, q)\big]$, and $\alpha^{(m)}(\hat{\theta}_t) := \operatorname{ess\,inf}_q \alpha^{(m)}(\hat{\theta}_t, q)$. Moreover, the ratio $Z_{p_0}^{(m)}(\hat{\theta}_t)/\alpha^{(m)}(\hat{\theta}_t)$ is non-increasing in $m$ and satisfies $\lim_{m \to \infty} Z_{p_0}^{(m)}(\hat{\theta}_t)/\alpha^{(m)}(\hat{\theta}_t) = 1$.*

Theorem 4.1 extends our setup in Section 3; it reduces to the setting in Section 3 by taking $m = 1$, in which case $Z_{p_0}^{(m)}(\hat{\theta}_t) = Z_{p_0}(\hat{\theta}_t)$ and $n_t^{(m)} = n_t$. Theorem 4.1 highlights the importance of finite-sample effects for characterizing self-improvement. Concretely, with infinite samples, the idealized update yields $\hat{\theta}_{t+1}$ satisfying

$$ \pi_{\hat{\theta}_{t+1}}(a \mid q) = \frac{\pi_{\hat{\theta}_t}(a \mid q)\, \mathbf{1}_{\{\, s(q,a) \geq \tau \,\}}}{\alpha(\hat{\theta}_t, q)}, $$

and hence $V_{p_0}(\hat{\theta}_{t+1}) \geq \tau$. In other words, an infinite-sample (population) update would suggest that a single iteration already guarantees performance above $\tau$ and that this guarantee is independent of $\hat{\theta}_t$, both of which are inconsistent with practice. This motivates our finite-sample regime analysis, which further shows that (i) the ratio $Z_{p_0}^{(m)}(\hat{\theta}_t)/\alpha^{(m)}(\hat{\theta}_t)$ decreases with $m$, and (ii) the effective sample size $n_t^{(m)}$ increases (in expectation) with both $n$ and $m$. Consequently, to obtain a stronger guarantee on self-improvement (i.e., a larger lower bound on $V_{p_0}(\hat{\theta}_{t+1})$), it is beneficial to increase both the question budget $n$ and the per-question answer budget $m$.

*Remark* 4.2 (*On the model class $\Pi$*). Our assumption of a finite model class $\Pi$ is consistent with prior theoretical treatments of self-improvement (Huang et al., 2025). More importantly, the effectiveness of recent work in formalizing self-improvement as tree search over a finite archive of candidate agents (Wang et al., 2026) suggests that the candidate set $|\Pi|$ is typically not very large in practice. Moreover, evidence that stronger language models admit a smaller intrinsic dimension during fine-tuning (Aghajanyan et al., 2021) indicates that $|\Pi|$ tends to be smaller for stronger base models.

## 4.2. Multi-Step Self-Improvement

We specialize to mathematical reasoning by adopting a binary reward $s(q, a) \in \{0, 1\}$, where $s(q, a) = 1$ indicates a correct (verifiable) solution and $s(q, a) = 0$ otherwise. In this regime, combined with Assumption 4.3 which posits a positive relationship between the per-question acceptance rate and the expected reward for all but a $\gamma$ fraction of questions, Corollary 4.4 relates $V_{p_0}(\hat{\theta}_{t+1})$ directly to the pretrained initialization performance $V_{p_0}(\hat{\theta}_0)$ through an iterated map.

**Assumption 4.3.** Let $\Theta$ be a small neighborhood of the pretrained initialization in which post-training is performed. Then there exist a constant $c \in (0, 1)$ and a small constant $\gamma \geq 0$ such that for any question distribution $p$ and model $\theta \in \Theta$, $\Pr_{q \sim p}\big[\alpha(\theta, q) < c\, V_p(\theta)\big] \leq \gamma$.

**Corollary 4.4.** *Consider the binary reward setting $s(q, a) \in \{0, 1\}$, where each iteration $t$ uses the same question budget $n$ with $n_t \leq n$ acceptances. Under Assumption 4.3, define*

$$F(x) \; := \; 1 - \gamma - \frac{c_\delta \nu}{c\sqrt{x - c_{\delta'}\nu}}$$

*for $x > c_{\delta'}\nu$, where $\nu := \sqrt{1/n}$, $c_\delta := \sqrt{2\log(|\Pi|\,\delta^{-1})}$, and $c_{\delta'} := \sqrt{\log(\delta'^{-1})/2}$. Then, with probability at least $1 - \delta - \delta'$, $V_{p_0}(\hat{\theta}_{t+1}) \geq F\big(V_{p_0}(\hat{\theta}_t)\big)$. Moreover, with probability at least $1 - t(\delta + \delta')$, $V_{p_0}(\hat{\theta}_t) \geq F^{\circ t}\big(V_{p_0}(\hat{\theta}_0)\big)$, where $F^{\circ t}$ denotes the $t$-fold composition of $F$.*

**Proposition 4.5.** *Under the setting of Corollary 4.4, let $\nu$ be sufficiently small such that $0 < \frac{c_\delta \nu}{c\,(1 - \gamma - c_{\delta'}\nu)^{3/2}} < \frac{2}{3\sqrt{3}}$. Let $\mathcal{I}(1, \nu) = (x_-(1, \nu), x_+(1, \nu)) \subset (c_{\delta'}\nu,\, 1 - \gamma)$ be the interval defined in Definition A.1 with $a = 1$. Then, for any non-negative integer $t$, $F^{\circ(t+1)}\big(V_{p_0}(\hat{\theta}_0)\big) > F^{\circ t}\big(V_{p_0}(\hat{\theta}_0)\big)$ and $F^{\circ t}\big(V_{p_0}(\hat{\theta}_0)\big) \in \mathcal{I}(1, \nu)$ hold if and only if $V_{p_0}(\hat{\theta}_0) \in \mathcal{I}(1, \nu)$. Moreover, $x_-(1, \nu)$ is increasing in $\nu$, $x_+(1, \nu)$ is decreasing in $\nu$, and the interval length $|\mathcal{I}(1, \nu)| = x_+(1, \nu) - x_-(1, \nu)$ is decreasing in $\nu$ and satisfies*

$$|\mathcal{I}(1, \nu)| \; \geq \; (1 - \gamma - c_{\delta'}\nu) - \frac{3\sqrt{3}}{2} \cdot \frac{c_\delta \nu}{c\sqrt{1 - \gamma - c_{\delta'}\nu}}.$$

*Remark 4.6.1* (*Moderate task difficulty benefits iterative self-improvement*). Corollary 4.4 and Proposition 4.5 suggest that iterative self-improvement admits monotonic lower-bound guarantees only when the task difficulty is neither too hard nor too easy for the pretrained initialization such that $V_{p_0}(\hat{\theta}_0) \in \mathcal{I}(1, \nu)$. Within $\mathcal{I}(1, \nu)$, better models admit more data per iteration being accepted, thereby sustaining self-improvement over successive iterations.

*Remark 4.6.2* (*Benefits of larger budgets*). Increasing the question budget $n$ (i.e., decreasing $\nu$) enlarges the interval $\mathcal{I}(1, \nu)$, and hence enlarges the set of initial performances

for which the bound sequence $\{F^{\circ t}(V_{p_0}(\hat{\theta}_0))\}_{t \geq 0}$ is guaranteed to be strictly increasing.

*Remark 4.6.3* (*Inherent upper bound*). Iterative self-improvement is inherently bounded: for $V_{p_0}(\hat{\theta}_0) \in \mathcal{I}(1, \nu)$, the lower bound cannot exceed $x_+(1, \nu)$, which is strictly below $1 - \gamma$. This provides a rationale for practical mathematical reasoning pipelines to incorporate additional optimization phases (e.g., reinforcement learning (Guo et al., 2025)) to push performance further.

# 5. Iterative Easy-to-Hard Curriculum for Self-Improvement

Combining self-improvement with an easy-to-hard curriculum across iterations has emerged as a promising approach for further improving model performance by progressively increasing the difficulty of questions encountered in different rounds (Ren et al., 2025; Koh et al., 2026; Lee et al., 2025). Despite its empirical appeal, a principled theoretical understanding of such curriculum-guided self-improvement remains limited. In this section, we extend the tools developed in Section 4 to study when and why integrating iterative self-improvement with an easy-to-hard curriculum can yield stronger self-improvement guarantees.

## 5.1. Easy-to-Hard Curriculum and Baseline

**Difficulty levels.** We assume there exist $L$ ($L \geq 2$) task distributions $p_1, \ldots, p_L$, where each $p_i$ is a valid question distribution, and the difficulty increases progressively from $p_1$ to $p_L$. Assumption 5.1 formalizes this notion via a power-law separation between adjacent tasks (in difficulty) $p_i$ and $p_{i+1}$. Concretely, under Assumption 5.1, for every $i \in [L-1]$ and every $\theta \in \Theta$, we have

$$1 \; < \; \frac{i^{-\beta'}}{(i+1)^{-\beta'}} \; \leq \; \frac{V_{p_i}(\theta)}{V_{p_{i+1}}(\theta)} \; \leq \; \frac{i^{-\beta}}{(i+1)^{-\beta}}.$$

This reflects the view that expected reward is a natural measure of task difficulty, and that the relative ordering of difficulty levels should be model-invariant for $\theta \in \Theta$. Moreover, a larger $\beta'$ corresponds to a larger difficulty ratio between adjacent tasks, while a larger uncertainty width $\Delta := \beta - \beta'$ indicates greater ambiguity in this difficulty ratio.

**Assumption 5.1.** Let $\Theta$ be a small neighborhood of the pretrained initialization in which post-training is performed. Consider $L$ question distributions $\{p_1, p_2, \ldots, p_L\}$. For $\theta \in \Theta$ and $i \in [L]$, define the expected reward $V_{p_i}(\theta) := \mathbb{E}_{(q,a) \sim D_{p_i, \theta}}\big[s(q, a)\big]$. Define

$$\beta' \; := \; \min_{i \in [L-1]} \inf_{\theta \in \Theta} \frac{\log(V_{p_i}(\theta)/V_{p_{i+1}}(\theta))}{\log(1 + 1/i)},$$

and

$$\beta := \max_{i \in [L-1]} \sup_{\theta \in \Theta} \frac{\log(V_{p_i}(\theta)/V_{p_{i+1}}(\theta))}{\log(1 + 1/i)}.$$

We assume that $0 < \beta' < \beta$.

**Easy-to-hard.** For the easy-to-hard curriculum, we consider $L$ iterations of self-improvement. At each iteration $t \in \{0, 1, \ldots, L-1\}$, we sample $n$ questions from $p_{t+1}$ to reflect progressively increasing difficulty, and perform one round of self-improvement using the current model. We initialize the curriculum with $\hat{\theta}_0^{\mathrm{E2H}} = \hat{\theta}_0$, and denote the model after iteration $t$ by $\hat{\theta}_{t+1}^{\mathrm{E2H}}$.

**Baseline.** The baseline we compare against trains for $L$ iterations using a fixed and uniform mixture over all difficulty levels. At each iteration $t \in \{0, 1, \ldots, L-1\}$, we always sample $n$ training questions from $p_0 := \frac{1}{L} \sum_{i=1}^{L} p_i$. We use the same initialization $\hat{\theta}_0^{\mathrm{B}} = \hat{\theta}_0$, and denote the model after iteration $t$ by $\hat{\theta}_{t+1}^{\mathrm{B}}$.

## 5.2. Main Results

We now present our core comparison between the final self-improvement performance under the baseline, $V_{p_0}(\hat{\theta}_L^{\mathrm{B}})$, and under the easy-to-hard curriculum, $V_{p_0}(\hat{\theta}_L^{\mathrm{E2H}})$. Theorem 5.2 provides feasibility conditions that characterize when the lower bound sequences for both training schemes are monotone across iterations, and an improvement condition under which the easy-to-hard curriculum yields a strictly tighter lower bound than the baseline. These sufficient conditions are highly predictive in practice: they closely track the trends observed in our Monte-Carlo simulations in this section and are consistent with the empirical gains on mathematical reasoning tasks reported in Section 6.

**Theorem 5.2.** *Follow the notation of Corollary 4.4. Fix all parameters except $\beta', \beta, \nu$ and $V_{p_0}(\hat{\theta}_0)$.*

(i) *Suppose feasibility conditions $\mathcal{M}_i(\beta', \beta, \nu, V_{p_0}(\hat{\theta}_0)) < 0$ hold[2] for all $i \in [4]$. Then, with high probability, the following statements hold. For the baseline, $V_{p_0}(\hat{\theta}_L^{\mathrm{B}}) \geq F^{\circ L}(V_{p_0}(\hat{\theta}_0))$, and the sequence $\{F^{\circ t}(V_{p_0}(\hat{\theta}_0))\}_{t \geq 0}$ is monotonically increasing in $t$. For the easy-to-hard curriculum, $V_{p_0}(\hat{\theta}_L^{\mathrm{E2H}}) \geq (G \circ H_{L-1} \circ H_{L-2} \circ \cdots \circ H_0)(V_{p_0}(\hat{\theta}_0))$, where for each $t \in \{0, 1, \ldots, L-1\}$,*

$$H_t(x) := 1 - \gamma - \frac{c_\delta \nu}{c\sqrt{a_t x - c_{\delta'}\nu}}, \qquad G(x) := a_L x,$$

*and $a_0 = L/\sum_{i=1}^{L} i^{-\beta'}$, $a_L = \sum_{i=1}^{L} i^{-\beta'}/L^{1-\beta'}$, $a_t = (t+1)^{-\beta}/t^{-\beta}$ for $t \in [L-1]$. Also, the sequence $\{(H_t \circ \cdots \circ H_0)(V_{p_0}(\hat{\theta}_0))\}_{t \geq 0}$ is monotonically increasing in $t$.*

---

[2]We defer the explicit forms of $\{\mathcal{M}_i\}_{i=1}^{4}$ and $\mathcal{N}$ to Definition A.2.

(ii) *If the improvement condition $\mathcal{N}(\beta', \beta, \nu, V_{p_0}(\hat{\theta}_0)) < 0$ further holds[2], then the easy-to-hard lower bound is strictly larger than the baseline lower bound: $(G \circ H_{L-1} \circ H_{L-2} \circ \cdots \circ H_0)(V_{p_0}(\hat{\theta}_0)) > F^{\circ L}(V_{p_0}(\hat{\theta}_0))$.*

**Interpreting $\{\mathcal{M}_i < 0\}$.** To enable meaningful comparisons across iterations, feasibility conditions $\{\mathcal{M}_i < 0\}_{i=1}^{4}$ in Theorem 5.2 rule out degenerate regimes in which the evolution of the expected reward lower bound becomes ill-defined or fail to be monotonically increasing. We provide a concrete interpretation of the region $\{\mathcal{M}_i < 0\}_{i=1}^{4}$ in Remark 5.4.

**Corollary 5.3.** *Follow the setting of Theorem 5.2. Let $\mathcal{I}(2^{-\beta}, \nu) = (x_-(2^{-\beta}, \nu), x_+(2^{-\beta}, \nu))$ be the interval in Definition A.1 with $a = 2^{-\beta}$. Then the feasibility conditions $\mathcal{M}_i(\beta', \beta, \nu, V_{p_0}(\hat{\theta}_0)) < 0$ for all $i \in [4]$ are equivalent to $V_{p_0}(\hat{\theta}_0) \in \mathcal{I}_{\mathcal{M}}(\beta', \beta, \nu)$, where $\mathcal{I}_{\mathcal{M}}(\beta', \beta, \nu) := (x_-(2^{-\beta}, \nu), \frac{2^{-\beta}}{a_0} x_+(2^{-\beta}, \nu))$. Moreover, the interval length $|\mathcal{I}_{\mathcal{M}}(\beta', \beta, \nu)|$ satisfies*

$$2^\beta c_{\delta'} \nu \leq |\mathcal{I}_{\mathcal{M}}(\beta', \beta, 0)| - |\mathcal{I}_{\mathcal{M}}(\beta', \beta, \nu)| \leq 2^\beta c_{\delta'} \nu$$
$$+ \frac{3\sqrt{3}}{2} \cdot \frac{c_\delta \nu}{c\sqrt{2^{-\beta}(1-\gamma) - c_{\delta'}\nu}}.$$

*Remark* 5.4 (*Feasibility disfavors small budgets and large adjacent difficulty ratios*). Corollaries 5.3 and B.3 together imply that $|\mathcal{I}_{\mathcal{M}}(\beta', \beta, \nu)|$ decreases in $\nu$, $\beta'$, and $\beta$, and that its shrinkage rate in $\nu$ is $\Theta(\nu)$ as $\nu \to 0$. Although $\{\mathcal{M}_i < 0\}_{i=1}^{4}$ only enforces feasibility (rather than directly characterizing when the easy-to-hard curriculum improves over the baseline), we generally prefer $|\mathcal{I}_{\mathcal{M}}(\beta', \beta, \nu)|$ not to be too small. Consequently, (i) an overly small question budget $n$ and (ii) overly large difficulty ratios between adjacent tasks are both undesirable from the standpoint of feasibility. Finally, Figure 1 shows that the condition $V_{p_0}(\hat{\theta}_0) \in \mathcal{I}_{\mathcal{M}}(\beta', \beta, \nu)$ closely matches the behavior observed in direct Monte-Carlo simulations, making $\{\mathcal{M}_i < 0\}_{i=1}^{4}$ a useful proxy.

**Interpreting $\mathcal{N} < 0$.** $\mathcal{N} < 0$ serves as the key criterion for improvement: it guarantees that the easy-to-hard curriculum attains a strictly larger final lower bound than the baseline. A concrete interpretation of $\mathcal{N} < 0$ is provided in Remarks 5.7.1–5.7.4. Notably, all the resulting predictions based on the improvement condition $\mathcal{N} < 0$ closely match the trends observed in direct Monte-Carlo simulations in Figure 2.

**Proposition 5.5.** *Follow the setting of Theorem 5.2. The improvement condition $\mathcal{N}(\beta', \beta, \nu, V_{p_0}(\hat{\theta}_0)) < 0$ is equivalent to $V_{p_0}(\hat{\theta}_0) \in \mathcal{I}_{\mathcal{N}}(\beta', \beta, \nu)$, where $\mathcal{I}_{\mathcal{N}}(\beta', \beta, \nu) := (x(\beta', \beta, \nu), 1 - \gamma)$. For fixed $(\beta', \beta)$, we write $x(\nu) := x(\beta', \beta, \nu)$ for brevity. Then, $x(\nu)$ is monotonically increas-*

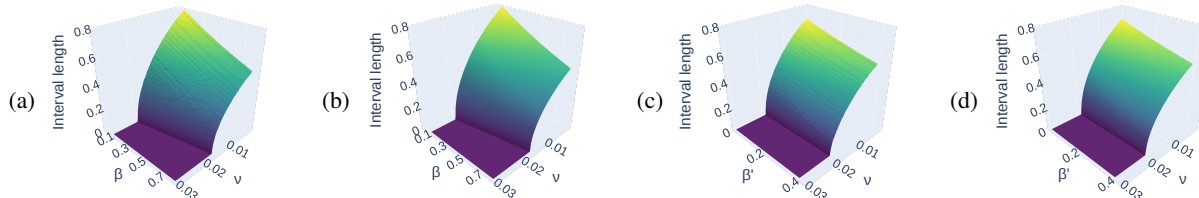

*Figure 1.* **Feasible initialization region.** Panels (a,c) report Monte-Carlo estimates of the length of the initialization interval $V_{p_0}(\hat{\theta}_0)$ for which $\{F^{\circ t}(V_{p_0}(\hat{\theta}_0))\}_{t \geq 0}$ and $\{(H_t \circ \cdots \circ H_0)(V_{p_0}(\hat{\theta}_0))\}_{t \geq 0}$ are both monotonically increasing in $t$, under different $(\beta', \beta, \nu)$ settings. Panels (b,d) show the length of the feasibility interval $\mathcal{I}_{\mathcal{M}}(\beta', \beta, \nu)$ in Corollary 5.3. Panels (a,b): fix $\beta' = 0.1$ and vary $(\beta, \nu)$. Panels (c,d): fix $\beta = 0.4$ and vary $(\beta', \nu)$.

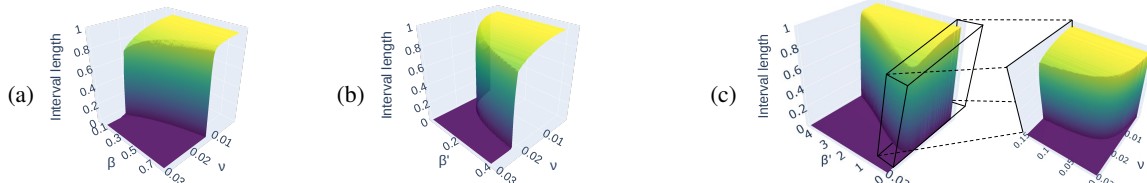

*Figure 2.* **Improvement initialization region.** Panels (a)-(c) report Monte-Carlo estimates of the length of the initialization interval $V_{p_0}(\hat{\theta}_0)$ for which $(G \circ H_{L-1} \circ H_{L-2} \circ \cdots \circ H_0)(V_{p_0}(\hat{\theta}_0)) > F^{\circ L}(V_{p_0}(\hat{\theta}_0))$ holds under different $(\beta', \beta, \nu)$ settings. Panel (a): fix $\beta' = 0.1$ and vary $(\beta, \nu)$. Panel (b): fix $\beta = 0.4$ and vary $(\beta', \nu)$. Panel (c): fix $\Delta = 0.1$ and vary $(\beta', \nu)$; the same panel also includes a zoomed-in view for small $\beta'$.

*ing in $\nu$ and satisfies $x(0) = 0$,*

$$x'(\nu) = \frac{c_{\delta'}}{a_0} + \frac{2}{a_0}\left(\frac{c_\delta}{c(1-\gamma)}\right)^2 \nu + O(\nu^{5/3}) \quad as\ \nu \to 0.$$

*Moreover, there exists a unique critical value $\nu_c > 0$ and a constant $C(\nu_c) > 0$ such that*

$$x'(\nu) = \frac{C(\nu_c)}{(\nu_c - \nu)^3}\left(1 + O(\nu_c - \nu)\right) \quad as\ \nu \uparrow \nu_c.$$

**Corollary 5.6** (Informal version; formal version stated in Corollary C.4). *In Proposition 5.5, fix any initialization $V_{p_0}(\hat{\theta}_0) \in (0, 1-\gamma)$, and let $\nu^\star(\beta', \beta)$ be defined by the threshold equation $x(\beta', \beta, \nu^\star(\beta', \beta)) = V_{p_0}(\hat{\theta}_0)$. Then*

$$\nu^\star(\beta', \beta) = \sup\{\nu > 0 : \mathcal{N}(\beta', \beta, \nu, V_{p_0}(\hat{\theta}_0)) < 0\}.$$

*Moreover,* (i) *fixing $\beta'$, for $\beta > \beta'$, $\nu^\star(\beta', \beta)$ is decreasing in $\beta$;*

(ii) *fixing $\beta$, for $\beta' \in (0, \beta)$, $\nu^\star(\beta', \beta)$ is increasing in $\beta'$;*

(iii) *fixing $\Delta = \beta - \beta'$ and writing $\nu^\star(\beta', \beta)$ as $\nu^\star(\beta', \beta' + \Delta)$, when $\beta'$ is sufficiently small, $\nu^\star(\beta', \beta' + \Delta)$ is increasing in $\beta'$ and scales as $\Theta(\beta')$; when $\beta'$ is sufficiently large, $\nu^\star(\beta', \beta' + \Delta) \to 0$, with the tail bound $O(2^{-\beta'})$. Furthermore, for any finite range $[0, B]$ whose endpoint $B$ is above an explicit computable threshold, in the large-sample regime, $\nu^\star(\beta', \beta' + \Delta)$ is first increasing and then decreasing in $\beta'$, with a unique optimizer on this range.*

*Remark 5.7.1* (*Phase transition with respect to the question budget*). Proposition 5.5 shows that the interval length

$|\mathcal{I}_{\mathcal{N}}(\beta', \beta, \nu)|$ decreases as $\nu$ increases. Moreover, the shrinkage rate is mild when $\nu$ is small (since $c_{\delta'}$ is typically small and $a_0 > 1$), but becomes steep as $\nu$ approaches the critical value $\nu_c$ ($x'(\nu)$ blows up on the order of $\Theta((\nu_c - \nu)^{-3})$). Equivalently, decreasing the question budget $n$ makes it harder for the easy-to-hard curriculum to provably outperform the baseline, and there is a critical sample size such that as $n$ decreases toward this threshold, the range of initializations $V_{p_0}(\hat{\theta}_0)$ for which easy-to-hard is provably advantageous collapses sharply.

*Remark 5.7.2* (*Smaller uncertainty in the adjacent difficulty ratio is better*). Parts (i)-(ii) of Corollary 5.6 imply that, whether we fix $\beta'$ or $\beta$, as the uncertainty width $\Delta = \beta - \beta'$ increases, the maximal admissible $\nu$ (and hence the minimal question budget $n$) for $\mathcal{N} < 0$ becomes more stringent.

*Remark 5.7.3* (*Moderate difficulty ratios between adjacent tasks are most favorable*). Part (iii) of Corollary 5.6 further shows that, when $\Delta$ is fixed, as the difficulty ratios between adjacent tasks (captured by $\beta'$) increase, the minimal admissible sample budget $n$ required for the easy-to-hard curriculum to be provably better than the baseline decreases in the small-ratio regime, but diverges to infinity once the difficulty ratios between adjacent tasks become sufficiently large. Moreover, over any sufficiently large finite interval of $\beta'$, in the large-sample regime, there exists a unique optimal $\beta'$ that minimizes the required sample budget.

*Remark 5.7.4* (*Improvement dominates for small budgets, while feasibility dominates for large budgets*). By Corollary 5.3 and Proposition 5.5, for any fixed $(\beta', \beta)$, both $|\mathcal{I}_{\mathcal{M}}(\beta', \beta, \nu)|$ and $|\mathcal{I}_{\mathcal{N}}(\beta', \beta, \nu)|$ decrease as $\nu$ increases.

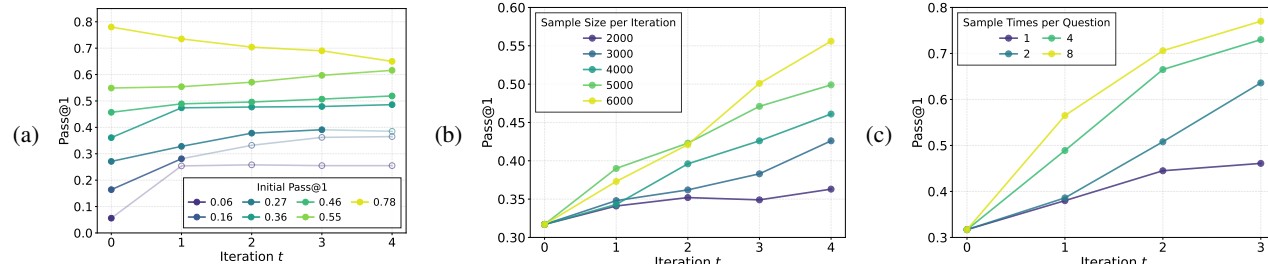

*Figure 3.* Iterative self-improvement results on the synthetic shortest path task. Panel (a) shows the self-improvement trajectories of a fixed $\hat{\theta}_0$ across tasks with different initial Pass@1 accuracies; hollow markers and faded line segments indicate model collapse (Pass@1$=0$ for at least one target distance $l$). Panel (b) shows the performance under different question budgets $n$, with $\hat{\theta}_0$ and the initial Pass@1 fixed. Panel (c) shows the performance under different per-question answer budgets $m$, with $\hat{\theta}_0$ and the initial Pass@1 fixed.

Moreover, since $c_{\delta'}/a_0 < 2^\beta c_{\delta'}$, we have the following dichotomy: as $\nu \to 0$, the shrinkage of the admissible range of $V_{p_0}(\hat{\theta}_0)$ is dominated by the feasibility conditions $\{\mathcal{M}_i < 0\}_{i=1}^4$; whereas as $\nu$ approaches $\nu_c$, the shrinkage is dominated by the improvement condition $\mathcal{N} < 0$.

# 6. Experiment

In this section, we empirically validate the main theoretical predictions developed in Sections 4 and 5 across diverse mathematical reasoning tasks.

## 6.1. Experiments on Synthetic Tasks

### 6.1.1. EXPERIMENTAL SETUP

**Shortest path.** Given the capability of LLMs to solve graph problems in natural language (Wang et al., 2023), we study self-improvement on a shortest path task using synthetically generated graphs. We consider a directed unweighted graph $\mathcal{G}$. Our task is: given $\mathcal{G}$ and two distinct vertices $v_s \neq v_t$ in $\mathcal{G}$, predict the shortest path length $l$, i.e., the minimum number of edges among all directed paths from $v_s$ to $v_t$; if no such path exists, we set $l = -1$.

**Datasets and model training setup.** Across experiments, we construct synthetic datasets from a sample pool consisting of a large collection of distinct graphs $\mathcal{G}$ together with vertex pairs $(v_s, v_t)$, spanning different choices of the number of nodes $N$, expected out-degree $\bar{d}$, and target distance $l$. This synthetic shortest path task enables direct control of several key variables, including the initialization expected reward $V_{p_0}(\hat{\theta}_0)$, sample budgets $n$ and $m$, and difficulty ratios between adjacent tasks. We use LLAMA-3.2-1B-INSTRUCT as our base LLM (Grattafiori et al., 2024) in Section 6.1. We follow the procedure described in Section 3 and the easy-to-hard/baseline setup in Section 5.1 to run iterative self-improvement. More details of dataset construction, parameter control, and self-improvement finetuning are provided in Appendix D.1.

**Evaluation metric.** In the binary reward setting, the expected reward is equivalent to the population Pass@1. Therefore, we report the Pass@1 accuracy on a held-out test set sampled from $p_0$ as our evaluation metric.

### 6.1.2. EXPERIMENTAL RESULTS

**Iterative self-improvement.** Figure 3(a) shows the iterative self-improvement performance under tasks of varying difficulty, where we construct task sets such that the initial Pass@1 accuracy (corresponding to $V_{p_0}(\hat{\theta}_0)$) ranges from 6% to 78%. When the task is overly easy, we observe a decreasing trend in the test Pass@1 across iterations, whereas when the task is overly hard, the Pass@1 becomes unstable and may collapse. These trends are consistent with the analysis in Remark 4.6.1, which predicts that effective iterative self-improvement is only guaranteed to occur in a moderate difficulty regime. We also note that the improvement in Pass@1 often slows down after $t = 2$ and the curves begin to plateau at values clearly below 1, which is consistent with Remark 4.6.3. Moreover, Figure 3(b)-(c) demonstrate that increasing either the question budget $n$ or the per-question answer budget $m$ consistently improves self-improvement performance. This aligns with the finite-sample interpretation in Section 4.1.

**Iterative self-improvement with easy-to-hard curriculum.** In Figure 4(a), each curve varies the difficulty ratios between adjacent tasks (controlled by $\beta'$) while keeping the initial Pass@1 (corresponding to $V_{p_0}(\hat{\theta}_0)$) fixed. Across different values of the initial Pass@1, the final Pass@1 gap between easy-to-hard and the baseline exhibits an overall trend of first increasing and then decreasing as $\beta'$ grows, with the largest gaps typically attained around $\beta' \in [0.2, 0.5]$. This aligns with the discussion in Remark 5.7.3, which suggests that moderate difficulty ratios between adjacent tasks are desirable.

Figure 4(b) reports results under different initial Pass@1 accuracies, with a fixed relative difficulty across tasks (i.e., fixed $\beta'$ and $\Delta$). First, we observe that larger question bud-

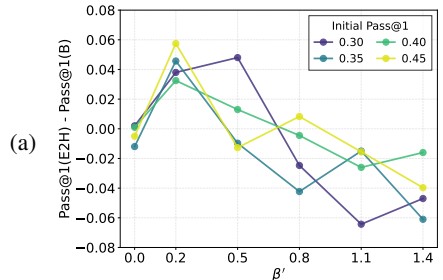
(a)

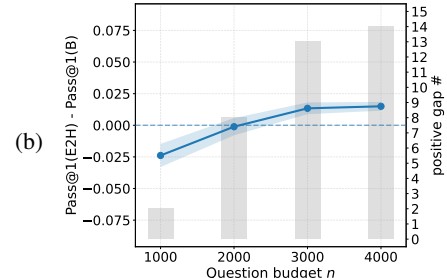
(b)

*Figure 4.* Iterative self-improvement with easy-to-hard curriculum on the synthetic shortest path task. Panel (a) fixes $\Delta = 0.04$ and $\hat{\theta}_0$, and shows for different initial Pass@1 accuracies, the final Pass@1 gap between easy-to-hard and the baseline (i.e., $V_{p_0}(\hat{\theta}_L^{\text{E2H}}) - V_{p_0}(\hat{\theta}_L^{\text{B}})$) as a function of the adjacent task difficulty ratio (captured by $\beta'$). Panel (b) fixes $\Delta = 0.04$ and $\beta' = 0.25$, and shows for different initial Pass@1 accuracies (spanning 35%–55%), how the final Pass@1 gap varies with the question budget $n$. The solid line reports the mean gap across 15 initializations and the shaded region indicates $\pm 1$ standard error; the gray bars (right axis) show the number of initializations with a positive gap at each $n$.

gets $n$ lead to a larger final Pass@1 gap of easy-to-hard over the baseline on average; moreover, as $n$ increases, the final Pass@1 gap varies less across different initial performances, leading to a smaller standard error (the number of initializations is fixed). Second, for a diverse set of initial Pass@1 accuracies in the range of 35%–55%, the question budget $n$ at which easy-to-hard starts to outperform the baseline mostly clusters in a relatively narrow range, roughly between 2000 and 3000. This agrees with the phase transition behavior predicted in Remark 5.7.1.

## 6.2. Experiments on Standard Mathematical Reasoning Benchmarks

### 6.2.1. EXPERIMENTAL SETUP

**Datasets and model training setup.** We further validate our theoretical analysis on standard mathematical reasoning benchmarks, including GSM8K (Cobbe et al., 2021) and DeepMind Mathematics (Saxton et al., 2019). GSM8K contains 8.5K grade-school-level math word problems. Deep-Mind Mathematics consists of mathematical reasoning problems spanning multiple categories (e.g., algebra, arithmetic, and numbers) with each category further containing a large collection of fine-grained modules. Notably, DeepMind Mathematics comes with a native easy/medium/hard difficulty partition, which naturally supports our modeling of the easy-to-hard curriculum. Unless otherwise specified, we use QWEN3-8B (Yang et al., 2025) as the base model throughout Section 6.2. Additional base models considered in this section include LLAMA-3.2-1B-INSTRUCT, LLAMA-3.2-3B-INSTRUCT (Grattafiori et al., 2024), QWEN2.5-3B-INSTRUCT, QWEN2.5-7B-INSTRUCT (Yang et al., 2024), and QWEN3-32B (Yang et al., 2025). As in Section 6.1, we follow the setup in Section 3 and Section 5.1 to run iterative self-improvement, and use Pass@1 accuracy as the evaluation metric. More details on dataset splits and self-improvement finetuning are provided in Appendix D.2.

### 6.2.2. EXPERIMENTAL RESULTS

**Iterative self-improvement.** Table 1(a) reports the self-improvement results on GSM8K and on the easy split of the nine DeepMind Mathematics modules with the highest initial QWEN3-8B accuracy. Across different base models, these results clearly show that when a task is too easy or too hard relative to the model, self-improvement tends to be limited or even deteriorate, whereas moderate task difficulty is more conducive to sustained iterative self-improvement. Moreover, Table 7 and Table 8 show that as the number of iterations increases, the incremental improvement from self-improvement tends to decrease. Table 1(b) further shows how the degree of self-improvement changes with the question budget $n$ and the per-question answer budget $m$. The results indicate that larger values of $n$ and $m$ are both more favorable to self-improvement.

Overall, these findings are consistent with the synthetic task results in Section 6.1.2, and provide empirical support for Remark 4.6.1, Remark 4.6.3, and the finite-sample analysis in Section 4.1. It is worth noting that, despite minor differences in self-improvement setups, a range of empirical studies further echo our findings on a broader set of real-world benchmarks: self-improvement tends to favor a moderate task difficulty regime (Singh et al., 2024), benefits from larger budgets ($n$ (Singh et al., 2024; Wilf et al., 2025) and $m$ (Zeng et al., 2025; Bansal et al., 2025; Yao et al., 2026)), and often exhibits a saturation limit (Song et al., 2025).

**Iterative self-improvement with easy-to-hard curriculum.** Beyond the synthetic experiments, here we further validate our analysis using the dataset's native difficulty partition. Concretely, for the DeepMind Mathematics dataset, we vary the difficulty ratios between adjacent tasks through a parameter $\rho$. For the $L = 3$ iteration self-improvement setting, let $\mathbf{W} = (w_{ij}) \in \mathbb{R}^{3 \times 3}$, where $w_{ij}$ denotes the sampling weight assigned at iteration $i-1$ to the $j$-th difficulty

| Model | GSM8K | | DeepMind Math | | | Budget | Value | Absolute Improvement |
|---|---|---|---|---|---|---|---|---|
| | Initial Pass@1 | Absolute Improvement | Initial Pass@1 | Absolute Improvement | | | | |
| Llama-3.2-1B-Instruct | 2.96 | -0.28 | 18.89 | 17.20 | | $n$ | 600 | -4.06 |
| Llama-3.2-3B-Instruct | 6.07 | -0.46 | 37.00 | 17.44 | | | 1200 | 0.56 |
| Qwen2.5-3B-Instruct | 10.39 | 3.23 | 55.89 | 14.53 | | | 1800 | 2.73 |
| Qwen2.5-7B-Instruct | 18.12 | 2.07 | 74.00 | 8.98 | | | 2400 | 3.15 |
| Qwen3-8B | 27.52 | 3.15 | 85.11 | 7.67 | | | 1 | 3.15 |
| Qwen3-32B | 38.59 | 5.18 | 97.89 | -1.20 | | $m$ | 4 | 3.99 |
| | | | | | | | 16 | 5.34 |
| (a) | | | | | | (b) | | |

*Table 1.* Summary of iterative self-improvement results on standard mathematical reasoning benchmarks. (a) Summary across different base models on GSM8K and DeepMind Mathematics. The table reports the initial Pass@1 accuracy and the absolute Pass@1 improvement after three self-improvement iterations. (b) Summary of budget effects on GSM8K for QWEN3-8B. The upper block varies the question budget $n$, and the lower block varies the per-question answer budget $m$. The table reports the absolute Pass@1 improvement after three self-improvement iterations. All Pass@1 values are percentages. Complete self-improvement trajectories are reported in Appendix Tables 7, 8, and 9.

| Category | Module | $\rho$ | | | | | |
|---|---|---|---|---|---|---|---|
| | | 0.0 | 0.2 | 0.4 | 0.6 | 0.8 | 1.0 |
| algebra | linear_2d | 0.00 | $0.04_{\pm 0.27}$ | $0.84_{\pm 0.11}$ | $\underline{2.52}_{\pm 0.54}$ | $1.96_{\pm 0.27}$ | $2.04_{\pm 0.25}$ |
| arithmetic | add_sub_multiple | 0.00 | $2.76_{\pm 2.23}$ | $7.68_{\pm 2.97}$ | $11.60_{\pm 3.17}$ | $\underline{17.08}_{\pm 2.12}$ | $10.64_{\pm 3.64}$ |
| | div | 0.00 | $0.16_{\pm 0.45}$ | $0.42_{\pm 0.50}$ | $\underline{0.98}_{\pm 0.61}$ | $0.88_{\pm 0.49}$ | $-0.54_{\pm 0.50}$ |
| | mixed | 0.00 | $-1.06_{\pm 1.10}$ | $1.64_{\pm 1.36}$ | $3.40_{\pm 1.20}$ | $\underline{4.62}_{\pm 0.51}$ | $3.20_{\pm 1.54}$ |
| | mul_div_multiple | 0.00 | $2.60_{\pm 1.62}$ | $4.76_{\pm 2.41}$ | $9.56_{\pm 2.20}$ | $\underline{12.32}_{\pm 2.09}$ | $10.48_{\pm 2.33}$ |
| | nearest_integer_root | 0.00 | $2.38_{\pm 2.80}$ | $\underline{2.50}_{\pm 1.09}$ | $-2.72_{\pm 2.14}$ | $-4.44_{\pm 3.14}$ | $-6.64_{\pm 2.55}$ |
| numbers | div_remainder | 0.00 | $2.80_{\pm 0.90}$ | $2.16_{\pm 0.72}$ | $3.78_{\pm 0.42}$ | $\underline{3.80}_{\pm 0.88}$ | $2.90_{\pm 0.67}$ |
| | gcd | 0.00 | $0.22_{\pm 0.96}$ | $0.32_{\pm 0.62}$ | $\underline{0.78}_{\pm 0.74}$ | $0.04_{\pm 0.91}$ | $-0.44_{\pm 1.11}$ |
| | lcm | 0.00 | $0.12_{\pm 0.60}$ | $0.26_{\pm 0.65}$ | $1.04_{\pm 0.65}$ | $0.96_{\pm 0.86}$ | $\underline{2.66}_{\pm 1.04}$ |
| | place_value | 0.00 | $0.02_{\pm 0.73}$ | $\underline{0.28}_{\pm 0.73}$ | $0.02_{\pm 0.54}$ | $-0.44_{\pm 0.78}$ | $-3.12_{\pm 1.52}$ |

*Table 2.* Final Pass@1 gap between easy-to-hard and the baseline on the DeepMind Mathematics Dataset using QWEN3-8B. Difficulty follows the dataset's native easy/medium/hard split. $\rho$ interpolates between fixed-mixture training in every round ($\rho=0$, baseline) and a fully staged curriculum ($\rho=1$: iteration 0 easy, iteration 1 medium, iteration 2 hard). We run experiments on all modules in the algebra, arithmetic, and numbers categories. To focus on modules for which curriculum effects are meaningfully measurable, we exclude modules whose canonical difficulty split yields $\beta' < 0.1$, or whose gap variation across $\rho$ is smaller than 1%. Underlined entries denote, for each module, the maximum gap across $\rho$. All values are percentages. Standard errors are shown as subscripts.

level, with $j = 1, 2, 3$ corresponding to the easy, medium, and hard splits, respectively. Let $\mathbf{U}$ be the matrix whose entries are all $1/3$, and let $\mathbf{I}$ be the identity matrix. We define $\mathbf{W} = (1 - \rho)\mathbf{U} + \rho\mathbf{I}$, where $\rho \in [0, 1]$. Thus, when $\rho = 0$, we have $\mathbf{W} = \mathbf{U}$, which corresponds to the baseline where each iteration samples uniformly from all three difficulty levels; when $\rho = 1$, we have $\mathbf{W} = \mathbf{I}$, which corresponds to a fully staged easy-to-hard curriculum in which each iteration samples exclusively from one difficulty level. As $\rho$ increases from 0 to 1, the difficulty ratios between adjacent tasks become larger.

Appendix Figure 5 shows that the estimates of $\beta$ and $\beta'$ both increase clearly with $\rho$, suggesting that these $\beta$ and $\beta'$ provide effective proxies for the dataset's native notion of difficulty separation. Table 2 further reports, for modules in the algebra, arithmetic, and numbers categories, the final Pass@1 gap between easy-to-hard and the baseline as a function of $\rho$. Across modules, as $\rho$ increases from 0 to 1, the final Pass@1 gap typically exhibits a trend of first increasing and then decreasing, with the largest gain

attained at an interior value of $\rho$. These results support Remark 5.7.3, which predicts that moderate difficulty ratios between adjacent tasks are the most favorable for easy-to-hard scheduling to realize its advantage within the iterative self-improvement framework.

## 7. Conclusion

In this work, we developed a task-centric framework for understanding LLM self-improvement. For a single task, our finite-sample analysis characterizes key factors (e.g., task difficulty and sampling budget) that determine when (multi-step) self-improvement happens, and explain eventual saturation of such improvement. Beyond single-task training, we further provide theoretical guidance to identify regimes where easy-to-hard scheduling yields a stronger lower bound guarantee than fixed-mixture training, and highlight the role of appropriate adjacent task difficulty ratios and a critical sample size. Finally, our predictions are supported by experiments on a synthetic shortest path task and multiple standard mathematical reasoning benchmarks.

## Acknowledgments

QL acknowledges support of NSF DMS-2523382 and DOE Office of Science under Award #DE-SC0024721. YS was supported in part by the National Institutes of Health (grant no. 1R01EB036530-01A1).

## Impact Statement

This paper advances the theoretical understanding of iterative self-improvement and easy-to-hard curricula for post-training large language models on reasoning tasks. The primary intended impact is to provide principled guidance for designing more reliable and effective training procedures for self-improvement. There are many further potential societal consequences of our work, none of which we feel must be specifically highlighted here.

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

# Appendix: Proofs

## A. Notation and Deferred Definitions

### A.1 NOTATION SUMMARY

| Symbol | Meaning |
| --- | --- |
| $q$ | question |
| $a$ | answer |
| $s(q, a)$ | reward; $s(q, a) \in [0, 1]$, where larger values indicate a better answer $a$ to the question $q$ |
| $\tau$ | acceptance threshold; $\tau \in (0, 1]$ for filtering $s(q, a) \geq \tau$ |
| $\theta, \hat{\theta}$ | model parameters; $\hat{\theta}$ denotes the empirical model; includes variants with superscripts or subscripts[3] |
| $p(\cdot)$ | question distribution; includes variants with subscripts |
| $\pi_\theta(\cdot \mid q)$ | answer distribution; conditional distribution induced by $\theta$ given $q$ |
| $D_{p,\theta}(q, a)$ | $D_{p,\theta}(q, a) = p(q)\pi_\theta(a \mid q)$ |
| $V_p(\theta)$ | expected reward; $V_p(\theta) = \mathbb{E}_{q \sim p(\cdot)} \mathbb{E}_{a \sim \pi_\theta(\cdot\mid q)} \big[ s(q, a) \big]$ |
| $\alpha(\theta, q)$ | per-question acceptance rate; $\alpha(\theta, q) = \Pr_{a \sim \pi_\theta(\cdot\mid q)}[s(q, a) \geq \tau]$ |
| $Z_p(\theta)$ | global acceptance rate; $Z_p(\theta) = \mathbb{E}_{q \sim p(\cdot)}[\alpha(\theta, q)]$ |
| $n$ | question sampling budget; total number of sampled questions per iteration |
| $m$ | per-question answer budget; number of sampled answers per question (unless otherwise stated, $m = 1$) |
| $\nu$ | $\nu = \sqrt{1/n}$ |
| $\Pi$ | model class; see Thm. 4.1 for details |
| $c, \gamma$ | reward-acceptance coupling constants; see Assump. 4.3 for details |
| $\delta, \delta'$ | failure probabilities; see Thm. 4.1 and Cor. 4.4 for details |
| $c_\delta, c_{\delta'}$ | $c_\delta = \sqrt{2\log(|\Pi|\,\delta^{-1})}$, $c_{\delta'} = \sqrt{\log(\delta'^{-1})/2}$ |
| $L$ | total number of iterations |
| $\beta', \beta$ | difficulty separation exponents; controls the difficulty ratios between adjacent tasks; see Assump. 5.1 for details |
| $\Delta$ | difficulty uncertainty width; $\Delta = \beta - \beta'$ |

*Table 3.* Common notation used throughout the paper.

### A.2 DEFERRED DEFINITIONS AND EXPLICIT EXPRESSIONS

For readability, several auxiliary quantities are only referenced in the main text. Here we collect their explicit definitions and expressions.

**Definition A.1** ($\mathcal{I}(a, \nu)$). Suppose $(a, \nu)$ are chosen such that

$$1 - \gamma - \frac{c_{\delta'}\nu}{a} > 0 \qquad \text{and} \qquad 0 < \frac{a\,c_\delta\nu}{c\,(a(1-\gamma) - c_{\delta'}\nu)^{3/2}} < \sqrt{\frac{4}{27}}.$$

Then the equation

$$y(1 - y)^2 = \left( \frac{a\,c_\delta\nu}{c\,(a(1-\gamma) - c_{\delta'}\nu)^{3/2}} \right)^2$$

admits two solutions in $(0, 1)$, denoted by $y_-(a, \nu) < y_+(a, \nu)$. Set

$$x_-(a, \nu) = \frac{c_{\delta'}\nu}{a} + \left( 1 - \gamma - \frac{c_{\delta'}\nu}{a} \right) y_-(a, \nu), \qquad x_+(a, \nu) = \frac{c_{\delta'}\nu}{a} + \left( 1 - \gamma - \frac{c_{\delta'}\nu}{a} \right) y_+(a, \nu).$$

We define

$$\mathcal{I}(a, \nu) := \big( x_-(a, \nu),\, x_+(a, \nu) \big) \subset \left( \frac{c_{\delta'}\nu}{a},\, 1 - \gamma \right).$$

---

[3] A subscript $t$ ($0 \leq t \leq L$) denotes the model before iteration $t$ (or after iteration $t - 1$). Superscripts B and E2H refer to the baseline and the easy-to-hard curriculum in Section 5.1, respectively.

**Definition A.2** ($\{\mathcal{M}_i\}_{i=1}^4$ and $\mathcal{N}$). Let $x_-(2^{-\beta}, \nu)$ and $x_+(2^{-\beta}, \nu)$ be the endpoints of the interval $\mathcal{I}(2^{-\beta}, \nu)$ defined in Definition A.1 with $a = 2^{-\beta}$. Let, $a_0 = L / \sum_{i=1}^L i^{-\beta'}$ and $a_L = \sum_{i=1}^L i^{-\beta'} / L^{1-\beta'}$. Then, we define

$$\mathcal{M}_1\big(\beta', \beta, \nu, V_{p_0}(\hat{\theta}_0)\big) := x_-(2^{-\beta}, \nu) - V_{p_0}(\hat{\theta}_0),$$

$$\mathcal{M}_2\big(\beta', \beta, \nu, V_{p_0}(\hat{\theta}_0)\big) := V_{p_0}(\hat{\theta}_0) - x_+(2^{-\beta}, \nu),$$

$$\mathcal{M}_3\big(\beta', \beta, \nu, V_{p_0}(\hat{\theta}_0)\big) := x_-(2^{-\beta}, \nu) - \left(1 - \gamma - \frac{c_\delta \nu}{c\sqrt{a_0 V_{p_0}(\hat{\theta}_0) - c_{\delta'}\nu}}\right),$$

$$\mathcal{M}_4\big(\beta', \beta, \nu, V_{p_0}(\hat{\theta}_0)\big) := \left(1 - \gamma - \frac{c_\delta \nu}{c\sqrt{a_0 V_{p_0}(\hat{\theta}_0) - c_{\delta'}\nu}}\right) - x_+(2^{-\beta}, \nu),$$

Next, define

$$\mathcal{N}\big(\beta', \beta, \nu, V_{p_0}(\hat{\theta}_0)\big) := -\frac{1}{2}(a_L - 1)(1 - \gamma) - \frac{c_\delta \nu}{c\sqrt{1 - \gamma - c_{\delta'}\nu}} \cdot \frac{1 - \left(\frac{c_\delta \nu}{2c(1-\gamma-c_{\delta'}\nu)^{3/2}}\right)^{L-1}}{1 - \frac{c_\delta \nu}{2c(1-\gamma-c_{\delta'}\nu)^{3/2}}}$$

$$+ a_L \left[\frac{c_\delta \nu}{c\sqrt{2^{-\beta}(1-\gamma) - c_{\delta'}\nu}} \cdot \frac{1}{1 - \frac{c_\delta \nu}{2c\left(2^{-\beta}\left(1-\gamma-\frac{c_\delta \nu}{c\sqrt{a_0 V_{p_0}(\hat{\theta}_0)-c_{\delta'}\nu}}\right)-c_{\delta'}\nu\right)^{3/2}}} \cdot e^{-\beta/L} \right.$$

$$\left. + \left(\frac{c_\delta \nu}{2c\left(2^{-\beta}\left(1-\gamma-\frac{c_\delta \nu}{c\sqrt{a_0 V_{p_0}(\hat{\theta}_0)-c_{\delta'}\nu}}\right)-c_{\delta'}\nu\right)^{3/2}}\right)^{L-1} L^{-\beta} \cdot \frac{c_\delta \nu}{c\sqrt{a_0 V_{p_0}(\hat{\theta}_0)-c_{\delta'}\nu}}\right].$$

# B. Proofs for Section 4

## B.1 PROOF OF THEOREM 4.1

*Proof.* At iteration $t$ we obtain $n_t^{(m)}$ accepted samples $\{(q_i, a_i)\}_{i=1}^{n_t^{(m)}} \sim D'^{(m)}_{p_0, \hat{\theta}_t}$. Under this scheme, the population joint distribution of accepted pairs can be written as

$$D'^{(m)}_{p_0, \hat{\theta}_t}(q, a) = p'^{(m)}_{p_0, \hat{\theta}_t}(q)\, p'_{\hat{\theta}_t}(a \mid q),$$

where the marginal over questions and the conditional over answers are, respectively,

$$p'^{(m)}_{p_0, \hat{\theta}_t}(q) = \frac{p_0(q)\, \alpha^{(m)}(\hat{\theta}_t, q)}{Z_{p_0}^{(m)}(\hat{\theta}_t)}, \qquad p'_{\hat{\theta}_t}(a \mid q) = \frac{\pi_{\hat{\theta}_t}(a \mid q)\, \mathbf{1}_{\{s(q,a) \geq \tau\}}}{\alpha(\hat{\theta}_t, q)},$$

where

$$\alpha(\hat{\theta}_t, q) := \Pr_{a \sim \pi_{\hat{\theta}_t}(\cdot \mid q)}\big[s(q, a) \geq \tau \mid q\big] = \sum_a \pi_{\hat{\theta}_t}(a \mid q)\, \mathbf{1}_{\{s(q,a) \geq \tau\}},$$

$$\alpha^{(m)}(\hat{\theta}_t, q) := 1 - \big(1 - \alpha(\hat{\theta}_t, q)\big)^m, \qquad Z_{p_0}^{(m)}(\hat{\theta}_t) := \mathbb{E}_{q \sim p_0}\big[\alpha^{(m)}(\hat{\theta}_t, q)\big].$$

For any $m \geq 1$, the population MLE objective $\mathbb{E}_{(q,a) \sim D'^{(m)}_{p_0, \hat{\theta}_t}}\big[\log \pi_\theta(a \mid q)\big]$ at iteration $t$ achieves its maximum at $\theta = \theta_{t+1}^\star$ where

$$\pi_{\theta_{t+1}^\star}(\cdot \mid q) = p'_{\hat{\theta}_t}(\cdot \mid q)$$

for almost every $q$. Note that $p'_{\hat{\theta}_t}(\cdot \mid q)$ is independent of $m$ and coincides with the conditional distribution over answers induced by $D'_{p_0, \hat{\theta}_t}$; hence, by assumption, $p'_{\hat{\theta}_t}(\cdot \mid q) \in \Pi$.

Define $A_q := \{a : s(q,a) \geq \tau\}$. By construction, $\pi_{\theta^\star_{t+1}}(A_q \mid q) = 1$ and hence $\pi_{\theta^\star_{t+1}}(A_q^c \mid q) = 0$. Further define

$$\delta_t(q) := \Pr_{a \sim \pi_{\hat{\theta}_{t+1}}(\cdot|q)}[s(q,a) < \tau] = \pi_{\hat{\theta}_{t+1}}(A_q^c \mid q).$$

By the definition of total variation distance,

$$\delta_t(q) = \left|\pi_{\theta^\star_{t+1}}(A_q^c \mid q) - \pi_{\hat{\theta}_{t+1}}(A_q^c \mid q)\right| \leq \mathrm{TV}\Big(\pi_{\theta^\star_{t+1}}(\cdot \mid q),\, \pi_{\hat{\theta}_{t+1}}(\cdot \mid q)\Big).$$

For any dominating measure $\omega$, it follows that

$$\mathrm{TV}\Big(\pi_{\theta^\star_{t+1}}(\cdot \mid q),\, \pi_{\hat{\theta}_{t+1}}(\cdot \mid q)\Big) = \frac{1}{2}\int\left|\frac{\mathrm{d}\,\pi_{\theta^\star_{t+1}}(\cdot \mid q)}{\mathrm{d}\omega} - \frac{\mathrm{d}\,\pi_{\hat{\theta}_{t+1}}(\cdot \mid q)}{\mathrm{d}\omega}\right|\mathrm{d}\omega$$

$$\leq \left(\int\left(\sqrt{\frac{\mathrm{d}\,\pi_{\theta^\star_{t+1}}(\cdot \mid q)}{\mathrm{d}\omega}} - \sqrt{\frac{\mathrm{d}\,\pi_{\hat{\theta}_{t+1}}(\cdot \mid q)}{\mathrm{d}\omega}}\right)^2 \mathrm{d}\omega\right)^{1/2}$$

$$=: \sqrt{D_{\mathrm{H}}^2\Big(\pi_{\theta^\star_{t+1}}(\cdot \mid q),\, \pi_{\hat{\theta}_{t+1}}(\cdot \mid q)\Big)}.$$

Here $D_{\mathrm{H}}^2(\cdot,\cdot)$ denotes the Hellinger distance. Taking expectation over $q \sim p_{p_0,\hat{\theta}_t}^{\prime(m)}$ and using the bound above, we obtain

$$\mathbb{E}_{q \sim p_{p_0,\hat{\theta}_t}^{\prime(m)}}\big[\delta_t(q)\big] \leq \left(\mathbb{E}_{q \sim p_{p_0,\hat{\theta}_t}^{\prime(m)}}\Big[D_{\mathrm{H}}^2\big(\pi_{\theta^\star_{t+1}}(\cdot \mid q),\, \pi_{\hat{\theta}_{t+1}}(\cdot \mid q)\big)\Big]\right)^{1/2}.$$

Based on Lemma B.1, with probability at least $1 - \delta$,

$$\mathbb{E}_{q \sim p^{\prime(m)}_{p_0,\hat{\theta}_t}}\big[\delta_t(q)\big] \leq \sqrt{\frac{2\log(|\Pi|\,\delta^{-1})}{n_t^{(m)}}}.$$

Therefore, w.h.p.,

$$\bar{\delta}_t^{(p_0)} := \mathbb{E}_{q \sim p_0}\big[\delta_t(q)\big] = \mathbb{E}_{q \sim p_{p_0,\hat{\theta}_t}^{\prime(m)}}\left[\frac{Z_{p_0}^{(m)}(\hat{\theta}_t)}{\alpha^{(m)}(\hat{\theta}_t, q)}\,\delta_t(q)\right] \leq \frac{Z_{p_0}^{(m)}(\hat{\theta}_t)}{\alpha^{(m)}(\hat{\theta}_t)}\sqrt{\frac{2\log(|\Pi|\,\delta^{-1})}{n_t^{(m)}}},$$

where $\alpha^{(m)}(\hat{\theta}_t) := \operatorname{ess\,inf}_q \alpha^{(m)}(\hat{\theta}_t, q)$. For any fixed $q$,

$$\mathbb{E}_{a \sim \pi_{\hat{\theta}_{t+1}}(\cdot|q)}[s(q,a)] = \mathbb{E}\big[s(q,a)\,\mathbf{1}_{A_q} \mid q\big] + \mathbb{E}\big[s(q,a)\,\mathbf{1}_{A_q^c} \mid q\big]$$

$$\geq \tau\,\pi_{\hat{\theta}_{t+1}}(A_q \mid q)$$

$$= \tau\big(1 - \delta_t(q)\big).$$

Taking expectation over $q \sim p_0$, with probability at least $1 - \delta$, we have

$$V_{p_0}(\hat{\theta}_{t+1}) = \mathbb{E}_{q \sim p_0}\mathbb{E}_{a \sim \pi_{\hat{\theta}_{t+1}}(\cdot|q)}[s(q,a)] \geq \tau\big(1 - \bar{\delta}_t^{(p_0)}\big) \geq \tau\left(1 - \frac{Z_{p_0}^{(m)}(\hat{\theta}_t)}{\alpha^{(m)}(\hat{\theta}_t)}\sqrt{\frac{2\log(|\Pi|\,\delta^{-1})}{n_t^{(m)}}}\right).$$

Recall the notation $\alpha(\hat{\theta}_t, q) \in [0,1]$ and $\alpha(\hat{\theta}_t) := \operatorname{ess\,inf}_q \alpha(\hat{\theta}_t, q) > 0$. For $m \in \mathbb{N}$ define

$$f_m(x) := 1 - (1-x)^m, \qquad x \in [0,1].$$

Then

$$\alpha^{(m)}(\hat{\theta}_t, q) = f_m(\alpha(\hat{\theta}_t, q)), \quad \alpha^{(m)}(\hat{\theta}_t) = \operatorname{ess\,inf}_q f_m\big(\alpha(\hat{\theta}_t, q)\big) = f_m(\alpha(\hat{\theta}_t)), \quad Z_{p_0}^{(m)}(\hat{\theta}_t) = \mathbb{E}_{q \sim p_0}\big[f_m(\alpha(\hat{\theta}_t, q))\big].$$

Define, for $y \in [0, 1)$,

$$h_m(y) := \frac{1 - y^{m+1}}{1 - y^m}.$$

Using $f_{m+1}(x)/f_m(x) = h_m(1 - x)$, we have

$$\frac{Z_{p_0}^{(m+1)}(\hat{\theta}_t)}{\alpha^{(m+1)}(\hat{\theta}_t)} = \mathbb{E}\left[\frac{f_{m+1}(\alpha(\hat{\theta}_t, q))}{f_m(\alpha(\hat{\theta}_t, q))} \cdot \frac{f_m(\alpha(\hat{\theta}_t, q))}{f_m(\alpha(\hat{\theta}_t))} \cdot \frac{f_m(\alpha(\hat{\theta}_t))}{f_{m+1}(\alpha(\hat{\theta}_t))}\right] = \mathbb{E}_{q \sim p_0}\left[\frac{h_m(1 - \alpha(\hat{\theta}_t, q))}{h_m(1 - \alpha(\hat{\theta}_t))} \cdot \frac{f_m(\alpha(\hat{\theta}_t, q))}{f_m(\alpha(\hat{\theta}_t))}\right].$$

We claim $h_m$ is increasing on $[0, 1)$. Indeed,

$$\frac{\mathrm{d}}{\mathrm{d}y} \log h_m(y) = \frac{y^{m-1}(m - (m+1)y + y^{m+1})}{(1 - y^m)(1 - y^{m+1})} \geq 0,$$

since the $m - (m+1)y + y^{m+1}$ is decreasing in $y$ with value $m$ at $y = 0$ and $0$ at $y = 1$. Because $\alpha(\hat{\theta}_t, q) \geq \alpha(\hat{\theta}_t)$ a.s., we have $1 - \alpha(\hat{\theta}_t, q) \leq 1 - \alpha(\hat{\theta}_t)$ and hence $h_m(1 - \alpha(\hat{\theta}_t, q)) \leq h_m(1 - \alpha(\hat{\theta}_t))$. Therefore,

$$\frac{Z_{p_0}^{(m+1)}(\hat{\theta}_t)}{\alpha^{(m+1)}(\hat{\theta}_t)} \leq \frac{Z_{p_0}^{(m)}(\hat{\theta}_t)}{\alpha^{(m)}(\hat{\theta}_t)}.$$

Since $1 > \alpha(\hat{\theta}_t) > 0$, we have $f_m(\alpha(\hat{\theta}_t, q)) \uparrow 1$ for every $q$ and $f_m(\alpha(\hat{\theta}_t)) \uparrow 1$ as $m \to \infty$. By the monotone convergence theorem, $Z_{p_0}^{(m)}(\hat{\theta}_t) = \mathbb{E}[f_m(\alpha(\hat{\theta}_t, q))] \to 1$ and $\alpha^{(m)}(\hat{\theta}_t) = f_m(\alpha(\hat{\theta}_t)) \to 1$, hence

$$\lim_{m \to \infty} \frac{Z_{p_0}^{(m)}(\hat{\theta}_t)}{\alpha^{(m)}(\hat{\theta}_t)} = 1.$$

$\square$

**Lemma B.1** (Wong & Shen (1995); Geer (2000); Zhang (2006); Huang et al. (2025)). *Fix iteration $t$ and $m \geq 1$. Let $\mathcal{Q}$ be the question space and $\Delta(\mathcal{A})$ the set of probability measures on the answer space $\mathcal{A}$. Let $\Pi \subset (\mathcal{Q} \to \Delta(\mathcal{A}))$ be a finite model class and suppose the population optimizer $\pi_{\theta_{t+1}^\star}(\cdot \mid q) = p_{\hat{\theta}_t}'(\cdot \mid q)$ belongs to $\Pi$. Draw $n_t^{(m)}$ accepted samples i.i.d.*

$$(q_i, a_i) \sim D_{p_0, \hat{\theta}_t}'^{(m)}(q, a) = p_{p_0, \hat{\theta}_t}'^{(m)}(q) \, \pi_{\theta_{t+1}^\star}(a \mid q), \qquad i = 1, \ldots, n_t^{(m)},$$

*and define the empirical MLE*

$$\hat{\theta}_{t+1} \in \arg \max_{\theta : \pi_\theta \in \Pi} \sum_{i=1}^{n_t^{(m)}} \log \pi_\theta(a_i \mid q_i).$$

*Then for any $\delta \in (0, 1)$, with probability at least $1 - \delta$,*

$$\mathbb{E}_{q \sim p_{p_0, \hat{\theta}_t}'^{(m)}}\left[D_{\mathrm{H}}^2\big(\pi_{\hat{\theta}_{t+1}}(\cdot \mid q), \pi_{\theta_{t+1}^\star}(\cdot \mid q)\big)\right] \leq \frac{2 \log\big(|\Pi| \, \delta^{-1}\big)}{n_t^{(m)}}.$$

## B.2  PROOF OF COROLLARY 4.4

*Proof.* Fix iteration $t$, recall

$$Z_{p_0}(\hat{\theta}_t) = \mathbb{E}_{q \sim p_0}\big[\alpha(\hat{\theta}_t, q)\big], \qquad \alpha(\hat{\theta}_t, q) = \Pr_{a \sim \pi_{\hat{\theta}_t}(\cdot \mid q)}\big[s(q, a) \geq \tau \mid q\big].$$

When $s \in \{0, 1\}$ and $\tau \in (0, 1]$, we have $\mathbf{1}_{\{s(q,a) \geq \tau\}} = s(q, a)$, hence $\alpha(\hat{\theta}_t, q) = \mathbb{E}_{a \sim \pi_{\hat{\theta}_t}(\cdot \mid q)}\big[s(q, a)\big]$. Averaging over $q \sim p_0$ yields

$$Z_{p_0}(\hat{\theta}_t) = \mathbb{E}_{q \sim p_0} \mathbb{E}_{a \sim \pi_{\hat{\theta}_t}(\cdot \mid q)}\big[s(q, a)\big] = V_{p_0}(\hat{\theta}_t).$$

Moreover, since $s \in \{0, 1\}$, for any fixed $q$,

$$\mathbb{E}_{a \sim \pi_{\hat{\theta}_{t+1}}(\cdot|q)}[s(q, a)] = \Pr_{a \sim \pi_{\hat{\theta}_{t+1}}(\cdot|q)}[s(q, a) = 1] = 1 - \delta_t(q),$$

where $\delta_t(q) := \Pr_{a \sim \pi_{\hat{\theta}_{t+1}}(\cdot|q)}[s(q, a) < \tau]$. Therefore,

$$V_{p_0}(\hat{\theta}_{t+1}) = \mathbb{E}_{q \sim p_0} \mathbb{E}_{a \sim \pi_{\hat{\theta}_{t+1}}(\cdot|q)}[s(q, a)] = 1 - \bar{\delta}_t^{(p_0)}, \qquad \bar{\delta}_t^{(p_0)} := \mathbb{E}_{q \sim p_0}[\delta_t(q)].$$

At iteration $t$, we propose $n$ i.i.d. pairs $(q_i, a_i) \sim p_0(q)\, \pi_{\hat{\theta}_t}(a \mid q)$, and accept those with $s(q_i, a_i) \geq \tau$. Let $X_i := \mathbf{1}_{\{s(q_i, a_i) \geq \tau\}} \in \{0, 1\}$ and $n_t := \sum_{i=1}^n X_i$. Then, by the binary reward identity above,

$$\mathbb{E}[X_i] = \mathbb{E}_{q \sim p_0} \Pr_{a \sim \pi_{\hat{\theta}_t}(\cdot|q)}[s(q, a) \geq \tau \mid q] = Z_{p_0}(\hat{\theta}_t) = V_{p_0}(\hat{\theta}_t).$$

Hoeffding's inequality gives, for any $\epsilon > 0$,

$$\Pr\left[\frac{n_t}{n} \leq V_{p_0}(\hat{\theta}_t) - \epsilon\right] \leq \exp(-2n\epsilon^2).$$

Choosing $\epsilon = \sqrt{\log(\delta'^{-1})/(2n)}$ implies that with probability at least $1 - \delta'$,

$$\frac{n_t}{n} \geq V_{p_0}(\hat{\theta}_t) - \sqrt{\frac{\log(\delta'^{-1})}{2n}}.$$

Define

$$\mathcal{G}_t := \left\{q : \alpha(\hat{\theta}_t, q) \geq c\, V_{p_0}(\hat{\theta}_t)\right\}.$$

By Assumption 4.3, $\Pr_{q \sim p_0}\left[q \notin \mathcal{G}_t\right] \leq \gamma$. Since $\delta_t(q) \in [0, 1]$, we have

$$\bar{\delta}_t^{(p_0)} = \mathbb{E}_{q \sim p_0}[\delta_t(q)\mathbf{1}_{\mathcal{G}_t}] + \mathbb{E}_{q \sim p_0}[\delta_t(q)\mathbf{1}_{\mathcal{G}_t^c}] \leq \mathbb{E}_{q \sim p_0}[\delta_t(q)\mathbf{1}_{\mathcal{G}_t}] + \Pr_{q \sim p_0}\left[q \notin \mathcal{G}_t\right] \leq \mathbb{E}_{q \sim p_0}[\delta_t(q)\mathbf{1}_{\mathcal{G}_t}] + \gamma.$$

Furthermore,

$$\mathbb{E}_{q \sim p_0}[\delta_t(q)\mathbf{1}_{\mathcal{G}_t}] = \mathbb{E}_{q \sim p'_{p_0, \hat{\theta}_t}}\left[\frac{Z_{p_0}(\hat{\theta}_t)}{\alpha(\hat{\theta}_t, q)}\, \delta_t(q)\mathbf{1}_{\mathcal{G}_t}\right] \leq \frac{1}{c}\, \mathbb{E}_{q \sim p'_{p_0, \hat{\theta}_t}}[\delta_t(q)],$$

where we use the fact that On $\mathcal{G}_t$, we have $\alpha(\hat{\theta}_t, q) \geq cV_{p_0}(\hat{\theta}_t) = cZ_{p_0}(\hat{\theta}_t)$. Thus

$$\bar{\delta}_t^{(p_0)} \leq \gamma + \frac{1}{c}\, \mathbb{E}_{q \sim p'_{p_0, \hat{\theta}_t}}[\delta_t(q)].$$

By repeating the identical argument in the proof of Theorem 4.1 with $m = 1$, we obtain that with probability at least $1 - \delta$,

$$\mathbb{E}_{q \sim p'_{p_0, \hat{\theta}_t}}[\delta_t(q)] \leq \sqrt{\frac{2\log(|\Pi|\,\delta^{-1})}{n_t}}.$$

Then, we get (w.p. $\geq 1 - \delta$):

$$V_{p_0}(\hat{\theta}_{t+1}) = 1 - \bar{\delta}_t^{(p_0)} \geq 1 - \gamma - \frac{1}{c}\sqrt{\frac{2\log(|\Pi|\,\delta^{-1})}{n_t}}.$$

Moreover, w.p. $\geq 1 - \delta'$,

$$\frac{1}{\sqrt{n_t}} \leq \frac{1}{\sqrt{n\left(V_{p_0}(\hat{\theta}_t) - \sqrt{\log(\delta'^{-1})/(2n)}\right)}}.$$

Therefore, by a union bound over the two events, with probability at least $1 - \delta - \delta'$,

$$V_{p_0}(\hat{\theta}_{t+1}) \geq 1 - \gamma - \frac{1}{c}\sqrt{\frac{2\log(|\Pi|\delta^{-1})/n}{V_{p_0}(\hat{\theta}_t) - \sqrt{\log(\delta'^{-1})/(2n)}}}.$$

Define

$$F(x) := 1 - \gamma - \frac{c_\delta \nu}{c\sqrt{x - c_{\delta'}\nu}}, \qquad \nu := \sqrt{\frac{1}{n}}, \qquad c_\delta := \sqrt{2\log(|\Pi|\,\delta^{-1})}, \qquad c_{\delta'} := \sqrt{\frac{\log(\delta'^{-1})}{2}},$$

on its natural domain $x > c_{\delta'}\nu$. Then the preceding bound can be rewritten as

$$V_{p_0}(\hat{\theta}_{t+1}) \geq F\big(V_{p_0}(\hat{\theta}_t)\big).$$

Finally, since $F$ is monotone increasing on its domain, iterating the one-step inequality yields that, with probability at least $1 - t(\delta + \delta')$, for every integer $t \geq 0$,

$$V_{p_0}(\hat{\theta}_t) \geq F^{\circ t}\big(V_{p_0}(\hat{\theta}_0)\big),$$

where $F^{\circ t}$ denotes the $t$-fold composition of $F$. This completes the proof.

$\square$

### B.3 Proof of Proposition 4.5

*Proof.* This proposition is an immediate specialization of Corollary B.3 by taking $a = 1$. $\square$

**Lemma B.2.** *Let*

$$F(x) = 1 - \frac{\sigma}{\sqrt{x}}, \qquad x \in (0,1),$$

*with parameter $\sigma > 0$. Assume $0 < \sigma < \sqrt{4/27}$. Let $x_- < x_+$ denote the two solutions in $(0,1)$ of $x(1-x)^2 = \sigma^2$. Then,*

$$x_+ - x_- \geq 1 - \frac{3\sqrt{3}}{2}\sigma,$$

*and for every $x \in (x_-, x_+)$ and every integer $t \geq 0$, all iterates $F^{\circ t}(x)$ stay in $(x_-, x_+)$ and satisfy $F^{\circ(t+1)}(x) > F^{\circ t}(x)$.*

*Proof.* Consider

$$h(x) := x(1-x)^2, \qquad x \in [0,1].$$

Then $h'(x) = (1-x)^2 - 2x(1-x) = (1-x)(1-3x)$, so $h$ is strictly increasing on $(0, 1/3)$ and strictly decreasing on $(1/3, 1)$, with a unique interior maximizer at $x = 1/3$. Moreover, $h(1/3) = 4/27$. So whenever $\sigma^2 < 4/27$, the equation $h(x) = \sigma^2$ has exactly two distinct solutions in $(0,1)$; we denote them by $x_- < x_+$.

Moreover, expanding $h(x) = \sigma^2$ gives the cubic $x^3 - 2x^2 + x - \sigma^2 = 0$. Set $x = z + 2/3$, we have

$$x^3 - 2x^2 + x - \sigma^2 = z^3 - \frac{1}{3}z + \left(\frac{2}{27} - \sigma^2\right) = 0.$$

Trigonometric solution of the roots of this depressed cubic is

$$z_\ell = \frac{2}{3}\cos\left[\frac{1}{3}\arccos\left(-1 + \frac{27}{2}\sigma^2\right) - \frac{2\pi\ell}{3}\right], \qquad \ell = 0, 1, 2.$$

so

$$x_\ell = \frac{2}{3} + \frac{2}{3}\cos\left[\frac{1}{3}\arccos\left(-1 + \frac{27}{2}\sigma^2\right) - \frac{2\pi\ell}{3}\right], \qquad \ell = 0, 1, 2.$$

Define

$$u := \frac{1}{3}\arccos\left(-1 + \frac{27}{2}\sigma^2\right) \in \left(0, \frac{\pi}{3}\right),$$

then

$$x_+ = \frac{2}{3} + \frac{2}{3} \cos\left(u - \frac{2\pi}{3}\right), \qquad x_- = \frac{2}{3} + \frac{2}{3} \cos\left(u - \frac{4\pi}{3}\right).$$

Hence

$$x_+ - x_- = \frac{2}{\sqrt{3}} \sin u = \frac{2}{\sqrt{3}} \sin\left(\frac{1}{3} \arccos\left(-1 + \frac{27}{2}\sigma^2\right)\right).$$

We now prove the desired bound $x_+ - x_- \geq 1 - \frac{3\sqrt{3}}{2}\sigma$. Let $\cos\theta := (3\sqrt{3}\sigma)/2 \in (0,1)$ with $\theta \in [0, \pi/2]$, then

$$-1 + \frac{27}{2}\sigma^2 = 2\cos^2\theta - 1 = \cos(2\theta),$$

Hence $x_+ - x_- = \frac{2}{\sqrt{3}} \sin\left(\frac{2\theta}{3}\right)$. and $1 - \frac{3\sqrt{3}}{2}\sigma = 1 - \cos\theta$.

Thus it suffices to show that for all $\theta \in [0, \pi/2]$,

$$\frac{2}{\sqrt{3}} \sin\left(\frac{2\theta}{3}\right) \geq 1 - \cos\theta.$$

Define

$$g(\theta) := \frac{2}{\sqrt{3}} \sin\left(\frac{2\theta}{3}\right) - \left(1 - \cos\theta\right), \qquad \theta \in [0, \frac{\pi}{2}].$$

We have $g(0) = g(\pi/2) = 0$. Moreover,

$$g''(\theta) = -\frac{8}{9\sqrt{3}} \sin\left(\frac{2\theta}{3}\right) - \cos\theta \leq 0$$

for $\theta \in [0, \pi/2]$ since we have $\sin(2\theta/3) \geq 0$ and $\cos\theta \geq 0$, thus $g$ is concave on $[0, \pi/2]$ and vanishes at both endpoints. So $g(\theta) \geq 0$, which is equivalent to

$$x_+ - x_- \geq 1 - \frac{3\sqrt{3}}{2}\sigma.$$

Next, we characterize where $F(x) > x$. For $x \in (0,1)$, this is equivalent to $\sigma^2 < x(1-x)^2$. Recall that $x_\pm$ are the two solutions to $x(1-x)^2 = \sigma^2$ in $(0,1)$, and that $h(x) = x(1-x)^2$ is strictly increasing on $(0, 1/3)$ and strictly decreasing on $(1/3, 1)$. Hence $x(1-x)^2 > \sigma^2$ is equivalent to $x \in (x_-, x_+)$, so we have $F(x) > x$ is equivalent to $x \in (x_-, x_+)$. Let $x \in (x_-, x_+)$ and define $x_t := F^{\circ t}(x)$ for $t \geq 0$.

We prove by induction that

$$x_- < x_t < x_+, \qquad x_{t+1} > x_t, \qquad \forall t \geq 0.$$

For $t = 0$, we have $x_0 = x \in (x_-, x_+)$ by assumption, and $x_1 = F(x_0) > x_0$. Using monotonicity of $F$ and $F(x_\pm) = x_\pm$,

$$x_- = F(x_-) < F(x_0) = x_1 < F(x_+) = x_+,$$

so $x_1 \in (x_-, x_+)$. Assume now that $x_t \in (x_-, x_+)$ and $x_t > x_{t-1}$ for some $t \geq 1$. Then, since $x_t \in (x_-, x_+)$, $x_{t+1} = F(x_t) > x_t$. Furthermore, monotonicity of $F$ and the fixed-point property at $x_\pm$ give

$$x_- = F(x_-) < F(x_t) = x_{t+1} < F(x_+) = x_+,$$

so $x_{t+1} \in (x_-, x_+)$. This completes the induction and shows that for every integer $t \geq 0$, $F^{\circ(t+1)}(x) > F^{\circ t}(x)$. $\qquad\square$

**Corollary B.3.** *Let*

$$F(x; a, \nu) := 1 - \gamma - \frac{c_\delta \nu}{c\sqrt{ax - c_{\delta'}\nu}}, \qquad x > \frac{c_{\delta'}\nu}{a}.$$

*Assume $(a, \nu)$ satisfies the validity conditions in Definition A.1, so that the interval*

$$\mathcal{I}(a, \nu) := \left(x_-(a, \nu), x_+(a, \nu)\right) \subset \left(\frac{c_{\delta'}\nu}{a}, 1 - \gamma\right)$$

*in Definition A.1 is well-defined. Then, for any $x \in \mathcal{I}(a, \nu)$ and any integer $t \geq 0$, $F^{\circ t}(x; a, \nu) \in \mathcal{I}(a, \nu)$ and $F^{\circ(t+1)}(x; a, \nu) > F^{\circ t}(x; a, \nu)$. Moreover, the interval length satisfies*

$$|\mathcal{I}(a, \nu)| \geq \left(1 - \gamma - \frac{c_{\delta'}\nu}{a}\right) - \frac{3\sqrt{3}}{2} \cdot \frac{c_\delta \nu}{c\sqrt{a(1 - \gamma) - c_{\delta'}\nu}}.$$

*Furthermore, the family $\{\mathcal{I}(a, \nu)\}$ is monotone in the sense of inclusion: (i) for fixed $\nu$, if $a_2 > a_1 > 0$, then $\mathcal{I}(a_1, \nu) \subset \mathcal{I}(a_2, \nu)$; (ii) for fixed $a$, if $\nu_2 > \nu_1$, then $\mathcal{I}(a, \nu_2) \subset \mathcal{I}(a, \nu_1)$.*

*Proof.* We follow the notations in Definition A.1. Define the affine change of variables

$$x = \frac{c_{\delta'}\nu}{a} + \left(1 - \gamma - \frac{c_{\delta'}\nu}{a}\right) y, \qquad y \in (0, 1),$$

A direct substitution shows that for every $y \in (0, 1)$,

$$F\left(\frac{c_{\delta'}\nu}{a} + \left(1 - \gamma - \frac{c_{\delta'}\nu}{a}\right) y; a, \nu\right) = 1 - \gamma - \frac{c_\delta \nu}{c\sqrt{a\left(\frac{c_{\delta'}\nu}{a} + (1 - \gamma - \frac{c_{\delta'}\nu}{a})y\right) - c_{\delta'}\nu}}$$

$$= 1 - \gamma - \frac{c_\delta \nu}{c\sqrt{(a(1 - \gamma) - c_{\delta'}\nu)y}}$$

$$= \frac{c_{\delta'}\nu}{a} + \left(1 - \gamma - \frac{c_{\delta'}\nu}{a}\right)\left(1 - \frac{a\, c_\delta \nu}{c\left(a(1 - \gamma) - c_{\delta'}\nu\right)^{3/2}} \cdot \frac{1}{\sqrt{y}}\right).$$

Therefore, if we denote

$$g_{a,\nu}(y) := 1 - \frac{a\, c_\delta \nu}{c\left(a(1 - \gamma) - c_{\delta'}\nu\right)^{3/2}} \cdot \frac{1}{\sqrt{y}}, \qquad y \in (0, 1),$$

then we have

$$F\left(\frac{c_{\delta'}\nu}{a} + \left(1 - \gamma - \frac{c_{\delta'}\nu}{a}\right) y; a, \nu\right) = \frac{c_{\delta'}\nu}{a} + \left(1 - \gamma - \frac{c_{\delta'}\nu}{a}\right) g_{a,\nu}(y).$$

The map $g_{a,\nu}$ is exactly of the form in Lemma B.2 with parameter

$$\frac{a\, c_\delta \nu}{c\left(a(1 - \gamma) - c_{\delta'}\nu\right)^{3/2}} \in \left(0, \sqrt{\frac{4}{27}}\right).$$

Then, applying Lemma B.2 to $g_{a,\nu}$ yields: for every $y \in \left(y_-(a, \nu), y_+(a, \nu)\right)$ and every $t \geq 0$,

$$g_{a,\nu}^{\circ t}(y) \in \left(y_-(a, \nu), y_+(a, \nu)\right) \quad \text{and} \quad g_{a,\nu}^{\circ(t+1)}(y) > g_{a,\nu}^{\circ t}(y),$$

and moreover

$$y_+(a, \nu) - y_-(a, \nu) \geq 1 - \frac{3\sqrt{3}}{2} \cdot \frac{a\, c_\delta \nu}{c\left(a(1 - \gamma) - c_{\delta'}\nu\right)^{3/2}}.$$

Now take any $x \in \mathcal{I}(a, \nu)$ and write it as $x = \frac{c_{\delta'}\nu}{a} + \left(1 - \gamma - \frac{c_{\delta'}\nu}{a}\right) y$ with $y \in \left(y_-(a, \nu), y_+(a, \nu)\right)$. Iterating the conjugacy identity gives

$$F^{\circ t}(x; a, \nu) = \frac{c_{\delta'}\nu}{a} + \left(1 - \gamma - \frac{c_{\delta'}\nu}{a}\right) g_{a,\nu}^{\circ t}(y), \qquad \forall\, t \geq 0,$$

so $g_{a,\nu}^{\circ t}(y) \in \left(y_-(a, \nu), y_+(a, \nu)\right)$ implies $F^{\circ t}(x; a, \nu) \in \mathcal{I}(a, \nu)$, and $g_{a,\nu}^{\circ(t+1)}(y) > g_{a,\nu}^{\circ t}(y)$ implies $F^{\circ(t+1)}(x; a, \nu) > F^{\circ t}(x; a, \nu)$.

For the interval length, using the lemma's bound,

$$|\mathcal{I}(a, \nu)| = x_+(a, \nu) - x_-(a, \nu) = \left(1 - \gamma - \frac{c_{\delta'}\nu}{a}\right)\left(y_+(a, \nu) - y_-(a, \nu)\right)$$

$$\geq \left(1 - \gamma - \frac{c_{\delta'}\nu}{a}\right)\left(1 - \frac{3\sqrt{3}}{2} \cdot \frac{a\, c_\delta \nu}{c\left(a(1 - \gamma) - c_{\delta'}\nu\right)^{3/2}}\right)$$

$$= \left(1 - \gamma - \frac{c_{\delta'}\nu}{a}\right) - \frac{3\sqrt{3}}{2} \cdot \frac{c_\delta \nu}{c\sqrt{a(1 - \gamma) - c_{\delta'}\nu}}.$$

For any parameters $(a, \nu)$ in the validity range of Definition A.1, recall that the endpoints $x_-(a, \nu) < x_+(a, \nu)$ are the two fixed points of $F(\cdot; a, \nu)$, i.e.,

$$F\big(x_\pm(a, \nu); a, \nu\big) = x_\pm(a, \nu).$$

Define the fixed-point equation

$$\Phi(x, a, \nu) := x - F(x; a, \nu) = 0.$$

Whenever $\partial_x \Phi\big(x_\pm(a, \nu), a, \nu\big) \neq 0$, the implicit function theorem gives

$$\frac{\partial}{\partial a} x_\pm(a, \nu) = -\frac{\partial_a \Phi\big(x_\pm(a, \nu), a, \nu\big)}{\partial_x \Phi\big(x_\pm(a, \nu), a, \nu\big)}, \qquad \frac{\partial}{\partial \nu} x_\pm(a, \nu) = -\frac{\partial_\nu \Phi\big(x_\pm(a, \nu), a, \nu\big)}{\partial_x \Phi\big(x_\pm(a, \nu), a, \nu\big)}.$$

Fix $\nu$ and view $\Phi$ as a function of $(a, x)$. First, for any $x > c_{\delta'}\nu/a$,

$$\partial_a \Phi(x, a, \nu) = -\partial_a F(x; a, \nu) = -\frac{c_\delta \nu}{c} \cdot \frac{x}{2\,(ax - c_{\delta'}\nu)^{3/2}} \;<\; 0.$$

Next, $\partial_x \Phi(x, a, \nu) = 1 - \partial_x F(x; a, \nu)$. To determine its sign at the fixed points, consider the affine map

$$x = \frac{c_{\delta'}\nu}{a} + \left(1 - \gamma - \frac{c_{\delta'}\nu}{a}\right) y,$$

which maps $F(x; a, \nu)$ to $g_{a,\nu}(y) = 1 - \sigma(a, \nu)/\sqrt{y}$. At a fixed point $y = g_{a,\nu}(y)$ we have $\sigma(a, \nu) = (1 - y)\sqrt{y}$, hence

$$g'_{a,\nu}(y) = \frac{\sigma(a, \nu)}{2y^{3/2}} = \frac{1 - y}{2y}.$$

Since $y_-(a, \nu) \in (0, 1/3)$ and $y_+(a, \nu) \in (1/3, 1)$, $g'_{a,\nu}\big(y_-(a, \nu)\big) > 1$ and $g'_{a,\nu}\big(y_+(a, \nu)\big) < 1$. Therefore $\partial_x \Phi\big(x_-(a, \nu), a, \nu\big) < 0$ and $\partial_x \Phi\big(x_+(a, \nu), a, \nu\big) > 0$. Combining with $\partial_a \Phi(x_\pm(a, \nu), a, \nu) < 0$ yields

$$\frac{\partial}{\partial a} x_-(a, \nu) < 0, \qquad \frac{\partial}{\partial a} x_+(a, \nu) > 0.$$

Consequently, for any $a_2 > a_1$ (with the same fixed $\nu$),

$$x_-(a_2, \nu) < x_-(a_1, \nu), \qquad x_+(a_2, \nu) > x_+(a_1, \nu),$$

i.e., $\mathcal{I}(a_1, \nu) \subset \mathcal{I}(a_2, \nu)$.

Finally, fix $a$ and view $\Phi$ as a function of $(\nu, x)$. For any $x > c_{\delta'}\nu/a$, $\partial_\nu \Phi(x, a, \nu) = -\partial_\nu F(x; a, \nu)$. A direct differentiation shows $\partial_\nu F(x; a, \nu) < 0$, hence $\partial_\nu \Phi(x, a, \nu) > 0$. Moreover, as established above, $\partial_x \Phi\big(x_-(a, \nu), a, \nu\big) < 0$ and $\partial_x \Phi\big(x_+(a, \nu), a, \nu\big) > 0$. Therefore,

$$\frac{\partial}{\partial \nu} x_-(a, \nu) > 0, \qquad \frac{\partial}{\partial \nu} x_+(a, \nu) < 0,$$

Consequently, for any $\nu_2 > \nu_1$ (with the same fixed $a$).

$$x_-(a, \nu_2) > x_-(a, \nu_1), \qquad x_+(a, \nu_2) < x_+(a, \nu_1),$$

which implies $\mathcal{I}(a, \nu_2) \subset \mathcal{I}(a, \nu_1)$. $\qquad\square$

# C. Proofs for Section 5

## C.1 PROOF OF THEOREM 5.2

*Proof.* We now prove the first part of the theorem. By Definition A.2, the conditions $\mathcal{M}_i\big(\beta', \beta, \nu, V_{p_0}(\hat{\theta}_0)\big) < 0$ for all $i \in [4]$ are equivalent to requiring that both $V_{p_0}(\hat{\theta}_0)$ and $1 - \gamma - c_\delta \nu/(c\sqrt{a_0 V_{p_0}(\hat{\theta}_0) - c_{\delta'}\nu})$ lie in the open interval

$(x_-(2^{-\beta}, \nu),\ x_+(2^{-\beta}, \nu))$. By Corollary B.3, since $2^{-\beta} < 1$, we have $V_{p_0}(\hat{\theta}_0) \in \mathcal{I}(2^{-\beta}, \nu) \subset \mathcal{I}(1, \nu)$. Therefore, by Corollary 4.4 and Proposition 4.5, it follows that, with high probability,

$$V_{p_0}(\hat{\theta}_L^{\mathrm{B}}) \ \geq \ F^{\circ L}\Big(V_{p_0}(\hat{\theta}_0)\Big),$$

and the sequence $\{F^{\circ t}(V_{p_0}(\hat{\theta}_0))\}_{t \geq 0}$ is monotonically increasing in $t$, where

$$F(x) \ = \ 1 - \gamma - \frac{c_\delta \nu}{c\sqrt{x - c_{\delta'} \nu}}.$$

Next, recall that $\hat{\theta}_0^{\mathrm{E2H}} = \hat{\theta}_0$ and for easy-to-hard, during iteration $t$ we use distribution $p_{t+1}$. By Assumption 5.1, for every $i \in [L-1]$ and every $\theta$,

$$\frac{V_{p_i}(\theta)}{V_{p_{i+1}}(\theta)} \ \geq \ \frac{i^{-\beta'}}{(i+1)^{-\beta'}}.$$

Equivalently, $V_{p_{i+1}}(\theta) \leq (i/(i+1))^{\beta'} V_{p_i}(\theta)$. Iterating from 1 to $i-1$ yields, $V_{p_i}(\theta) \leq i^{-\beta'} V_{p_1}(\theta)$. Applying this with $\theta = \hat{\theta}_0$ and averaging over $i$ gives

$$V_{p_0}(\hat{\theta}_0) \ = \ \frac{1}{L} \sum_{i=1}^{L} V_{p_i}(\hat{\theta}_0) \ \leq \ \frac{1}{L}\Big(\sum_{i=1}^{L} i^{-\beta'}\Big) V_{p_1}(\hat{\theta}_0),$$

hence

$$V_{p_1}(\hat{\theta}_0) \ \geq \ a_0 V_{p_0}(\hat{\theta}_0), \qquad a_0 \ := \ \frac{L}{\sum_{i=1}^{L} i^{-\beta'}}.$$

Note that since $i^{-\beta'} < 1$ for all $i \geq 2$, we have $\sum_{i=1}^{L} i^{-\beta'} < L$ (for $L \geq 2$), and therefore $a_0 > 1$.

Then, at iteration $t$, easy-to-hard trains on $p_{t+1}$. By Corollary 4.4 (substitute $p_0$ with $p_{t+1}$), with high probability,

$$V_{p_{t+1}}(\hat{\theta}_{t+1}^{\mathrm{E2H}}) \ \geq \ F\Big(V_{p_{t+1}}(\hat{\theta}_t^{\mathrm{E2H}})\Big), \qquad t = 0, 1, \ldots, L-1.$$

Again by Assumption 5.1, for every $t \in [L-1]$ and every $\theta \in \Theta$,

$$\frac{V_{p_t}(\theta)}{V_{p_{t+1}}(\theta)} \ \leq \ \frac{t^{-\beta}}{(t+1)^{-\beta}},$$

equivalently,

$$V_{p_{t+1}}(\theta) \ \geq \ a_t V_{p_t}(\theta), \qquad a_t \ := \ \frac{(t+1)^{-\beta}}{t^{-\beta}} \ < \ 1.$$

Define

$$H_t(x) \ := \ F(a_t x) \ = \ 1 - \gamma - \frac{c_\delta \nu}{c\sqrt{a_t x - c_{\delta'} \nu}}, \qquad t = 0, 1, \ldots, L-1,$$

with $\{a_t\}$ defined above, We now chain the previous steps. First, since $V_{p_1}(\hat{\theta}_0) \geq a_0 V_{p_0}(\hat{\theta}_0)$, we have

$$V_{p_1}(\hat{\theta}_1^{\mathrm{E2H}}) \ \geq \ F\Big(V_{p_1}(\hat{\theta}_0)\Big) \ \geq \ F\Big(a_0 V_{p_0}(\hat{\theta}_0)\Big) \ = \ H_0\Big(V_{p_0}(\hat{\theta}_0)\Big).$$

Next, for $t = 1$,

$$V_{p_2}(\hat{\theta}_2^{\mathrm{E2H}}) \ \geq \ F\Big(V_{p_2}(\hat{\theta}_1^{\mathrm{E2H}})\Big) \ \geq \ F\Big(a_1 V_{p_1}(\hat{\theta}_1^{\mathrm{E2H}})\Big) \ = \ H_1\Big(V_{p_1}(\hat{\theta}_1^{\mathrm{E2H}})\Big).$$

Continuing this argument inductively for $t = 2, \ldots, L-1$ yields the recursion

$$V_{p_{t+1}}(\hat{\theta}_{t+1}^{\mathrm{E2H}}) \ \geq \ H_t\Big(V_{p_t}(\hat{\theta}_t^{\mathrm{E2H}})\Big), \qquad t = 0, 1, \ldots, L-1,$$

and therefore, after $L$ steps,

$$V_{p_L}(\hat{\theta}_L^{\mathrm{E2H}}) \ \geq \ (H_{L-1} \circ H_{L-2} \circ \cdots \circ H_0)\Big(V_{p_0}(\hat{\theta}_0)\Big).$$

We finally lower bound $V_{p_0}(\hat{\theta}_L^{\text{E2H}})$ in terms of $V_{p_L}(\hat{\theta}_L^{\text{E2H}})$. Using Assumption 5.1 and iterating as in for $a_0$, for every $i \in \{1, \ldots, L\}$ and every $\theta \in \Theta$ we have

$$V_{p_i}(\theta) \geq \left( \frac{i^{-\beta'}}{L^{-\beta'}} \right) V_{p_L}(\theta) = \left( \frac{L}{i} \right)^{\beta'} V_{p_L}(\theta).$$

Averaging over $i$ gives

$$V_{p_0}(\theta) = \frac{1}{L} \sum_{i=1}^{L} V_{p_i}(\theta) \geq \frac{1}{L} \sum_{i=1}^{L} \left( \frac{L}{i} \right)^{\beta'} V_{p_L}(\theta) = \frac{\sum_{i=1}^{L} i^{-\beta'}}{L^{1-\beta'}} V_{p_L}(\theta).$$

Applying this with $\theta = \hat{\theta}_L^{\text{E2H}}$ yields

$$V_{p_0}\big(\hat{\theta}_L^{\text{E2H}}\big) \geq a_L V_{p_L}\big(\hat{\theta}_L^{\text{E2H}}\big), \qquad a_L := \frac{\sum_{i=1}^{L} i^{-\beta'}}{L^{1-\beta'}},$$

and clearly $a_L > 1$ since $\sum_{i=1}^{L} i^{-\beta'} > L \cdot L^{-\beta'} = L^{1-\beta'}$. Defining $G(x) := a_L x$, we obtain

$$V_{p_0}\big(\hat{\theta}_L^{\text{E2H}}\big) \geq (G \circ H_{L-1} \circ H_{L-2} \circ \cdots \circ H_0)\Big(V_{p_0}(\hat{\theta}_0)\Big),$$

as claimed. Moreover, note that for $t \in [L-1]$, $a_t$ is strictly increasing in $t$. Therefore, by Corollary B.3, the associated intervals $\mathcal{I}(a_t, \nu)$ expand as $t$ increases. Since $V_{p_0}(\hat{\theta}_0)$ and $H_t(V_{p_0}(\hat{\theta}_0)) = 1 - \gamma - c_\delta \nu/(c\sqrt{a_0 V_{p_0}(\hat{\theta}_0) - c_{\delta'}\nu})$ lies in the smallest admissible interval $(x_-(2^{-\beta}, \nu), x_+(2^{-\beta}, \nu))$, the chained lower bounds stay within the corresponding invariant intervals and satisfy $\{(H_t \circ H_{t-1} \circ \cdots \circ H_0)(V_{p_0}(\hat{\theta}_0))\}_{t \geq 0}$ is monotonically increasing in $t$.

We now prove the second part of the theorem. Although the statement of Theorem 5.2 concerns $\nu > 0$, the maps involved are continuous in $\nu$, and it is convenient to first analyze the case $\nu = 0$. To distinguish the dependence on $\nu$, we write the baseline map as $F_\nu(\cdot)$ and the easy-to-hard maps as $H_{t,\nu}(\cdot)$.

When $\nu = 0$, the maps simplify to constants:

$$F_0(x) = 1 - \gamma, \qquad H_{t,0}(x) = 1 - \gamma, \qquad t = 0, 1, \ldots, L-1, \qquad G(x) = a_L x.$$

Define the comparison gap

$$\Delta(\nu, x) := \big(G \circ H_{L-1,\nu} \circ H_{L-2,\nu} \circ \cdots \circ H_{0,\nu}\big)(x) - F_\nu^{\circ L}(x).$$

Then we have

$$\Delta(0, x) = G\big(H_{L-1,0} \circ \cdots \circ H_{0,0}(x)\big) - F_0^{\circ L}(x) = a_L(1 - \gamma) - (1 - \gamma) = (a_L - 1)(1 - \gamma).$$

Since $a_L > 1$ and $1 - \gamma > 0$, it follows that $\Delta\big(0, V_{p_0}(\hat{\theta}_0)\big) = (a_L - 1)(1 - \gamma) > 0$. Therefore, our next step is to lower bound $\Delta\big(\nu, V_{p_0}(\hat{\theta}_0)\big) - \Delta\big(0, V_{p_0}(\hat{\theta}_0)\big)$. For convenience, denote $x_0 := V_{p_0}(\hat{\theta}_0)$.[4] Observe that the above difference involves two scalar sequences induced by the iterated maps. For baseline, define $\{x_t^{\text{B}}(\nu)\}_{t=0}^{L}$ by

$$x_0^{\text{B}}(\nu) = x_0, \qquad x_{t+1}^{\text{B}}(\nu) = F_\nu\big(x_t^{\text{B}}(\nu)\big), \qquad t = 0, 1, \ldots, L-1.$$

By construction, $F_\nu^{\circ L}(x_0) = x_L^{\text{B}}(\nu)$. Similarly, for easy-to-hard define $\{x_t^{\text{E2H}}(\nu)\}_{t=0}^{L}$ by

$$x_0^{\text{E2H}}(\nu) = x_0, \qquad x_{t+1}^{\text{E2H}}(\nu) = H_{t,\nu}\big(x_t^{\text{E2H}}(\nu)\big), \qquad t = 0, 1, \ldots, L-1.$$

By construction, $\big(G \circ H_{L-1,\nu} \circ H_{L-2,\nu} \circ \cdots \circ H_{0,\nu}\big)(x_0) = a_L x_L^{\text{E2H}}(\nu)$.

---

[4]We will use this shorthand throughout the remainder of this proof (and in subsequent proofs) whenever the dependence on the initialization is through $V_{p_0}(\hat{\theta}_0)$.

Combining the above identities, we obtain

$$\Delta(\nu, x_0) - \Delta(0, x_0) = \left(a_L x_L^{\text{E2H}}(\nu) - x_L^{\text{B}}(\nu)\right) - \left(a_L x_L^{\text{E2H}}(0) - x_L^{\text{B}}(0)\right)$$
$$= \left(x_L^{\text{B}}(0) - x_L^{\text{B}}(\nu)\right) - a_L\left(x_L^{\text{E2H}}(0) - x_L^{\text{E2H}}(\nu)\right).$$

Therefore, if we define the deviation sequences

$$e_t^{\text{B}}(\nu) := x_t^{\text{B}}(0) - x_t^{\text{B}}(\nu), \qquad e_t^{\text{E2H}}(\nu) := x_t^{\text{E2H}}(0) - x_t^{\text{E2H}}(\nu),$$

then the quantity of interest can be written succinctly as $\Delta(\nu, x_0) - \Delta(0, x_0) = e_L^{\text{B}}(\nu) - a_L\, e_L^{\text{E2H}}(\nu)$. Hence, it suffices to derive a lower bound on $e_L^{\text{B}}(\nu)$ and an upper bound on $e_L^{\text{E2H}}(\nu)$.

For the lower bound on $e_L^{\text{B}}(\nu)$, ignoring the trivial case $t = 0$, we start from

$$e_1^{\text{B}}(\nu) = F_0(x_0) - F_\nu(x_0) = \frac{c_\delta \nu}{c\sqrt{x_0 - c_{\delta'}\nu}}.$$

More generally, for every $t \geq 1$ we can write

$$e_t^{\text{B}}(\nu) = F_0\left(x_{t-1}^{\text{B}}(0)\right) - F_\nu\left(x_{t-1}^{\text{B}}(\nu)\right)$$
$$= \left(F_0\left(x_{t-1}^{\text{B}}(0)\right) - F_\nu\left(x_{t-1}^{\text{B}}(0)\right)\right) + \left(F_\nu\left(x_{t-1}^{\text{B}}(0)\right) - F_\nu\left(x_{t-1}^{\text{B}}(\nu)\right)\right).$$

The first term is nonnegative because $F_\nu(x) \leq F_0(x) = 1 - \gamma$ for all admissible $x$ when $\nu > 0$. The second term is also nonnegative because $F_\nu$ is increasing and $x_{t-1}^{\text{B}}(0) \geq x_{t-1}^{\text{B}}(\nu)$. Hence,

$$e_t^{\text{B}}(\nu) \geq 0, \qquad \forall t \geq 1.$$

Next we derive a quantitative lower bound. The first difference is explicit:

$$F_0\left(x_{t-1}^{\text{B}}(0)\right) - F_\nu\left(x_{t-1}^{\text{B}}(0)\right) = \frac{c_\delta \nu}{c\sqrt{1 - \gamma - c_{\delta'}\nu}}.$$

For the second difference, note that

$$F_\nu'(x) = \frac{c_\delta \nu}{2c\,(x - c_{\delta'}\nu)^{3/2}},$$

which is monotonically decreasing in $x$ over its domain. By the mean value theorem, there exists $\xi_{t-1} \in \left[x_{t-1}^{\text{B}}(\nu),\, 1 - \gamma\right]$ such that $F_\nu\left(x_{t-1}^{\text{B}}(0)\right) - F_\nu\left(x_{t-1}^{\text{B}}(\nu)\right) = F_\nu'(\xi_{t-1})\,e_{t-1}^{\text{B}}(\nu)$. Since $F_\nu'$ is decreasing and $\xi_{t-1} \leq 1 - \gamma$, we have $F_\nu'(\xi_{t-1}) \geq F_\nu'(1 - \gamma)$, and thus

$$F_\nu\left(x_{t-1}^{\text{B}}(0)\right) - F_\nu\left(x_{t-1}^{\text{B}}(\nu)\right) \geq \frac{c_\delta \nu}{2c\,(1 - \gamma - c_{\delta'}\nu)^{3/2}}\, e_{t-1}^{\text{B}}(\nu).$$

Combining the two pieces yields the recursion: for every $t = 2, 3, \ldots, L$,

$$e_t^{\text{B}}(\nu) \geq \frac{c_\delta \nu}{c\sqrt{1 - \gamma - c_{\delta'}\nu}} + \frac{c_\delta \nu}{2c\,(1 - \gamma - c_{\delta'}\nu)^{3/2}}\, e_{t-1}^{\text{B}}(\nu).$$

Iterating from $t = 2$ up to $t = L$ gives

$$e_L^{\text{B}}(\nu) \geq \frac{c_\delta \nu}{c\sqrt{1 - \gamma - c_{\delta'}\nu}} \cdot \frac{1 - \left(\frac{c_\delta \nu}{2c\,(1 - \gamma - c_{\delta'}\nu)^{3/2}}\right)^{L-1}}{1 - \frac{c_\delta \nu}{2c\,(1 - \gamma - c_{\delta'}\nu)^{3/2}}} + \left(\frac{c_\delta \nu}{2c\,(1 - \gamma - c_{\delta'}\nu)^{3/2}}\right)^{L-1} \frac{c_\delta \nu}{c\sqrt{x_0 - c_{\delta'}\nu}}.$$

For the upper bound on $e_L^{\text{E2H}}(\nu)$, ignoring the trivial case $t = 0$, we start from

$$e_1^{\text{E2H}}(\nu) = H_{0,0}(x_0) - H_{0,\nu}(x_0) = \frac{c_\delta \nu}{c\sqrt{a_0 x_0 - c_{\delta'}\nu}}.$$

More generally, for every $t \geq 1$ we can write

$$
\begin{aligned}
e_t^{\text{E2H}}(\nu) &= H_{t-1,0}\big(x_{t-1}^{\text{E2H}}(0)\big) - H_{t-1,\nu}\big(x_{t-1}^{\text{E2H}}(\nu)\big) \\
&= \Big(H_{t-1,0}\big(x_{t-1}^{\text{E2H}}(0)\big) - H_{t-1,\nu}\big(x_{t-1}^{\text{E2H}}(0)\big)\Big) + \Big(H_{t-1,\nu}\big(x_{t-1}^{\text{E2H}}(0)\big) - H_{t-1,\nu}\big(x_{t-1}^{\text{E2H}}(\nu)\big)\Big).
\end{aligned}
$$

Similarly, these two terms are nonnegative. Hence, $e_t^{\text{E2H}}(\nu) \geq 0$, $\forall t \geq 1$.

Next we derive a quantitative upper bound. For the first difference, note that $x_{t-1}^{\text{E2H}}(0) = 1 - \gamma$ for all $t \geq 1$, and thus

$$
H_{t-1,0}\big(x_{t-1}^{\text{E2H}}(0)\big) - H_{t-1,\nu}\big(x_{t-1}^{\text{E2H}}(0)\big) = \frac{c_\delta \nu}{c\sqrt{a_{t-1}(1-\gamma) - c_{\delta'}\nu}}.
$$

For the second difference, we compute the derivative

$$
H'_{t-1,\nu}(x) = \frac{c_\delta \nu}{2c} \cdot \frac{a_{t-1}}{(a_{t-1}x - c_{\delta'}\nu)^{3/2}},
$$

which is monotonically decreasing in $x$ over its domain. By the mean value theorem, there exists $\zeta_{t-1} \in \big[x_{t-1}^{\text{E2H}}(\nu), 1 - \gamma\big]$ such that $H_{t-1,\nu}\big(x_{t-1}^{\text{E2H}}(0)\big) - H_{t-1,\nu}\big(x_{t-1}^{\text{E2H}}(\nu)\big) = H'_{t-1,\nu}(\zeta_{t-1})\, e_{t-1}^{\text{E2H}}(\nu)$. Since $H'_{t-1,\nu}$ is decreasing and $\zeta_{t-1} \geq x_{t-1}^{\text{E2H}}(\nu)$, we have $H'_{t-1,\nu}(\zeta_{t-1}) \leq H'_{t-1,\nu}(x_{t-1}^{\text{E2H}}(\nu))$. Moreover, since $\{x_t^{\text{E2H}}(\nu)\}_{t\geq 0}$ is monotonically increasing and

$$
x_1^{\text{E2H}}(\nu) = H_{0,\nu}(x_0) = 1 - \gamma - \frac{c_\delta \nu}{c\sqrt{a_0 x_0 - c_{\delta'}\nu}},
$$

we have $x_{t-1}^{\text{E2H}}(\nu) \geq x_1^{\text{E2H}}(\nu)$ for all $t \geq 2$. Using again that $H'_{t-1,\nu}$ is decreasing, it follows that for all $t \geq 2$, $H'_{t-1,\nu}(x_{t-1}^{\text{E2H}}(\nu)) \leq H'_{t-1,\nu}(x_1^{\text{E2H}}(\nu))$. Combining the above displays yields, for every $t = 2, 3, \ldots, L$,

$$
H_{t-1,\nu}\big(x_{t-1}^{\text{E2H}}(0)\big) - H_{t-1,\nu}\big(x_{t-1}^{\text{E2H}}(\nu)\big) \leq H'_{t-1,\nu}\big(x_1^{\text{E2H}}(\nu)\big)\, e_{t-1}^{\text{E2H}}(\nu),
$$

where

$$
H'_{t-1,\nu}\big(x_1^{\text{E2H}}(\nu)\big) = \frac{c_\delta \nu}{2c} \cdot \frac{a_{t-1}}{\Big(a_{t-1}x_1^{\text{E2H}}(\nu) - c_{\delta'}\nu\Big)^{3/2}}.
$$

Therefore, combining the two pieces, we obtain the recursion: for every $t = 2, 3, \ldots, L$,

$$
e_t^{\text{E2H}}(\nu) \leq \frac{c_\delta \nu}{c\sqrt{a_{t-1}(1-\gamma) - c_{\delta'}\nu}} + \frac{c_\delta \nu}{2c} \cdot \frac{a_{t-1}}{\Big(a_{t-1}x_1^{\text{E2H}}(\nu) - c_{\delta'}\nu\Big)^{3/2}} e_{t-1}^{\text{E2H}}(\nu).
$$

Unrolling gives

$$
\begin{aligned}
e_L^{\text{E2H}}(\nu) \leq{}& \sum_{j=1}^{L-1} \left(\prod_{s=j+1}^{L-1} \frac{c_\delta \nu}{2c} \cdot \frac{a_s}{\Big(a_s x_1^{\text{E2H}}(\nu) - c_{\delta'}\nu\Big)^{3/2}}\right) \cdot \frac{c_\delta \nu}{c\sqrt{a_j(1-\gamma) - c_{\delta'}\nu}} \\
&+ \left(\prod_{s=1}^{L-1} \frac{c_\delta \nu}{2c} \cdot \frac{a_s}{\Big(a_s x_1^{\text{E2H}}(\nu) - c_{\delta'}\nu\Big)^{3/2}}\right) \cdot \frac{c_\delta \nu}{c\sqrt{a_0 x_0 - c_{\delta'}\nu}}.
\end{aligned}
$$

Since $a_t = (t/(t+1))^\beta$ is increasing in $t$ for $t \geq 1$, we have $a_t \geq a_1 = 2^{-\beta}$ for all $t \in [L-1]$. Hence, for every $j \in [L-1]$,

$$
\frac{1}{\sqrt{a_j(1-\gamma) - c_{\delta'}\nu}} \leq \frac{1}{\sqrt{2^{-\beta}(1-\gamma) - c_{\delta'}\nu}},
$$

and for every $s \in [L-1]$,

$$
\frac{1}{\Big(a_s x_1^{\text{E2H}}(\nu) - c_{\delta'}\nu\Big)^{3/2}} \leq \frac{1}{\Big(2^{-\beta} x_1^{\text{E2H}}(\nu) - c_{\delta'}\nu\Big)^{3/2}}.
$$

Applying these bounds to the unrolled expression yields

$$e_L^{\text{E2H}}(\nu) \leq \frac{c_\delta \nu}{c\sqrt{2^{-\beta}(1-\gamma) - c_{\delta'}\nu}} \sum_{j=1}^{L-1} \left( \frac{c_\delta \nu}{2c\left(2^{-\beta}x_1^{\text{E2H}}(\nu) - c_{\delta'}\nu\right)^{3/2}} \right)^{L-1-j} \left( \prod_{s=j+1}^{L-1} a_s \right)$$

$$+ \left( \frac{c_\delta \nu}{2c\left(2^{-\beta}x_1^{\text{E2H}}(\nu) - c_{\delta'}\nu\right)^{3/2}} \right)^{L-1} \left( \prod_{s=1}^{L-1} a_s \right) \cdot \frac{c_\delta \nu}{c\sqrt{a_0 x_0 - c_{\delta'}\nu}}.$$

Next, For $0 \leq j \leq L - 1$, $\prod_{s=j+1}^{L-1} a_s = \prod_{s=j+1}^{L-1} \left(\frac{s}{s+1}\right)^\beta = \left(\frac{j+1}{L}\right)^\beta$. Substituting these identities gives

$$e_L^{\text{E2H}}(\nu) \leq \frac{c_\delta \nu}{c\sqrt{2^{-\beta}(1-\gamma) - c_{\delta'}\nu}} \sum_{j=1}^{L-1} \left( \frac{c_\delta \nu}{2c\left(2^{-\beta}x_1^{\text{E2H}}(\nu) - c_{\delta'}\nu\right)^{3/2}} \right)^{L-1-j} \left(\frac{j+1}{L}\right)^\beta$$

$$+ \left( \frac{c_\delta \nu}{2c\left(2^{-\beta}x_1^{\text{E2H}}(\nu) - c_{\delta'}\nu\right)^{3/2}} \right)^{L-1} L^{-\beta} \cdot \frac{c_\delta \nu}{c\sqrt{a_0 x_0 - c_{\delta'}\nu}}$$

$$= \frac{c_\delta \nu}{c\sqrt{2^{-\beta}(1-\gamma) - c_{\delta'}\nu}} \sum_{m=0}^{L-2} \left( \frac{c_\delta \nu}{2c\left(2^{-\beta}x_1^{\text{E2H}}(\nu) - c_{\delta'}\nu\right)^{3/2}} \right)^{m} \left(1 - \frac{m}{L}\right)^\beta$$

$$+ \left( \frac{c_\delta \nu}{2c\left(2^{-\beta}x_1^{\text{E2H}}(\nu) - c_{\delta'}\nu\right)^{3/2}} \right)^{L-1} L^{-\beta} \cdot \frac{c_\delta \nu}{c\sqrt{a_0 x_0 - c_{\delta'}\nu}}.$$

Finally, since for $m \in \{0, 1, \ldots, L-2\}$, $(1 - \frac{m}{L})^\beta \leq e^{-\frac{\beta}{L}m}$,

$$e_L^{\text{E2H}}(\nu) \leq \frac{c_\delta \nu}{c\sqrt{2^{-\beta}(1-\gamma) - c_{\delta'}\nu}} \sum_{m=0}^{L-2} \left[ \frac{c_\delta \nu}{2c\left(2^{-\beta}x_1^{\text{E2H}}(\nu) - c_{\delta'}\nu\right)^{3/2}} \cdot e^{-\beta/L} \right]^{m}$$

$$+ \left( \frac{c_\delta \nu}{2c\left(2^{-\beta}x_1^{\text{E2H}}(\nu) - c_{\delta'}\nu\right)^{3/2}} \right)^{L-1} L^{-\beta} \cdot \frac{c_\delta \nu}{c\sqrt{a_0 x_0 - c_{\delta'}\nu}}$$

$$= \frac{c_\delta \nu}{c\sqrt{2^{-\beta}(1-\gamma) - c_{\delta'}\nu}} \cdot \frac{1 - \left[ \dfrac{c_\delta \nu}{2c\left(2^{-\beta}\left(1-\gamma - \frac{c_\delta \nu}{c\sqrt{a_0 x_0 - c_{\delta'}\nu}}\right) - c_{\delta'}\nu\right)^{3/2}} \cdot e^{-\beta/L} \right]^{L-1}}{1 - \dfrac{c_\delta \nu}{2c\left(2^{-\beta}\left(1-\gamma - \frac{c_\delta \nu}{c\sqrt{a_0 x_0 - c_{\delta'}\nu}}\right) - c_{\delta'}\nu\right)^{3/2}} \cdot e^{-\beta/L}}$$

$$+ \left( \frac{c_\delta \nu}{2c\left(2^{-\beta}\left(1 - \gamma - \frac{c_\delta \nu}{c\sqrt{a_0 x_0 - c_{\delta'}\nu}}\right) - c_{\delta'}\nu\right)^{3/2}} \right)^{L-1} L^{-\beta} \cdot \frac{c_\delta \nu}{c\sqrt{a_0 x_0 - c_{\delta'}\nu}}.$$

Therefore, given $\Delta(\nu, x_0) - \Delta(0, x_0) = e_L^{\text{B}}(\nu) - a_L\, e_L^{\text{E2H}}(\nu)$, combining the lower bound on $e_L^{\text{B}}(\nu)$ and the upper bound

on $e_L^{\mathrm{E2H}}(\nu)$ yields

$$\Delta(\nu, x_0) - \Delta(0, x_0)$$

$$\geq \left[ \frac{c_\delta \nu}{c\sqrt{1-\gamma-c_{\delta'}\nu}} \cdot \frac{1 - \left( \frac{c_\delta \nu}{2c\,(1-\gamma-c_{\delta'}\nu)^{3/2}} \right)^{L-1}}{1 - \frac{c_\delta \nu}{2c\,(1-\gamma-c_{\delta'}\nu)^{3/2}}} + \left( \frac{c_\delta \nu}{2c\,(1-\gamma-c_{\delta'}\nu)^{3/2}} \right)^{L-1} \frac{c_\delta \nu}{c\sqrt{x_0-c_{\delta'}\nu}} \right]$$

$$- a_L \left[ \frac{c_\delta \nu}{c\sqrt{2^{-\beta}(1-\gamma)-c_{\delta'}\nu}} \cdot \frac{1 - \left[ \frac{c_\delta \nu}{2c\left(2^{-\beta}\left(1-\gamma-\frac{c_\delta\nu}{c\sqrt{a_0x_0-c_{\delta'}\nu}}\right)-c_{\delta'}\nu\right)^{3/2}} \cdot e^{-\beta/L} \right]^{L-1}}{1 - \frac{c_\delta \nu}{2c\left(2^{-\beta}\left(1-\gamma-\frac{c_\delta\nu}{c\sqrt{a_0x_0-c_{\delta'}\nu}}\right)-c_{\delta'}\nu\right)^{3/2}} \cdot e^{-\beta/L}} \right.$$

$$\left. + \left( \frac{c_\delta \nu}{2c\left(2^{-\beta}\left(1-\gamma-\frac{c_\delta\nu}{c\sqrt{a_0x_0-c_{\delta'}\nu}}\right)-c_{\delta'}\nu\right)^{3/2}} \right)^{L-1} L^{-\beta} \cdot \frac{c_\delta \nu}{c\sqrt{a_0x_0-c_{\delta'}\nu}} \right]$$

$$> \frac{c_\delta \nu}{c\sqrt{1-\gamma-c_{\delta'}\nu}} \cdot \frac{1 - \left( \frac{c_\delta \nu}{2c\,(1-\gamma-c_{\delta'}\nu)^{3/2}} \right)^{L-1}}{1 - \frac{c_\delta \nu}{2c\,(1-\gamma-c_{\delta'}\nu)^{3/2}}}$$

$$- a_L \left[ \frac{c_\delta \nu}{c\sqrt{2^{-\beta}(1-\gamma)-c_{\delta'}\nu}} \cdot \frac{1}{1 - \frac{c_\delta \nu}{2c\left(2^{-\beta}\left(1-\gamma-\frac{c_\delta\nu}{c\sqrt{a_0x_0-c_{\delta'}\nu}}\right)-c_{\delta'}\nu\right)^{3/2}} \cdot e^{-\beta/L}} \right.$$

$$\left. + \left( \frac{c_\delta \nu}{2c\left(2^{-\beta}\left(1-\gamma-\frac{c_\delta\nu}{c\sqrt{a_0x_0-c_{\delta'}\nu}}\right)-c_{\delta'}\nu\right)^{3/2}} \right)^{L-1} L^{-\beta} \cdot \frac{c_\delta \nu}{c\sqrt{a_0x_0-c_{\delta'}\nu}} \right]$$

When the derived lower bound is greater than $-\frac{1}{2}\Delta(0, x_0) = -\frac{1}{2}(a_L - 1)(1 - \gamma)$, then it follows immediately that

$$\Delta(\nu, x_0) = \Delta(0, x_0) + \big(\Delta(\nu, x_0) - \Delta(0, x_0)\big) > \frac{1}{2}(a_L - 1)(1 - \gamma) > 0.$$

Therefore, we define the corresponding constraint by $\mathcal{N}\big(\beta', \beta, \nu, V_{p_0}(\hat{\theta}_0)\big) < 0$ as in Definition A.2, under this constraint, we conclude that

$$\big(G \circ H_{L-1,\nu} \circ \cdots \circ H_{0,\nu}\big)(x_0) > F_\nu^{\circ L}(x_0),$$

i.e., the easy-to-hard lower bound is strictly larger than the baseline lower bound. $\qquad\square$

## C.2 PROOF OF COROLLARY 5.3

*Proof.* Let

$$H_0(V) := 1 - \gamma - \frac{c_\delta \nu}{c\sqrt{a_0 V - c_{\delta'}\nu}},$$

which is strictly increasing in $V$. Then, the inequality $x_-(2^{-\beta}, \nu) < H_0\big(V_{p_0}(\hat{\theta}_0)\big) < x_+(2^{-\beta}, \nu)$ is equivalent to $V_-(\nu) < V_{p_0}(\hat{\theta}_0) < V_+(\nu)$, where $V_\pm(\nu)$ are the unique solutions to $H_0(V) = x_\pm(2^{-\beta}, \nu)$, respectively. Solving this gives

$$\sqrt{a_0 V_\pm(\nu) - c_{\delta'}\nu} = \frac{c_\delta \nu}{c\left(1 - \gamma - x_\pm(2^{-\beta}, \nu)\right)}.$$

Moreover, using the fact that $x_\pm(2^{-\beta}, \nu)$ are the fixed points, i.e.,

$$x_\pm(2^{-\beta}, \nu) = 1 - \gamma - \frac{c_\delta \nu}{c\sqrt{2^{-\beta} x_\pm(2^{-\beta}, \nu) - c_{\delta'}\nu}},$$

we obtain

$$\frac{c_\delta \nu}{c\left(1 - \gamma - x_\pm(2^{-\beta}, \nu)\right)} = \sqrt{2^{-\beta} x_\pm(2^{-\beta}, \nu) - c_{\delta'}\nu}.$$

Therefore, $V_\pm(\nu) = \frac{2^{-\beta}}{a_0} x_\pm(2^{-\beta}, \nu)$. Consequently, the second constraint is equivalent to

$$\frac{2^{-\beta}}{a_0} x_-(2^{-\beta}, \nu) < V_{p_0}(\hat\theta_0) < \frac{2^{-\beta}}{a_0} x_+(2^{-\beta}, \nu).$$

Intersecting with the first constraint $x_-(2^{-\beta}, \nu) < V_{p_0}(\hat\theta_0) < x_+(2^{-\beta}, \nu)$ yields that the feasible set of $V_{p_0}(\hat\theta_0)$ is

$$\mathcal{I}_{\mathcal{M}}(\beta', \beta, \nu) = \left(x_-(2^{-\beta}, \nu), \frac{2^{-\beta}}{a_0} x_+(2^{-\beta}, \nu)\right).$$

For the interval length $\left|\mathcal{I}_{\mathcal{M}}(\beta', \beta, \nu)\right|$, we have

$$\left|\mathcal{I}_{\mathcal{M}}(\beta', \beta, \nu)\right| = \left(x_+(2^{-\beta}, \nu) - x_-(2^{-\beta}, \nu)\right) - \left(1 - \frac{2^{-\beta}}{a_0}\right) x_+(2^{-\beta}, \nu)$$

$$\geq \left(x_+(2^{-\beta}, \nu) - x_-(2^{-\beta}, \nu)\right) - \left(1 - \frac{2^{-\beta}}{a_0}\right)(1 - \gamma).$$

Applying Corollary B.3 with parameter $2^{-\beta}$ gives

$$\left|\mathcal{I}_{\mathcal{M}}(\beta', \beta, \nu)\right| \geq \frac{2^{-\beta}}{a_0}(1 - \gamma) - 2^\beta c_{\delta'}\nu - \frac{3\sqrt{3}}{2} \cdot \frac{c_\delta \nu}{c\sqrt{2^{-\beta}(1 - \gamma) - c_{\delta'}\nu}}.$$

By Corollary B.3, we have $x_-(2^{-\beta}, 0) = 0$ and $x_+(2^{-\beta}, 0) = 1 - \gamma$, hence $\left|\mathcal{I}_{\mathcal{M}}(\beta', \beta, 0)\right| = \frac{2^{-\beta}}{a_0}(1 - \gamma)$. Therefore,

$$\left|\mathcal{I}_{\mathcal{M}}(\beta', \beta, 0)\right| - \left|\mathcal{I}_{\mathcal{M}}(\beta', \beta, \nu)\right| \leq 2^\beta c_{\delta'}\nu + \frac{3\sqrt{3}}{2} \cdot \frac{c_\delta \nu}{c\sqrt{2^{-\beta}(1 - \gamma) - c_{\delta'}\nu}}.$$

Moreover,

$$\left|\mathcal{I}_{\mathcal{M}}(\beta', \beta, 0)\right| - \left|\mathcal{I}_{\mathcal{M}}(\beta', \beta, \nu)\right| = \frac{2^{-\beta}}{a_0}(1 - \gamma) - \left(\frac{2^{-\beta}}{a_0} x_+(2^{-\beta}, \nu) - x_-(2^{-\beta}, \nu)\right)$$

$$= \frac{2^{-\beta}}{a_0}\left(1 - \gamma - x_+(2^{-\beta}, \nu)\right) + x_-(2^{-\beta}, \nu).$$

Since $x_+(2^{-\beta}, \nu) < 1 - \gamma$, the first term above is nonnegative, hence $\left|\mathcal{I}_{\mathcal{M}}(\beta', \beta, 0)\right| - \left|\mathcal{I}_{\mathcal{M}}(\beta', \beta, \nu)\right| \geq x_-(2^{-\beta}, \nu)$. By the definition of $x_-(2^{-\beta}, \nu)$,

$$x_-(2^{-\beta}, \nu) = 2^\beta c_{\delta'}\nu + \left(1 - \gamma - 2^\beta c_{\delta'}\nu\right) y_-(\nu) \geq 2^\beta c_{\delta'}\nu,$$

which completes the proof. $\qquad\square$

### C.3  PROOF OF PROPOSITION 5.5

*Proof.* Proposition 5.5 follows immediately by combining Lemma C.1, Lemma C.2, and Lemma C.3. $\qquad\square$

**Lemma C.1.** *Under the notation of Proposition 5.5, for fixed $(\beta', \beta)$, the length of the interval $\mathcal{I}_{\mathcal{N}}(\beta', \beta, \nu)$ for which $\mathcal{N}\left(\beta', \beta, \nu, V_{p_0}(\hat\theta_0)\right) < 0$ is monotonically decreasing in $\nu$, and $\mathcal{I}_{\mathcal{N}}(\beta', \beta, \nu)$ can be written in the form*

$$\mathcal{I}_{\mathcal{N}}(\beta', \beta, \nu) = \left(x(\beta', \beta, \nu), 1 - \gamma\right).$$

*Proof.* Let $x_0 := V_{p_0}(\hat{\theta}_0)$ for convenience. First, we show that the additional constraint $\mathcal{N}(\beta', \beta, \nu, x_0) < 0$ induces an admissible region that is monotonically decreasing in $\nu$.

Define

$$
\mathcal{E}(\beta', \beta, \nu, x_0) := \frac{c_\delta \nu}{c\sqrt{1 - \gamma - c_{\delta'}\nu}} \cdot \frac{1 - \left(\frac{c_\delta \nu}{2c\,(1-\gamma-c_{\delta'}\nu)^{3/2}}\right)^{L-1}}{1 - \frac{c_\delta \nu}{2c\,(1-\gamma-c_{\delta'}\nu)^{3/2}}}
$$
$$
- a_L \Bigg[ \frac{c_\delta \nu}{c\sqrt{2^{-\beta}(1-\gamma) - c_{\delta'}\nu}} \cdot \frac{1}{1 - \frac{c_\delta \nu}{2c\left(2^{-\beta}\left(1-\gamma-\frac{c_\delta \nu}{c\sqrt{a_0 x_0 - c_{\delta'}\nu}}\right) - c_{\delta'}\nu\right)^{3/2}} \cdot e^{-\beta/L}}
$$
$$
+ \left( \frac{c_\delta \nu}{2c\left(2^{-\beta}\left(1-\gamma-\frac{c_\delta \nu}{c\sqrt{a_0 x_0 - c_{\delta'}\nu}}\right) - c_{\delta'}\nu\right)^{3/2}} \right)^{L-1} L^{-\beta} \cdot \frac{c_\delta \nu}{c\sqrt{a_0 x_0 - c_{\delta'}\nu}} \Bigg].
$$

Then, by Theorem 5.2, $\mathcal{N}(\beta', \beta, \nu, x_0) < 0$ is equivalent to $\mathcal{E}(\beta', \beta, \nu, x_0) > -\frac{1}{2}(a_L - 1)(1-\gamma)$. Moreover, the right-hand side above is a fixed constant since $\beta'$ is fixed.

Next, note that $\mathcal{E}(\beta', \beta, \nu, x_0)$ is monotonically increasing in $x_0$. Moreover, as $x_0$ decreases to 0, for denominators, the term

$$
1 - \frac{c_\delta \nu}{2c\left(2^{-\beta}\left(1 - \gamma - \frac{c_\delta \nu}{c\sqrt{a_0 x_0 - c_{\delta'}\nu}}\right) - c_{\delta'}\nu\right)^{3/2}} \cdot e^{-\beta/L}
$$

is the first to approach 0, which implies that $\mathcal{E}(\beta', \beta, \nu, x_0) \to -\infty$. On the other hand, when $\nu = 0$ we have

$$
\mathcal{E}(\beta', \beta, 0, x_0) = 0 > -\frac{1}{2}(a_L - 1)(1-\gamma)
$$

for all $x_0$. Hence, by continuity, for sufficiently small $\nu > 0$, the set of $x_0$ satisfying $\mathcal{E}(\beta', \beta, \nu, x_0) > -\frac{1}{2}(a_L - 1)(1-\gamma)$ is non-empty. Combining the above, as $\nu$ increases gradually from 0, the set of $x_0$ such that $\mathcal{N}(\beta', \beta, \nu, x_0) < 0$ must lie in a non-empty interval of the form

$$
\mathcal{I}_{\mathcal{N}}(\beta', \beta, \nu) = \big(x(\beta', \beta, \nu),\, 1 - \gamma\big).
$$

Therefore, it suffices to show that the minimal admissible threshold $x(\beta', \beta, \nu)$ is monotonically increasing in $\nu$. Define

$$
\Phi(\beta', \beta, \nu, x) := \mathcal{E}(\beta', \beta, \nu, x) + \frac{1}{2}(a_L - 1)(1-\gamma),
$$

By definition of $x(\beta', \beta, \nu)$, we have $\Phi\big(\beta', \beta, \nu, x(\beta', \beta, \nu)\big) = 0$. The implicit function theorem implies that

$$
x'(\nu) = -\frac{\partial_\nu \Phi\big(\beta', \beta, \nu, x(\beta', \beta, \nu)\big)}{\partial_x \Phi\big(\beta', \beta, \nu, x(\beta', \beta, \nu)\big)} = -\frac{\partial_\nu \mathcal{E}\big(\beta', \beta, \nu, x(\beta', \beta, \nu)\big)}{\partial_x \mathcal{E}\big(\beta', \beta, \nu, x(\beta', \beta, \nu)\big)}.
$$

Since $\partial_x \mathcal{E}(\beta', \beta, \nu, x) > 0$, to show $x'(\nu) > 0$, it suffices to show that $\mathcal{E}(\beta', \beta, \nu, x_0)$ is monotonically decreasing in $\nu$.

Note first that the last term inside the brackets,

$$
\left( \frac{c_\delta \nu}{2c\left(2^{-\beta}\left(1 - \gamma - \frac{c_\delta \nu}{c\sqrt{a_0 x_0 - c_{\delta'}\nu}}\right) - c_{\delta'}\nu\right)^{3/2}} \right)^{L-1} L^{-\beta} \cdot \frac{c_\delta \nu}{c\sqrt{a_0 x_0 - c_{\delta'}\nu}},
$$

is increasing as $\nu$ increases. Hence, after multiplying by the coefficient $-a_L$, this contribution is decreasing in $\nu$. It therefore suffices to prove that the remaining part,

$$
\frac{c_\delta \nu}{c\sqrt{1 - \gamma - c_{\delta'}\nu}} \cdot \frac{1 - \left(\frac{c_\delta \nu}{2c\,(1-\gamma-c_{\delta'}\nu)^{3/2}}\right)^{L-1}}{1 - \frac{c_\delta \nu}{2c\,(1-\gamma-c_{\delta'}\nu)^{3/2}}}
$$
$$
- a_L \Bigg[ \frac{c_\delta \nu}{c\sqrt{2^{-\beta}(1-\gamma) - c_{\delta'}\nu}} \cdot \frac{1}{1 - \frac{c_\delta \nu}{2c\left(2^{-\beta}\left(1-\gamma-\frac{c_\delta \nu}{c\sqrt{a_0 x_0 - c_{\delta'}\nu}}\right) - c_{\delta'}\nu\right)^{3/2}} \cdot e^{-\beta/L}} \Bigg].
$$

has strictly negative derivative with respect to $\nu$.

To this end, it is enough to verify the following two comparison statements: first, both the function value and the derivative (with respect to $\nu$) of

$$U(\nu) = \frac{c_\delta \nu}{c\sqrt{1 - \gamma - c_{\delta'} \nu}}$$

are strictly smaller than those of

$$\widetilde{U}(\nu) = \frac{c_\delta \nu}{c\sqrt{2^{-\beta}(1 - \gamma) - c_{\delta'} \nu}},$$

and second, both the function value and the derivative (with respect to $\nu$) of

$$V(\nu) = \frac{1 - \left(\frac{c_\delta \nu}{2c\,(1 - \gamma - c_{\delta'}\nu)^{3/2}}\right)^{L-1}}{1 - \frac{c_\delta \nu}{2c\,(1 - \gamma - c_{\delta'}\nu)^{3/2}}}$$

are strictly smaller than those of

$$\widetilde{V}(\nu) = \frac{1}{1 - \frac{c_\delta \nu}{2c\left(2^{-\beta}\left(1 - \gamma - \frac{c_\delta \nu}{c\sqrt{a_0 x_0 - c_{\delta'}\nu}}\right) - c_{\delta'}\nu\right)^{3/2}} \cdot e^{-\beta/L}}.$$

Indeed, by the product rule,

$$\frac{\mathrm{d}}{\mathrm{d}\nu}\left(U(\nu)V(\nu) - a_L\,\widetilde{U}(\nu)\widetilde{V}(\nu)\right) = \left(U'(\nu)V(\nu) + U(\nu)V'(\nu)\right) - a_L\left(\widetilde{U}'(\nu)\widetilde{V}(\nu) + \widetilde{U}(\nu)\widetilde{V}'(\nu)\right) < 0,$$

since $a_L > 1$, and

$$U(\nu), \widetilde{U}(\nu), V(\nu), \widetilde{V}(\nu), U'(\nu), \widetilde{U}'(\nu), V'(\nu), \widetilde{V}'(\nu) > 0.$$

We first prove that $U(\nu) < \widetilde{U}(\nu)$. Since $0 < 2^{-\beta} < 1$, we have

$$2^{-\beta}(1 - \gamma) - c_{\delta'}\nu \;<\; 1 - \gamma - c_{\delta'}\nu,$$

and hence $U(\nu) < \widetilde{U}(\nu)$.

We second prove that $U'(\nu) < \widetilde{U}'(\nu)$. By direct differentiation, we have

$$U'(\nu) = \frac{c_\delta}{c}\left((1 - \gamma - c_{\delta'}\nu)^{-1/2} + \frac{c_{\delta'}\nu}{2}(1 - \gamma - c_{\delta'}\nu)^{-3/2}\right).$$

Similarly,

$$\widetilde{U}'(\nu) = \frac{c_\delta}{c}\left((2^{-\beta}(1 - \gamma) - c_{\delta'}\nu)^{-1/2} + \frac{c_{\delta'}\nu}{2}(2^{-\beta}(1 - \gamma) - c_{\delta'}\nu)^{-3/2}\right).$$

Since $2^{-\beta}(1 - \gamma) - c_{\delta'}\nu < (1 - \gamma) - c_{\delta'}\nu$, $U'(\nu) < \widetilde{U}'(\nu)$.

We third prove that $V(\nu) < \widetilde{V}(\nu)$ We can rewrite $V(\nu)$ as the finite geometric sum

$$V(\nu) = \sum_{j=0}^{L-2}\left(\frac{c_\delta \nu}{2c\,(1 - \gamma - c_{\delta'}\nu)^{3/2}}\right)^j.$$

Similarly,

$$\widetilde{V}(\nu) = \sum_{j=0}^{\infty}\left(\frac{c_\delta \nu}{2c\left(2^{-\beta}\left(1 - \gamma - \frac{c_\delta \nu}{c\sqrt{a_0 x_0 - c_{\delta'}\nu}}\right) - c_{\delta'}\nu\right)^{3/2}} \cdot e^{-\beta/L}\right)^j.$$

Therefore, it is enough to show that the common ratio of the geometric sum defining $V(\nu)$ is strictly smaller than the common ratio of the geometric series defining $\widetilde{V}(\nu)$, namely,

$$\frac{c_\delta \nu}{2c\left(1 - \gamma - c_{\delta'}\nu\right)^{3/2}} \; < \; \frac{c_\delta \nu}{2c\left(2^{-\beta}\left(1 - \gamma - \frac{c_\delta \nu}{c\sqrt{a_0 x_0 - c_{\delta'}\nu}}\right) - c_{\delta'}\nu\right)^{3/2}} \cdot e^{-\beta/L}.$$

This is equivalent to

$$2^{-\beta}\left(1 - \gamma - \frac{c_\delta \nu}{c\sqrt{a_0 x_0 - c_{\delta'}\nu}}\right) - c_{\delta'}\nu \; < \; e^{-2\beta/(3L)}\left(1 - \gamma - c_{\delta'}\nu\right).$$

Since $L \geq 2$ implies $\frac{1}{L} < \frac{3}{2}\log 2$, we have $e^{\beta\left(\frac{3}{2}\log 2 - \frac{1}{L}\right)} > 1$. Rearranging gives $e^{-2\beta/(3L)} > 2^{-\beta}$. Moreover,

$$2^{-\beta}\left(1 - \gamma - \frac{c_\delta \nu}{c\sqrt{a_0 x_0 - c_{\delta'}\nu}}\right) - c_{\delta'}\nu \; \leq \; 2^{-\beta}(1 - \gamma) - c_{\delta'}\nu,$$

and

$$2^{-\beta}(1 - \gamma) - c_{\delta'}\nu \; \leq \; e^{-2\beta/(3L)}(1 - \gamma) - e^{-2\beta/(3L)}c_{\delta'}\nu = e^{-2\beta/(3L)}(1 - \gamma - c_{\delta'}\nu).$$

Combining the last two displays gives

$$2^{-\beta}\left(1 - \gamma - \frac{c_\delta \nu}{c\sqrt{a_0 x_0 - c_{\delta'}\nu}}\right) - c_{\delta'}\nu \; < \; e^{-2\beta/(3L)}(1 - \gamma - c_{\delta'}\nu),$$

which proves the desired ratio inequality.

We finally prove that $V'(\nu) < \widetilde{V}'(\nu)$. Differentiating term-by-term, we obtain

$$V'(\nu) = \frac{\mathrm{d}}{\mathrm{d}\nu}\left(\frac{c_\delta \nu}{2c\left(1 - \gamma - c_{\delta'}\nu\right)^{3/2}}\right) \cdot \sum_{j=1}^{L-2} j\left(\frac{c_\delta \nu}{2c\left(1 - \gamma - c_{\delta'}\nu\right)^{3/2}}\right)^{j-1},$$

and

$$\widetilde{V}'(\nu) = \frac{\mathrm{d}}{\mathrm{d}\nu}\left(\frac{c_\delta \nu}{2c\left(2^{-\beta}\left(1 - \gamma - \frac{c_\delta \nu}{c\sqrt{a_0 x_0 - c_{\delta'}\nu}}\right) - c_{\delta'}\nu\right)^{3/2}} \cdot e^{-\beta/L}\right)$$

$$\cdot \sum_{j=1}^{\infty} j\left(\frac{c_\delta \nu}{2c\left(2^{-\beta}\left(1 - \gamma - \frac{c_\delta \nu}{c\sqrt{a_0 x_0 - c_{\delta'}\nu}}\right) - c_{\delta'}\nu\right)^{3/2}} \cdot e^{-\beta/L}\right)^{j-1}.$$

Since we have already shown that

$$\frac{c_\delta \nu}{2c\left(1 - \gamma - c_{\delta'}\nu\right)^{3/2}} \; < \; \frac{c_\delta \nu}{2c\left(2^{-\beta}\left(1 - \gamma - \frac{c_\delta \nu}{c\sqrt{a_0 x_0 - c_{\delta'}\nu}}\right) - c_{\delta'}\nu\right)^{3/2}} \cdot e^{-\beta/L},$$

it remains to prove that

$$\frac{\mathrm{d}}{\mathrm{d}\nu}\left(\frac{c_\delta \nu}{2c\left(1 - \gamma - c_{\delta'}\nu\right)^{3/2}}\right) \; < \; \frac{\mathrm{d}}{\mathrm{d}\nu}\left(\frac{c_\delta \nu}{2c\left(2^{-\beta}\left(1 - \gamma - \frac{c_\delta \nu}{c\sqrt{a_0 x_0 - c_{\delta'}\nu}}\right) - c_{\delta'}\nu\right)^{3/2}} \cdot e^{-\beta/L}\right).$$

For the left-hand side, direct differentiation gives

$$\frac{\mathrm{d}}{\mathrm{d}\nu}\left(\frac{c_\delta \nu}{2c\left(1 - \gamma - c_{\delta'}\nu\right)^{3/2}}\right) = \frac{c_\delta}{2c}\left((1 - \gamma - c_{\delta'}\nu)^{-3/2} + \frac{3}{2}c_{\delta'}\nu\left(1 - \gamma - c_{\delta'}\nu\right)^{-5/2}\right)$$

$$= \frac{c_\delta}{2c} \cdot \frac{1 - \gamma + \frac{1}{2}c_{\delta'}\nu}{(1 - \gamma - c_{\delta'}\nu)^{5/2}}.$$

For the right-hand side, direct differentiation yields

$$\frac{\mathrm{d}}{\mathrm{d}\nu}\left(\frac{c_\delta \nu}{2c\left(2^{-\beta}\left(1-\gamma-\frac{c_\delta \nu}{c\sqrt{a_0 x_0 - c_{\delta'}\nu}}\right)-c_{\delta'}\nu\right)^{3/2}}\cdot e^{-\beta/L}\right)$$

$$=\frac{c_\delta}{2c}e^{-\beta/L}\left[\left(2^{-\beta}\left(1-\gamma-\frac{c_\delta \nu}{c\sqrt{a_0 x_0 - c_{\delta'}\nu}}\right)-c_{\delta'}\nu\right)^{-3/2}\right.$$

$$\left.-\frac{3}{2}\nu\frac{\mathrm{d}}{\mathrm{d}\nu}\left(2^{-\beta}\left(1-\gamma-\frac{c_\delta \nu}{c\sqrt{a_0 x_0 - c_{\delta'}\nu}}\right)-c_{\delta'}\nu\right)\cdot\left(2^{-\beta}\left(1-\gamma-\frac{c_\delta \nu}{c\sqrt{a_0 x_0 - c_{\delta'}\nu}}\right)-c_{\delta'}\nu\right)^{-5/2}\right]$$

$$=\frac{c_\delta}{2c}e^{-\beta/L}\cdot\frac{2^{-\beta}\left(1-\gamma-\frac{c_\delta \nu}{c\sqrt{a_0 x_0 - c_{\delta'}\nu}}\right)-c_{\delta'}\nu-\frac{3}{2}\nu\frac{\mathrm{d}}{\mathrm{d}\nu}\left(2^{-\beta}\left(1-\gamma-\frac{c_\delta \nu}{c\sqrt{a_0 x_0 - c_{\delta'}\nu}}\right)-c_{\delta'}\nu\right)}{\left(2^{-\beta}\left(1-\gamma-\frac{c_\delta \nu}{c\sqrt{a_0 x_0 - c_{\delta'}\nu}}\right)-c_{\delta'}\nu\right)^{5/2}}.$$

Next we compute the inner derivative explicitly:

$$\frac{\mathrm{d}}{\mathrm{d}\nu}\left(2^{-\beta}\left(1-\gamma-\frac{c_\delta \nu}{c\sqrt{a_0 x_0 - c_{\delta'}\nu}}\right)-c_{\delta'}\nu\right)=2^{-\beta}\left(-\frac{c_\delta}{c\sqrt{a_0 x_0 - c_{\delta'}\nu}}-\frac{c_\delta \nu}{c}\cdot\frac{c_{\delta'}}{2}\left(a_0 x_0 - c_{\delta'}\nu\right)^{-3/2}\right)-c_{\delta'}.$$

Substituting this expression back and simplifying, we obtain

$$2^{-\beta}\left(1-\gamma-\frac{c_\delta \nu}{c\sqrt{a_0 x_0 - c_{\delta'}\nu}}\right)-c_{\delta'}\nu-\frac{3}{2}\nu\frac{\mathrm{d}}{\mathrm{d}\nu}\left(2^{-\beta}\left(1-\gamma-\frac{c_\delta \nu}{c\sqrt{a_0 x_0 - c_{\delta'}\nu}}\right)-c_{\delta'}\nu\right)$$

$$=2^{-\beta}(1-\gamma)+2^{-\beta-1}\frac{c_\delta \nu}{c\sqrt{a_0 x_0 - c_{\delta'}\nu}}+\frac{1}{2}c_{\delta'}\nu+3\cdot 2^{-\beta-2}\frac{c_\delta c_{\delta'}\nu^2}{c\left(a_0 x_0 - c_{\delta'}\nu\right)^{3/2}}$$

$$\geq 2^{-\beta}(1-\gamma)+\frac{1}{2}c_{\delta'}\nu.$$

Therefore,

$$\frac{\mathrm{d}}{\mathrm{d}\nu}\left(\frac{c_\delta \nu}{2c\left(2^{-\beta}\left(1-\gamma-\frac{c_\delta \nu}{c\sqrt{a_0 x_0 - c_{\delta'}\nu}}\right)-c_{\delta'}\nu\right)^{3/2}}\cdot e^{-\beta/L}\right)$$

$$\geq\frac{c_\delta}{2c}e^{-\beta/L}\cdot\frac{2^{-\beta}(1-\gamma)+\frac{1}{2}c_{\delta'}\nu}{\left(2^{-\beta}\left(1-\gamma-\frac{c_\delta \nu}{c\sqrt{a_0 x_0 - c_{\delta'}\nu}}\right)-c_{\delta'}\nu\right)^{5/2}}.$$

Moreover, we have

$$2^{-\beta}\left(1-\gamma-\frac{c_\delta \nu}{c\sqrt{a_0 x_0 - c_{\delta'}\nu}}\right)-c_{\delta'}\nu\;\leq\;2^{-\beta}(1-\gamma)-c_{\delta'}\nu\;\leq\;2^{-\beta}(1-\gamma-c_{\delta'}\nu),$$

Hence, plugging this into the previous display yields

$$\frac{\mathrm{d}}{\mathrm{d}\nu}\left(\frac{c_\delta \nu}{2c\left(2^{-\beta}\left(1-\gamma-\frac{c_\delta \nu}{c\sqrt{a_0 x_0 - c_{\delta'}\nu}}\right)-c_{\delta'}\nu\right)^{3/2}}\cdot e^{-\beta/L}\right)$$

$$\geq\frac{c_\delta}{2c}\cdot\frac{e^{-\beta/L}\left(2^{3\beta/2}(1-\gamma)+2^{5\beta/2}\cdot\frac{1}{2}c_{\delta'}\nu\right)}{(1-\gamma-c_{\delta'}\nu)^{5/2}}.$$

Finally, since $L\geq 2$ implies $\frac{1}{L}<\frac{3}{2}\log 2$, we have $e^{-\beta/L}2^{3\beta/2}=e^{\beta(\frac{3}{2}\log 2-\frac{1}{L})}>1$, and similarly (because $\frac{1}{L}<\frac{5}{2}\log 2$) we have $e^{-\beta/L}2^{5\beta/2}=e^{\beta(\frac{5}{2}\log 2-\frac{1}{L})}>1$. Therefore,

$$e^{-\beta/L}\left(2^{3\beta/2}(1-\gamma)+2^{5\beta/2}\cdot\frac{1}{2}c_{\delta'}\nu\right)>1-\gamma+\frac{1}{2}c_{\delta'}\nu,$$

which implies

$$\frac{d}{d\nu}\left(\frac{c_\delta\nu}{2c\left(2^{-\beta}\left(1-\gamma-\frac{c_\delta\nu}{c\sqrt{a_0 x_0-c_{\delta'}\nu}}\right)-c_{\delta'}\nu\right)^{3/2}}\cdot e^{-\beta/L}\right)$$
$$> \frac{c_\delta}{2c}\cdot\frac{1-\gamma+\frac{1}{2}c_{\delta'}\nu}{(1-\gamma-c_{\delta'}\nu)^{5/2}} = \frac{d}{d\nu}\left(\frac{c_\delta\nu}{2c\left(1-\gamma-c_{\delta'}\nu\right)^{3/2}}\right)$$

as desired. □

**Lemma C.2.** *Under the notation of Proposition 5.5 and Lemma C.1, for fixed $(\beta', \beta)$, the threshold $x(\nu)$ satisfies $x(0) = 0$ and*

$$x'(\nu) = \frac{c_{\delta'}}{a_0} + \frac{2}{a_0}\left(\frac{c_\delta}{c(1-\gamma)}\right)^2 \nu + O(\nu^{5/3}) \quad \text{as } \nu \to 0,$$

*where $a_0 = L/\sum_{i=1}^{L} i^{-\beta'}$.*

*Proof.* By the notation and proof of Lemma C.1, for fixed $(\beta', \beta)$, it suffices to prove that, as $\nu \to 0$,

$$x(\nu) = \frac{c_{\delta'}}{a_0}\nu + \frac{1}{a_0}\left(\frac{c_\delta}{c(1-\gamma)}\right)^2 \nu^2 + O(\nu^{8/3}).$$

By definition of the threshold $x(\nu)$, it satisfies the boundary condition $\mathcal{E}(\beta', \beta, \nu, x(\nu)) = -\frac{1}{2}(a_L - 1)(1 - \gamma)$. Note that the right-hand side is $\Theta(1)$. Consequently, as $\nu \to 0$, the left-hand side must also be $\Theta(1)$. Equivalently, if we expand $\mathcal{E}(\beta', \beta, \nu, x(\nu))$ in a Puiseux series, then the smallest exponent of $\nu$ appearing in the expansion of $\mathcal{E}(\beta', \beta, \nu, x(\nu))$ must be 0.

We analyze $\mathcal{E}(\beta', \beta, \nu, x(\nu))$ term-by-term. Consider first

$$T_1(\nu) := \frac{c_\delta\nu}{c\sqrt{1-\gamma-c_{\delta'}\nu}} \cdot \frac{1-\left(\frac{c_\delta\nu}{2c\,(1-\gamma-c_{\delta'}\nu)^{3/2}}\right)^{L-1}}{1-\frac{c_\delta\nu}{2c\,(1-\gamma-c_{\delta'}\nu)^{3/2}}}.$$

As $\nu \to 0$,

$$\frac{c_\delta\nu}{c\sqrt{1-\gamma-c_{\delta'}\nu}} = \frac{c_\delta}{c\sqrt{1-\gamma}}\nu + O(\nu^2).$$

Next, define

$$q(\nu) := \frac{c_\delta\nu}{2c\,(1-\gamma-c_{\delta'}\nu)^{3/2}}.$$

Then we have

$$q(\nu) = \frac{c_\delta}{2c(1-\gamma)^{3/2}}\nu + O(\nu^2),$$

Moreover,

$$\frac{1-q(\nu)^{L-1}}{1-q(\nu)} = \sum_{j=0}^{L-2} q(\nu)^j = 1 + q(\nu) + O\big(q(\nu)^2\big) = 1 + O(\nu),$$

Therefore,

$$T_1(\nu) = \frac{c_\delta}{c\sqrt{1-\gamma}}\nu + O(\nu^2).$$

In particular, the smallest power of $\nu$ appearing in the Puiseux expansion of $T_1(\nu)$ is $\nu$. Consequently, $T_1(\nu)$ cannot contribute a $\Theta(1)$ term to $\mathcal{E}(\beta', \beta, \nu, x(\nu))$ as $\nu \to 0$.

Next we analyze

$$T_2(\nu) := \left(\frac{c_\delta\nu}{2c\left(2^{-\beta}\left(1-\gamma-\frac{c_\delta\nu}{c\sqrt{a_0 x(\nu)-c_{\delta'}\nu}}\right)-c_{\delta'}\nu\right)^{3/2}}\right)^{L-1} L^{-\beta} \cdot \frac{c_\delta\nu}{c\sqrt{a_0 x(\nu)-c_{\delta'}\nu}}.$$

Our goal is to determine whether there exists a choice of $x(\nu)$ such that $T_2(\nu)$ can contribute a $\Theta(1)$ term to $\mathcal{E}(\beta', \beta, \nu, x(\nu))$ as $\nu \to 0$. To this end, consider the Puiseux expansion of

$$r(\nu) := \frac{c_\delta \nu}{c\sqrt{a_0 x(\nu) - c_{\delta'}\nu}}.$$

When $r(\nu)$ has a negative leading exponent, $r(\nu) \to +\infty$ as $\nu \to 0$. Then the inner radicand $2^{-\beta}\left(1 - \gamma - r(\nu)\right) - c_{\delta'}\nu$ tends to $-\infty$, and hence becomes negative for all sufficiently small $\nu$. This violates our standing well-definedness requirement that every denominator appearing in $\mathcal{E}(\beta', \beta, \nu, x(\nu))$ remain strictly positive.

When $r(\nu)$ has a positive leading exponent, $r(\nu) = o(1)$ as $\nu \to 0$. In order for $T_2(\nu)$ to be $\Theta(1)$, the factor

$$\left(\frac{c_\delta \nu}{2c\left(2^{-\beta}\left(1 - \gamma - r(\nu)\right) - c_{\delta'}\nu\right)^{3/2}}\right)^{L-1}$$

must contribute a negative power of $\nu$, which forces

$$\frac{c_\delta \nu}{2c\left(2^{-\beta}\left(1 - \gamma - r(\nu)\right) - c_{\delta'}\nu\right)^{3/2}} \to +\infty \qquad \text{as } \nu \to 0.$$

However, this implies that in the term

$$T_3(\nu) := \frac{c_\delta \nu}{c\sqrt{2^{-\beta}(1 - \gamma) - c_{\delta'}\nu}} \cdot \frac{1}{1 - \dfrac{c_\delta \nu}{2c\left(2^{-\beta}\left(1 - \gamma - r(\nu)\right) - c_{\delta'}\nu\right)^{3/2}} \cdot e^{-\beta/L}},$$

the denominator becomes negative for all sufficiently small $\nu$, again contradicting the requirement that all denominators remain strictly positive.

Consequently, the only remaining possibility consistent with well-definedness is that $r(\nu)$ has leading exponent $0$. Therefore,

$$\frac{c_\delta \nu}{2c\left(2^{-\beta}\left(1 - \gamma - r(\nu)\right) - c_{\delta'}\nu\right)^{3/2}}$$

must also have leading exponent $0$. We now show that this is equivalent to the existence of a constant $d_2 > 0$ such that $r(\nu) = 1 - \gamma - d_2\,\nu^{2/3} + o(\nu^{2/3})$. Since the above quantity is $\Theta(1)$, its denominator must satisfy $2^{-\beta}\left(1 - \gamma - r(\nu)\right) - c_{\delta'}\nu = \Theta(\nu^{2/3})$. Since $\nu = o(\nu^{2/3})$ as $\nu \to 0$, we have $1 - \gamma - r(\nu) = \Theta(\nu^{2/3})$. Thus there exists a constant $d > 0$ such that $1 - \gamma - r(\nu) = d_2\,\nu^{2/3} + o(\nu^{2/3})$, where the sign $d > 0$ is required to ensure the radicand $2^{-\beta}(1 - \gamma - r(\nu)) - c_{\delta'}\nu$ (and hence the denominator) remains positive. Note that the value of $d$ is not universal: it is determined by the matching condition coming from the $\Theta(1)$ order balance (in particular, it depends on $-\frac{1}{2}\,(a_L - 1)(1 - \gamma)$). Equivalently, $r(\nu) = 1 - \gamma - d_2\,\nu^{2/3} + o(\nu^{2/3})$, as claimed.

Next, from

$$\frac{c_\delta \nu}{c\sqrt{a_0 x(\nu) - c_{\delta'}\nu}} = (1 - \gamma) - d_2\,\nu^{2/3} + o(\nu^{2/3}),$$

rearranging gives

$$\sqrt{a_0 x(\nu) - c_{\delta'}\nu} = \frac{c_\delta \nu}{c\left((1 - \gamma) - d_2\,\nu^{2/3} + o(\nu^{2/3})\right)}.$$

Using the expansion

$$\frac{1}{(1 - \gamma) - d_2\,\nu^{2/3} + o(\nu^{2/3})} = \frac{1}{1 - \gamma}\left(1 + \frac{d_2}{1 - \gamma}\nu^{2/3} + o(\nu^{2/3})\right),$$

we obtain

$$\sqrt{a_0 x(\nu) - c_{\delta'}\nu} = \frac{c_\delta}{c(1 - \gamma)}\nu + \frac{c_\delta d_2}{c(1 - \gamma)^2}\nu^{5/3} + o(\nu^{5/3}).$$

Squaring yields

$$a_0 x(\nu) - c_{\delta'}\nu = \left(\frac{c_\delta}{c(1-\gamma)}\right)^2 \nu^2 + O(\nu^{8/3}),$$

Therefore,

$$x(\nu) = \frac{c_{\delta'}}{a_0}\nu + \frac{1}{a_0}\left(\frac{c_\delta}{c(1-\gamma)}\right)^2 \nu^2 + O(\nu^{8/3}),$$

as claimed.

Finally, we analyze the choice of $x(\nu)$ for which

$$T_3(\nu) := \frac{c_\delta \nu}{c\sqrt{2^{-\beta}(1-\gamma) - c_{\delta'}\nu}} \cdot \frac{1}{1 - \frac{c_\delta \nu}{2c\left(2^{-\beta}\left(1-\gamma-r(\nu)\right) - c_{\delta'}\nu\right)^{3/2}} \cdot e^{-\beta/L}}$$

contributes a $\Theta(1)$ term. First, as $\nu \to 0$ we have

$$\frac{c_\delta \nu}{c\sqrt{2^{-\beta}(1-\gamma) - c_{\delta'}\nu}} = \frac{c_\delta}{c\sqrt{2^{-\beta}(1-\gamma)}}\nu + O(\nu^2).$$

Hence, in order for $T_3(\nu)$ to be $\Theta(1)$,

$$1 - \frac{c_\delta \nu}{2c\left(2^{-\beta}\left(1-\gamma-r(\nu)\right) - c_{\delta'}\nu\right)^{3/2}} \cdot e^{-\beta/L} = \Theta(\nu)$$

Equivalently, there exists a constant $d_3 > 0$ such that

$$\frac{c_\delta \nu}{2c\left(2^{-\beta}\left(1-\gamma-r(\nu)\right) - c_{\delta'}\nu\right)^{3/2}} = e^{\beta/L}\left(1 - d_3\nu + o(\nu)\right).$$

Invert the above display to obtain

$$\left(2^{-\beta}\left(1-\gamma-r(\nu)\right) - c_{\delta'}\nu\right)^{3/2} = \frac{c_\delta}{2c}e^{-\beta/L}\nu \cdot \frac{1}{1 - d_3'\nu + o(\nu)} = \frac{c_\delta}{2c}e^{-\beta/L}\nu\left(1 + d_3'\nu + o(\nu)\right),$$

i.e.,

$$2^{-\beta}\left(1-\gamma-r(\nu)\right) - c_{\delta'}\nu = \left(\frac{c_\delta}{2c}\right)^{2/3}e^{-2\beta/(3L)}\nu^{2/3}\left(1 + d_3'\nu + o(\nu)\right)^{2/3}.$$

Using the binomial expansion $(1+t)^{2/3} = 1 + \frac{2}{3}t + o(t)$ as $t \to 0$, we obtain

$$2^{-\beta}\left(1-\gamma-r(\nu)\right) - c_{\delta'}\nu = \left(\frac{c_\delta}{2c}\right)^{2/3}e^{-2\beta/(3L)}\nu^{2/3} + \frac{2}{3}\left(\frac{c_\delta}{2c}\right)^{2/3}e^{-2\beta/(3L)}d_3'\nu^{5/3} + o(\nu^{5/3}).$$

Rearranging yields

$$r(\nu) = 1 - \gamma - 2^\beta\left(\frac{c_\delta}{2c}\right)^{2/3}e^{-2\beta/(3L)}\nu^{2/3} + o(\nu^{2/3}).$$

Substituting this asymptotic form of $r(\nu)$ back into its definition $r(\nu) = \frac{c_\delta \nu}{c\sqrt{a_0 x(\nu) - c_{\delta'}\nu}}$ and repeating the same algebra as above yields

$$x(\nu) = \frac{c_{\delta'}}{a_0}\nu + \frac{1}{a_0}\left(\frac{c_\delta}{c(1-\gamma)}\right)^2 \nu^2 + O(\nu^{8/3}).$$

Moreover, it is straightforward to verify that if we substitute the $r(\nu)$ obtained from the $\Theta(1)$-balancing of $T_2(\nu)$ into $T_3(\nu)$, or conversely substitute the $r(\nu)$ obtained from the $\Theta(1)$-balancing of $T_3(\nu)$ into $T_2(\nu)$, then in both cases the resulting Puiseux expansion does not introduce any negative leading power of $\nu$. In particular, neither substitution violates the well-definedness regime.

Consequently, regardless of whether the $\Theta(1)$ constant term in $\mathcal{E}(\beta', \beta, \nu, x(\nu))$ is predominantly contributed by $T_2(\nu)$ or by $T_3(\nu)$, and regardless of the specific constant value on the right-hand side $-\frac{1}{2}(a_L - 1)(1 - \gamma)$, both scenarios yield the same expansion for the threshold $x(\nu)$ up to order $\nu^2$:

$$x(\nu) = \frac{c_{\delta'}}{a_0}\nu + \frac{1}{a_0}\left(\frac{c_\delta}{c(1 - \gamma)}\right)^2 \nu^2 + O(\nu^{8/3}).$$

The only difference lies in higher-order coefficients (beyond the $\nu^2$ term), which does not affect our conclusion. Which of $T_2(\nu)$ or $T_3(\nu)$ provides the dominant $\Theta(1)$ contribution depends on the finer $\Theta(1)$ matching (i.e., the constant-level balance) in the boundary condition $\mathcal{E}(\beta', \beta, \nu, x(\nu)) = -\frac{1}{2}(a_L - 1)(1 - \gamma)$, and hence on the specific interplay among $(\beta', \beta, L)$ and the constants $(c_\delta, c_{\delta'}, c, \gamma)$ through the corresponding $\Theta(1)$ coefficients. $\qquad\square$

**Lemma C.3.** *Under the notation of Proposition 5.5 and Lemma C.1, for fixed $(\beta', \beta)$, there exists a unique critical value $\nu_c > 0$ and a constant $C(\nu_c) > 0$ such that*

$$x'(\nu) = \frac{C(\nu_c)}{(\nu_c - \nu)^3}\left(1 + O(\nu_c - \nu)\right) \quad \text{as } \nu \uparrow \nu_c.$$

*Proof.* By the notation and proof of Lemma C.1, for fixed $(\beta', \beta)$, define $\mathcal{E}_\infty(\nu) := \lim_{x_0 \to \infty} \mathcal{E}(\beta', \beta, \nu, x_0)$.[5] Then,

$$\mathcal{E}_\infty(\nu) = \frac{c_\delta \nu}{c\sqrt{1 - \gamma - c_{\delta'}\nu}} \cdot \frac{1 - \left(\frac{c_\delta \nu}{2c(1 - \gamma - c_{\delta'}\nu)^{3/2}}\right)^{L-1}}{1 - \frac{c_\delta \nu}{2c(1 - \gamma - c_{\delta'}\nu)^{3/2}}}$$

$$- a_L \cdot \frac{c_\delta \nu}{c\sqrt{2^{-\beta}(1 - \gamma) - c_{\delta'}\nu}} \cdot \frac{1}{1 - \frac{c_\delta \nu}{2c(2^{-\beta}(1 - \gamma) - c_{\delta'}\nu)^{3/2}} \cdot e^{-\beta/L}}.$$

We further define $\mathcal{N}_\infty(\nu) := \lim_{x_0 \to \infty} \mathcal{N}(\beta', \beta, \nu, x_0)$.

We first prove $\mathcal{N}'_\infty(\nu) > 0$ and the existence of a unique $\nu_c > 0$ such that $\mathcal{N}_\infty(\nu_c) = 0$. It suffices to prove that $\mathcal{E}'_\infty(\nu) < 0$ and that there exists a unique $\nu_c > 0$ such that $\mathcal{E}_\infty(\nu_c) = -\frac{1}{2}(a_L - 1)(1 - \gamma)$.

In the proof of Lemma C.1, it is straightforward to verify that $\mathcal{E}(\beta', \beta, \nu, x_0)$ is monotonically decreasing in $\nu$ also holds when $x_0 \to \infty$, i.e., setting

$$r(\nu, x_0) := \frac{c_\delta \nu}{c\sqrt{a_0 x_0 - c_{\delta'}\nu}} = 0,$$

and the same monotonicity argument continues to hold. Hence $\mathcal{E}'_\infty(\nu) < 0$. Moreover, we clearly have $\mathcal{E}_\infty(0) = 0$. On the other hand, when $\nu$ is large enough so that

$$1 - \frac{c_\delta \nu}{2c(2^{-\beta}(1 - \gamma) - c_{\delta'}\nu)^{3/2}} \cdot e^{-\beta/L}$$

approaches $0$, the second term in $\mathcal{E}_\infty(\nu)$ diverges to $-\infty$, and thus $\mathcal{E}_\infty(\nu) \to -\infty$. Therefore, by continuity and strict monotonicity of $\mathcal{E}_\infty(\nu)$ in $\nu$, there exists a unique $\nu_c > 0$ such that $\mathcal{E}_\infty(\nu_c) = -\frac{1}{2}(a_L - 1)(1 - \gamma)$.

Next, define

$$T(\nu, x_0) := 1 - \frac{c_\delta \nu}{2c(2^{-\beta}(1 - \gamma - r(\nu, x_0)) - c_{\delta'}\nu)^{3/2}} \cdot e^{-\beta/L},$$

and

$$T(\nu, \infty) = 1 - \frac{c_\delta \nu}{2c(2^{-\beta}(1 - \gamma) - c_{\delta'}\nu)^{3/2}} \cdot e^{-\beta/L}.$$

Moreover, since

$$2^{-\beta}(1 - \gamma - r(\nu, x_0)) - c_{\delta'}\nu = \left(2^{-\beta}(1 - \gamma) - c_{\delta'}\nu\right) - 2^{-\beta}r(\nu, x_0),$$

---

[5]The limit $x_0 \to \infty$ is taken only for the auxiliary scalar argument in the analytic extension of $\mathcal{N}$ and $\mathcal{E}$. It does not assert that the original quantity $V_{p_0}(\hat\theta_0)$ can exceed its admissible range.

a Taylor expansion yields, as $x_0 \to \infty$,

$$T(\nu, \infty) - T(\nu, x_0) = \frac{3c_\delta \nu \, 2^{-\beta} e^{-\beta/L}}{4c \left(2^{-\beta}(1-\gamma) - c_{\delta'} \nu\right)^{5/2}} \, r(\nu, x_0) \; + \; O\big(r(\nu, x_0)^2\big),$$

Next, since $T(\nu, \infty) > 0$ in the well-definedness regime, we may expand the reciprocal around $T(\nu, \infty)$ as

$$\frac{1}{T(\nu, x_0)} - \frac{1}{T(\nu, \infty)} = \frac{T(\nu, \infty) - T(\nu, x_0)}{T(\nu, \infty)^2} \; + \; O\Big(\big(T(\nu, \infty) - T(\nu, x_0)\big)^2\Big), \qquad x_0 \to \infty.$$

Furthermore, since $T(\nu, \infty) - T(\nu, x_0) = O\big(r(\nu, x_0)\big)$,

$$\frac{c_\delta \nu}{c\sqrt{2^{-\beta}(1-\gamma) - c_{\delta'} \nu}} \left(\frac{1}{T(\nu, x_0)} - \frac{1}{T(\nu, \infty)}\right) = C_1(\nu) \, r(\nu, x_0) \; + \; O\big(r(\nu, x_0)^2\big), \qquad x_0 \to \infty,$$

where, after combining like terms, the coefficient $C_1(\nu)$ is given by

$$C_1(\nu) = \frac{3 \cdot 2^{-\beta} e^{-\beta/L} (c_\delta \nu)^2}{4} \, \frac{1}{c^2 \left(2^{-\beta}(1-\gamma) - c_{\delta'} \nu\right)^3 \left(1 - \frac{c_\delta \nu}{2c \left(2^{-\beta}(1-\gamma) - c_{\delta'} \nu\right)^{3/2}} e^{-\beta/L}\right)^2}.$$

Second, we can similarly obtain, by a Taylor expansion around $r(\nu, x_0) = 0$, that

$$\left(\frac{c_\delta \nu}{2c \left(2^{-\beta}(1 - \gamma - r(\nu, x_0)) - c_{\delta'} \nu\right)^{3/2}}\right)^{L-1} L^{-\beta} \cdot r(\nu, x_0)$$

$$= \left(\frac{c_\delta \nu}{2c \left(2^{-\beta}(1-\gamma) - c_{\delta'} \nu\right)^{3/2}}\right)^{L-1} L^{-\beta} \cdot r(\nu, x_0) \; + \; O\big(r(\nu, x_0)^2\big), \qquad x_0 \to \infty.$$

Therefore, if we define

$$C_2(\nu) := \left(\frac{c_\delta \nu}{2c \left(2^{-\beta}(1-\gamma) - c_{\delta'} \nu\right)^{3/2}}\right)^{L-1} L^{-\beta},$$

then combining the previous expansion with the definition of $\mathcal{E}_\infty(\nu)$ yields, as $x_0 \to \infty$,

$$\mathcal{E}_\infty(\nu) - \mathcal{E}(\beta', \beta, \nu, x_0) = a_L \Big(C_1(\nu) + C_2(\nu)\Big) r(\nu, x_0) \; + \; O\big(r(\nu, x_0)^2\big).$$

Moreover, by the boundary equation defining $x(\nu)$, $\mathcal{E}(\beta', \beta, \nu, x(\nu)) = -\frac{1}{2}(a_L - 1)(1 - \gamma)$. Therefore

$$\mathcal{E}_\infty(\nu) + \frac{1}{2}(a_L - 1)(1 - \gamma) = a_L \Big(C_1(\nu) + C_2(\nu)\Big) r\big(\nu, x(\nu)\big) \; + \; O\big(r\big(\nu, x(\nu)\big)^2\big).$$

We next justify that $x(\nu)$ goes to infinity as $\nu$ approaches $\nu_c$. Suppose, for contradiction, that $x(\nu)$ does not diverge as $\nu \uparrow \nu_c$. Then there exist a sequence $\nu_n \uparrow \nu_c$ and a constant $M < \infty$ such that $x(\nu_n) \leq M$ for all $n$. By passing to a subsequence, we may assume $x(\nu_n) \to x_\star$ for some $x_\star \in (0, M]$. By continuity of $\mathcal{E}(\beta', \beta, \nu, x_0)$ in $(\nu, x_0)$ within the well-definedness regime, and using the boundary equation $\mathcal{E}(\beta', \beta, \nu_n, x(\nu_n)) = -\frac{1}{2}(a_L - 1)(1 - \gamma)$, we obtain after taking $n \to \infty$ that

$$\mathcal{E}(\beta', \beta, \nu_c, x_\star) = -\frac{1}{2}(a_L - 1)(1 - \gamma).$$

However, since $x_\star < \infty$, the strict inequality above gives

$$\mathcal{E}(\beta', \beta, \nu_c, x_\star) \; < \; \mathcal{E}_\infty(\nu_c) \; = \; -\frac{1}{2}(a_L - 1)(1 - \gamma),$$

a contradiction. Therefore $x(\nu)$ must be unbounded as $\nu \uparrow \nu_c$. Finally, since Lemma C.1 shows that $x(\nu)$ is monotonically increasing in $\nu$, the only possibility is $x(\nu)$ goes to infinity as $\nu$ approaches $\nu_c$ as claimed.

We next start from the expansion obtained above:

$$\mathcal{E}_\infty(\nu) + \frac{1}{2}(a_L - 1)(1 - \gamma) = a_L\big(C_1(\nu) + C_2(\nu)\big)\, r\big(\nu, x(\nu)\big) + R(\nu), \qquad R(\nu) = O\Big(r\big(\nu, x(\nu)\big)^2\Big),$$

as $\nu \uparrow \nu_c$. Obviously $C_1(\nu) > 0$ and $C_2(\nu) > 0$, hence $a_L\big(C_1(\nu) + C_2(\nu)\big) > 0$. Then, by continuity, there exist $\varepsilon > 0$ and a constant $M', m > 0$ such that $a_L\big(C_1(\nu) + C_2(\nu)\big) \geq m > 0$ and $|R(\nu)| \leq M'\, r\big(\nu, x(\nu)\big)^2$ for all $\nu \in (\nu_c - \varepsilon, \nu_c)$. Since $x(\nu) \to \infty$ as $\nu \uparrow \nu_c$, we have $r\big(\nu, x(\nu)\big) \to 0$. Hence we may further assume that

$$M\, r\big(\nu, x(\nu)\big) \leq \frac{1}{2}\, m.$$

when $\nu \in (\nu_c - \varepsilon, \nu_c)$. Then,

$$\begin{aligned}
\mathcal{E}_\infty(\nu) + \frac{1}{2}(a_L - 1)(1 - \gamma) &\geq a_L\big(C_1(\nu) + C_2(\nu)\big)\, r\big(\nu, x(\nu)\big) - |R(\nu)| \\
&\geq m\, r\big(\nu, x(\nu)\big) - M\, r\big(\nu, x(\nu)\big)^2 \\
&\geq \frac{1}{2}\, m\, r\big(\nu, x(\nu)\big),
\end{aligned}$$

Consequently,

$$r\big(\nu, x(\nu)\big)^2 = O\left(\left(\mathcal{E}_\infty(\nu) + \frac{1}{2}(a_L - 1)(1 - \gamma)\right)^2\right).$$

Now,

$$r\big(\nu, x(\nu)\big) = \frac{\mathcal{E}_\infty(\nu) + \frac{1}{2}(a_L - 1)(1 - \gamma)}{a_L\big(C_1(\nu) + C_2(\nu)\big)} - \frac{R(\nu)}{a_L\big(C_1(\nu) + C_2(\nu)\big)}.$$

Therefore

$$\frac{R(\nu)}{a_L\big(C_1(\nu) + C_2(\nu)\big)} = O\Big(r\big(\nu, x(\nu)\big)^2\Big) = O\left(\left(\mathcal{E}_\infty(\nu) + \frac{1}{2}(a_L - 1)(1 - \gamma)\right)^2\right),$$

and

$$r\big(\nu, x(\nu)\big) = \frac{\mathcal{E}_\infty(\nu) + \frac{1}{2}(a_L - 1)(1 - \gamma)}{a_L\big(C_1(\nu) + C_2(\nu)\big)} + O\left(\left(\mathcal{E}_\infty(\nu) + \frac{1}{2}(a_L - 1)(1 - \gamma)\right)^2\right), \qquad \nu \uparrow \nu_c.$$

By differentiability of $\mathcal{E}_\infty$ at $\nu_c$ and the identity $\mathcal{E}_\infty(\nu_c) = -\frac{1}{2}(a_L - 1)(1 - \gamma)$, we have the first-order expansion

$$\mathcal{E}_\infty(\nu) + \frac{1}{2}(a_L - 1)(1 - \gamma) = \mathcal{E}'_\infty(\nu_c)\,(\nu - \nu_c) + o(\nu - \nu_c) = \big(-\mathcal{E}'_\infty(\nu_c)\big)\,(\nu_c - \nu) + o(\nu_c - \nu), \qquad \nu \uparrow \nu_c,$$

where $-\mathcal{E}'_\infty(\nu_c) > 0$. Hence, we obtain

$$r\big(\nu, x(\nu)\big) = \frac{-\mathcal{E}'_\infty(\nu_c)}{a_L\big(C_1(\nu) + C_2(\nu)\big)}\,(\nu_c - \nu) \; + \; O\big((\nu_c - \nu)^2\big) = \frac{-\mathcal{E}'_\infty(\nu_c)(\nu_c - \nu)}{a_L\big(C_1(\nu) + C_2(\nu)\big)}\,\Big(1 + O(\nu_c - \nu)\Big).$$

Now invert the definition

$$r\big(\nu, x(\nu)\big) = \frac{c_\delta \nu}{c\sqrt{a_0 x(\nu)} - c_{\delta'}\nu}$$

to get the identity

$$a_0 x(\nu) - c_{\delta'}\nu = \left(\frac{c_\delta \nu}{c\, r\big(\nu, x(\nu)\big)}\right)^2.$$

Since

$$\left(\frac{c_\delta \nu}{c\, r\big(\nu, x(\nu)\big)}\right)^2 = \left(\frac{c_\delta \nu\, a_L\big(C_1(\nu) + C_2(\nu)\big)}{c\,\big(-\mathcal{E}'_\infty(\nu_c)\big)}\right)^2 \frac{1}{(\nu_c - \nu)^2}\big(1 + O(\nu_c - \nu)\big), \qquad \nu \uparrow \nu_c,$$

we have,

$$x(\nu) = \frac{c_{\delta'}\nu}{a_0} + \frac{1}{a_0}\left(\frac{c_\delta \nu\, a_L\big(C_1(\nu) + C_2(\nu)\big)}{c\,\big(-\mathcal{E}'_\infty(\nu_c)\big)}\right)^2 \frac{1}{(\nu_c - \nu)^2}\big(1 + O(\nu_c - \nu)\big).$$

Finally, since $\nu = \nu_c + O(\nu_c - \nu)$ and $C_1(\nu) + C_2(\nu) = C_1(\nu_c) + C_2(\nu_c) + O(\nu_c - \nu)$ by continuity, we may replace $\nu$ by $\nu_c$ and $C_1(\nu) + C_2(\nu)$ by $C_1(\nu_c) + C_2(\nu_c)$ inside the prefactor at the cost of a multiplicative $(1 + O(\nu_c - \nu))$ factor, giving

$$x(\nu) = \frac{1}{a_0}\left(\frac{c_\delta \nu_c\, a_L\big(C_1(\nu_c) + C_2(\nu_c)\big)}{c\,\big(-\mathcal{E}'_\infty(\nu_c)\big)}\right)^2 \frac{1}{(\nu_c - \nu)^2}\big(1 + O(\nu_c - \nu)\big).$$

Differentiating the asymptotic expansion gives

$$x'(\nu) = \frac{C(\nu_c)}{(\nu_c - \nu)^3}\big(1 + O(\nu_c - \nu)\big),$$

where

$$C(\nu_c) = \frac{2}{a_0}\left(\frac{c_\delta \nu_c\, a_L\big(C_1(\nu_c) + C_2(\nu_c)\big)}{c\,\big(-\mathcal{E}'_\infty(\nu_c)\big)}\right)^2.$$

This completes the proof. $\qquad\square$

## C.4 Proof of Corollary 5.6

**Corollary C.4** (Formal version of Corollary 5.6). *In Proposition 5.5, fix any initialization $V_{p_0}(\hat\theta_0) \in (0, 1 - \gamma)$, and let $\nu^\star(\beta', \beta)$ be defined by the threshold equation $x\big(\beta', \beta, \nu^\star(\beta', \beta)\big) = V_{p_0}(\hat\theta_0)$. Then*

$$\nu^\star(\beta', \beta) \;=\; \sup\big\{\nu > 0 : \mathcal{N}\big(\beta', \beta, \nu, V_{p_0}(\hat\theta_0)\big) < 0\big\}.$$

*Moreover, (i) fixing $\beta'$, for $\beta > \beta'$, $\nu^\star(\beta', \beta)$ is decreasing in $\beta$; (ii) fixing $\beta$, for $\beta' \in (0, \beta)$, $\nu^\star(\beta', \beta)$ is increasing in $\beta'$; (iii) fixing $\Delta = \beta - \beta'$ and writing $\nu^\star(\beta', \beta)$ as $\nu^\star(\beta', \beta' + \Delta)$, when $\beta'$ is small, $\nu^\star(\beta', \beta' + \Delta)$ is increasing in $\beta'$ and*

$$\nu^\star(\beta', \beta' + \Delta) = \frac{c(1-\gamma)^{3/2}\log\big(\frac{L}{(L!)^{1/L}}\big)}{2c_\delta\big(2^{\Delta/2} - 1\big)}\,\beta' \;+\; o(\beta')$$

*as $\beta' \to 0$. Moreover, let $\nu_T > 0$ be the (unique) solution to*

$$\frac{c_\delta \nu}{c\sqrt{1-\gamma-c_{\delta'}\nu}}\cdot\frac{1 - \left(\frac{c_\delta \nu}{2c\,(1-\gamma-c_{\delta'}\nu)^{3/2}}\right)^{L-1}}{1 - \frac{c_\delta \nu}{2c\,(1-\gamma-c_{\delta'}\nu)^{3/2}}} = \frac{1-\gamma}{2}.$$

*Define*

$$h(\beta') := \frac{a_L(\beta')\big(a_L(\beta') - 1\big)}{a'_L(\beta')}.$$

*Then for any fixed $B > h^{-1}\big(\frac{2}{\log 2}\big) > 0$, there exists a constant $\nu_0(B) > 0$ such that whenever $\nu_T < \nu_0(B)$, we have $\nu^\star(\beta', \beta' + \Delta) < \nu_T$ for all $\beta' \in (0, B]$, and $\nu^\star(\beta', \beta' + \Delta)$ is guaranteed to be first increasing and then decreasing in $\beta'$ on $[0, B]$, with a unique optimizer on this interval. Moreover, for sufficiently large $\beta'$, the threshold $\nu^\star(\beta', \beta' + \Delta)$ satisfies $\nu^\star(\beta', \beta' + \Delta) = O(2^{-\beta'})$ and hence $\nu^\star(\beta', \beta' + \Delta) \to 0$ as $\beta' \to \infty$.*

*Proof.* By Proposition 5.5, for fixed $(\beta', \beta)$ we have $\mathcal{I}_{\mathcal{N}}(\beta', \beta, \nu) = \big(x(\beta', \beta, \nu), 1 - \gamma\big)$ and $x(\beta', \beta, \nu)$ is monotonically increasing in $\nu$. Therefore, for any $0 < \nu < \nu^{\star}(\beta', \beta)$ we have $x(\beta', \beta, \nu) < V_{p_0}(\hat{\theta}_0)$, so $V_{p_0}(\hat{\theta}_0) \in \mathcal{I}_{\mathcal{N}}(\beta', \beta, \nu)$, i.e., $\mathcal{N}\big(\beta', \beta, \nu, V_{p_0}(\hat{\theta}_0)\big) < 0$. Conversely, for any $\nu \geq \nu^{\star}(\beta', \beta)$ we have $x(\beta', \beta, \nu) \geq V_{p_0}(\hat{\theta}_0)$, so $V_{p_0}(\hat{\theta}_0) \notin \mathcal{I}_{\mathcal{N}}(\beta', \beta, \nu)$, i.e., $\mathcal{N}\big(\beta', \beta, \nu, V_{p_0}(\hat{\theta}_0)\big) \geq 0$. Hence,

$$\nu^{\star}(\beta', \beta) \;=\; \sup\Big\{ \nu > 0 : \; \mathcal{N}\big(\beta', \beta, \nu, V_{p_0}(\hat{\theta}_0)\big) < 0 \Big\}.$$

The remaining claims follow by directly combining Lemma C.5, Lemma C.6, Lemma C.7, and Lemma C.8. $\qquad\square$

**Lemma C.5.** *In Proposition 5.5, fix any initialization $V_{p_0}(\hat{\theta}_0) \in (0, 1 - \gamma)$, and define*

$$\nu^{\star}(\beta', \beta) \;=\; \sup\Big\{ \nu > 0 : \; \mathcal{N}\big(\beta', \beta, \nu, V_{p_0}(\hat{\theta}_0)\big) < 0 \Big\}.$$

*Then, fixing $\beta'$, for $\beta > \beta'$, $\nu^{\star}(\beta', \beta)$ is decreasing in $\beta$.*

*Proof.* Let $x_0 := V_{p_0}(\hat{\theta}_0)$ for convenience. By Theorem 5.2, the inequality $\mathcal{N}(\beta', \beta, \nu, x_0) < 0$ is equivalent to

$$\mathcal{E}(\beta', \beta, \nu, x_0) > -\frac{1}{2}\,(a_L - 1)(1 - \gamma).$$

Note that when $\beta' > 0$ is fixed, the right-hand side above is a fixed constant. Moreover, Lemma C.1 has already shown that $\mathcal{E}(\beta', \beta, \nu, x_0)$ is monotonically (strictly) decreasing in $\nu$. Therefore, to prove that $\nu^{\star}(\beta', \beta)$ is monotonically decreasing in $\beta$, it suffices to show that for each fixed $(\beta', \nu, x_0)$ in the well-definedness regime, $\mathcal{E}(\beta', \beta, \nu, x_0)$ is strictly decreasing as a function of $\beta$ (with $\beta > \beta'$). Indeed, define

$$\Phi(\beta', \beta, \nu) \;:=\; \mathcal{E}(\beta', \beta, \nu, x_0) + \frac{1}{2}\,(a_L - 1)(1 - \gamma).$$

Then $\Phi\big(\beta', \beta, \nu^{\star}(\beta', \beta)\big) = 0$. If $\mathcal{E}(\beta', \beta, \nu, x_0)$ is strictly decreasing in $\beta$, equivalently $\partial_{\beta}\Phi(\beta', \beta, \nu) < 0$, since $\partial_{\nu}\Phi(\beta', \beta, \nu) < 0$ and $\partial_{\nu}\Phi\big(\beta', \beta, \nu^{\star}(\beta', \beta)\big) \neq 0$, the implicit function theorem applies to the equation $\Phi(\beta', \beta, \nu) = 0$ around $\nu = \nu^{\star}(\beta', \beta)$ and yields

$$\frac{\partial}{\partial \beta}\nu^{\star}(\beta', \beta) \;=\; -\frac{\partial_{\beta}\Phi\big(\beta', \beta, \nu^{\star}(\beta', \beta)\big)}{\partial_{\nu}\Phi\big(\beta', \beta, \nu^{\star}(\beta', \beta)\big)} \;<\; 0.$$

Therefore, it suffices to prove that

$$\frac{c_{\delta}\nu}{c\sqrt{2^{-\beta}(1 - \gamma) - c_{\delta'}\nu}} \cdot \frac{1}{1 - \dfrac{c_{\delta}\nu}{2c\Big(2^{-\beta}\big(1 - \gamma - \frac{c_{\delta}\nu}{c\sqrt{a_0 x_0 - c_{\delta'}\nu}}\big) - c_{\delta'}\nu\Big)^{3/2}} \cdot e^{-\beta/L}}$$

and

$$\left(\frac{c_{\delta}\nu}{2c\Big(2^{-\beta}\big(1 - \gamma - \frac{c_{\delta}\nu}{c\sqrt{a_0 x_0 - c_{\delta'}\nu}}\big) - c_{\delta'}\nu\Big)^{3/2}}\right)^{L-1} L^{-\beta} \cdot \frac{c_{\delta}\nu}{c\sqrt{a_0 x_0 - c_{\delta'}\nu}}$$

are both strictly increasing as a function of $\beta$. For

$$\widetilde{U}(\beta) = \frac{c_{\delta}\nu}{c\sqrt{2^{-\beta}(1 - \gamma) - c_{\delta'}\nu}}.$$

Clearly, as $\beta$ increases, $2^{-\beta}$ decreases, and hence $\widetilde{U}(\beta)$ is strictly increasing in $\beta$. Next consider

$$\widetilde{V}(\beta) = \frac{1}{1 - \dfrac{c_{\delta}\nu}{2c\Big(2^{-\beta}\big(1 - \gamma - \frac{c_{\delta}\nu}{c\sqrt{a_0 x_0 - c_{\delta'}\nu}}\big) - c_{\delta'}\nu\Big)^{3/2}} \cdot e^{-\beta/L}}.$$

Differentiating gives

$$\frac{\mathrm{d}}{\mathrm{d}\beta} \log\left( \frac{c_\delta \nu}{2c\left(2^{-\beta}\left(1 - \gamma - \frac{c_\delta \nu}{c\sqrt{a_0 x_0 - c_{\delta'}\nu}}\right) - c_{\delta'}\nu\right)^{3/2}} \cdot e^{-\beta/L} \right)$$

$$= -\frac{3}{2} \cdot \frac{-\log 2 \cdot 2^{-\beta}\left(1 - \gamma - \frac{c_\delta \nu}{c\sqrt{a_0 x_0 - c_{\delta'}\nu}}\right)}{2^{-\beta}\left(1 - \gamma - \frac{c_\delta \nu}{c\sqrt{a_0 x_0 - c_{\delta'}\nu}}\right) - c_{\delta'}\nu} - \frac{1}{L}$$

$$= \frac{3}{2} \log 2 \cdot \frac{2^{-\beta}\left(1 - \gamma - \frac{c_\delta \nu}{c\sqrt{a_0 x_0 - c_{\delta'}\nu}}\right)}{2^{-\beta}\left(1 - \gamma - \frac{c_\delta \nu}{c\sqrt{a_0 x_0 - c_{\delta'}\nu}}\right) - c_{\delta'}\nu} - \frac{1}{L}$$

$$> \frac{3}{2} \log 2 - \frac{1}{L}.$$

Since $L \geq 2$, we have $\frac{3}{2}\log 2 - \frac{1}{L} > 0$, and hence the above derivative is strictly positive. Thus

$$\frac{c_\delta \nu}{2c\left(2^{-\beta}\left(1 - \gamma - \frac{c_\delta \nu}{c\sqrt{a_0 x_0 - c_{\delta'}\nu}}\right) - c_{\delta'}\nu\right)^{3/2}} \cdot e^{-\beta/L}$$

is strictly increasing in $\beta$, which implies that $\widetilde{V}(\beta)$ is strictly increasing in $\beta$. Combining with the monotonicity of $\widetilde{U}(\beta)$, we conclude that $\widetilde{U}(\beta)\widetilde{V}(\beta)$ is strictly increasing in $\beta$.

Next, consider

$$T_2(\beta) = \left( \frac{c_\delta \nu}{2c\left(2^{-\beta}\left(1 - \gamma - \frac{c_\delta \nu}{c\sqrt{a_0 x_0 - c_{\delta'}\nu}}\right) - c_{\delta'}\nu\right)^{3/2}} \right)^{L-1} L^{-\beta} \cdot \frac{c_\delta \nu}{c\sqrt{a_0 x_0 - c_{\delta'}\nu}}.$$

We compute the derivative of $\log T_2(\beta)$. Since the factor $\frac{c_\delta \nu}{c\sqrt{a_0 x_0 - c_{\delta'}\nu}}$ does not depend on $\beta$, we have

$$\frac{\mathrm{d}}{\mathrm{d}\beta} \log T_2(\beta) = (L-1) \cdot \frac{\mathrm{d}}{\mathrm{d}\beta} \log\left( \frac{c_\delta \nu}{2c\left(2^{-\beta}\left(1 - \gamma - \frac{c_\delta \nu}{c\sqrt{a_0 x_0 - c_{\delta'}\nu}}\right) - c_{\delta'}\nu\right)^{3/2}} \right) - \log L$$

$$= (L-1) \cdot \frac{3}{2}\log 2 \cdot \frac{2^{-\beta}\left(1 - \gamma - \frac{c_\delta \nu}{c\sqrt{a_0 x_0 - c_{\delta'}\nu}}\right)}{2^{-\beta}\left(1 - \gamma - \frac{c_\delta \nu}{c\sqrt{a_0 x_0 - c_{\delta'}\nu}}\right) - c_{\delta'}\nu} - \log L$$

$$> (L-1) \cdot \frac{3}{2}\log 2 - \log L.$$

In particular, since $L \geq 2$, we have $(L-1) \cdot \frac{3}{2}\log 2 - \log L > 0$, and therefore $T_2(\beta)$ is strictly increasing in $\beta$. Consequently, the $\beta$-dependent terms inside the bracket in $\mathcal{E}(\beta', \beta, \nu, x_0)$ are strictly increasing in $\beta$, and since the bracket is multiplied by the negative coefficient $-a_L$, it follows that $\mathcal{E}(\beta', \beta, \nu, x_0)$ is strictly decreasing in $\beta$. This completes the proof. $\square$

**Lemma C.6.** *In Proposition 5.5, fix any initialization $V_{p_0}(\hat{\theta}_0) \in (0, 1-\gamma)$, and define*

$$\nu^\star(\beta', \beta) = \sup\left\{ \nu > 0 : \mathcal{N}\left(\beta', \beta, \nu, V_{p_0}(\hat{\theta}_0)\right) < 0 \right\}.$$

*Then, fixing $\beta$, for $\beta' \in (0, \beta)$, $\nu^\star(\beta', \beta)$ is increasing in $\beta'$.*

*Proof.* Let $x_0 := V_{p_0}(\hat{\theta}_0)$ for convenience. Define

$$T_1(\nu) := \frac{c_\delta \nu}{c\sqrt{1 - \gamma - c_{\delta'}\nu}} \cdot \frac{1 - \left(\frac{c_\delta \nu}{2c\,(1-\gamma-c_{\delta'}\nu)^{3/2}}\right)^{L-1}}{1 - \frac{c_\delta \nu}{2c\,(1-\gamma-c_{\delta'}\nu)^{3/2}}},$$

$$T_2(\nu, \beta') := \left( \frac{c_\delta \nu}{2c\left(2^{-\beta}\left(1 - \gamma - \frac{c_\delta \nu}{c\sqrt{a_0(\beta')x_0 - c_{\delta'}\nu}}\right) - c_{\delta'}\nu\right)^{3/2}} \right)^{L-1} L^{-\beta} \cdot \frac{c_\delta \nu}{c\sqrt{a_0(\beta')x_0 - c_{\delta'}\nu}},$$

and

$$T_3(\nu, \beta') := \frac{c_\delta \nu}{c\sqrt{2^{-\beta}(1-\gamma) - c_{\delta'}\nu}} \cdot \frac{1}{1 - \frac{c_\delta \nu}{2c\left(2^{-\beta}\left(1-\gamma-\frac{c_\delta \nu}{c\sqrt{a_0(\beta')x_0 - c_{\delta'}\nu}}\right) - c_{\delta'}\nu\right)^{3/2}}} \cdot e^{-\beta/L}.$$

Then $\mathcal{N}(\beta', \beta, \nu, x_0) < 0$ is equivalent to

$$T_1(\nu) - a_L(\beta')\big(T_2(\nu, \beta') + T_3(\nu, \beta')\big) > -\frac{1}{2}(a_L(\beta') - 1)(1 - \gamma).$$

Equivalently,

$$T_1(\nu) - \frac{1-\gamma}{2} + a_L(\beta')\left(\frac{1-\gamma}{2} - T_2(\nu, \beta') - T_3(\nu, \beta')\right) > 0.$$

We first verify that both $a_0(\beta')$ and $a_L(\beta')$ are strictly increasing in $\beta'$. Recall the definitions

$$a_0(\beta') = \frac{L}{\sum_{i=1}^{L} i^{-\beta'}}, \qquad a_L(\beta') = \frac{\sum_{i=1}^{L} i^{-\beta'}}{L^{1-\beta'}}.$$

$a_0(\beta')$ is strictly increasing in $\beta'$ since

$$\frac{\mathrm{d}}{\mathrm{d}\beta'}\left(\sum_{i=1}^{L} i^{-\beta'}\right) < 0,$$

Next,

$$a_L(\beta') = \frac{1}{L}\sum_{i=1}^{L}\left(\frac{L}{i}\right)^{\beta'}.$$

Therefore $a_L(\beta')$ is strictly increasing in $\beta'$. Then, we conclude that both $T_2(\nu, \beta')$ and $T_3(\nu, \beta')$ are strictly decreasing in $\beta'$.

Suppose that for every $\beta' \in (0, \beta)$, we have

$$\frac{1-\gamma}{2} - T_2\big(\nu^\star(\beta', \beta), \beta'\big) - T_3\big(\nu^\star(\beta', \beta), \beta'\big) \geq 0.$$

Then for any $0 < \beta_1' < \beta_2' < \beta$, using that $a_L(\beta')$ is strictly increasing in $\beta'$ and that $T_2(\nu, \beta')$ and $T_3(\nu, \beta')$ are strictly decreasing in $\beta'$, we obtain

$$T_1\big(\nu^\star(\beta_1', \beta)\big) - \frac{1-\gamma}{2} + a_L(\beta_2')\left(\frac{1-\gamma}{2} - T_2\big((\nu^\star(\beta_1', \beta), \beta_2'\big) - T_3\big((\nu^\star(\beta_1', \beta), \beta_2'\big)\right)$$

$$> T_1\big(\nu^\star(\beta_1', \beta)\big) - \frac{1-\gamma}{2} + a_L(\beta_1')\left(\frac{1-\gamma}{2} - T_2\big((\nu^\star(\beta_1', \beta), \beta_1'\big) - T_3\big((\nu^\star(\beta_1', \beta), \beta_1'\big)\right) = 0.$$

Since Lemma C.1 shows that $\mathcal{E}(\beta', \beta, \nu, x_0)$ is strictly decreasing in $\nu$, it follows that $\nu^\star(\beta_2', \beta) > \nu^\star(\beta_1', \beta)$. Therefore, it remains to prove that for every $\beta' \in (0, \beta)$,

$$\frac{1-\gamma}{2} - T_2\big(\nu^\star(\beta', \beta), \beta'\big) - T_3\big(\nu^\star(\beta', \beta), \beta'\big) \geq 0.$$

Note that

$$T_1\big(\nu^\star(\beta', \beta)\big) - \frac{1-\gamma}{2} + a_L(\beta')\left(\frac{1-\gamma}{2} - T_2\big(\nu^\star(\beta', \beta), \beta'\big) - T_3\big(\nu^\star(\beta', \beta), \beta'\big)\right) = 0.$$

Since $a_L(\beta') > 0$, it suffices to prove that $T_1\big(\nu^\star(\beta', \beta)\big) - \frac{1-\gamma}{2} < 0$.

We next prove that $T_1(\nu)$ is strictly increasing in $\nu$ within the well-definedness regime. This is because $\frac{c_\delta \nu}{c\sqrt{1-\gamma-c_{\delta'}\nu}}$ is positive and clearly strictly increasing in $\nu$. Moreover,

$$\frac{1 - \left(\frac{c_\delta \nu}{2c\,(1-\gamma-c_{\delta'}\nu)^{3/2}}\right)^{L-1}}{1 - \frac{c_\delta \nu}{2c\,(1-\gamma-c_{\delta'}\nu)^{3/2}}} = \sum_{j=0}^{L-2}\left(\frac{c_\delta \nu}{2c\,(1-\gamma-c_{\delta'}\nu)^{3/2}}\right)^j,$$

where $\frac{c_\delta \nu}{2c\,(1-\gamma-c_{\delta'}\nu)^{3/2}}$ is positive and clearly strictly increasing in $\nu$. Therefore, let $\nu_T$ denote a (necessarily unique, by strict monotonicity) value satisfying $T_1(\nu_T) - \frac{1-\gamma}{2} = 0$. Then, since $T_1(\nu)$ is increasing in $\nu$, it suffices to prove that $0 < \nu^\star(\beta', \beta) < \nu_T$.

By the proof of Lemma C.1, we have $T_3(\nu, \beta') > T_1(\nu)$. Hence $T_3(\nu, \beta') + T_2(\nu, \beta') > T_1(\nu)$, and therefore,

$$T_1\big(\nu_T\big) - \frac{1-\gamma}{2} + a_L(\beta')\left(\frac{1-\gamma}{2} - T_2\big(\nu_T, \beta'\big) - T_3\big(\nu_T, \beta'\big)\right)$$

$$= a_L(\beta')\left(\frac{1-\gamma}{2} - T_2\big(\nu_T, \beta'\big) - T_3\big(\nu_T, \beta'\big)\right)$$

$$< a_L(\beta')\left(\frac{1-\gamma}{2} - T_1(\nu_T)\right) = 0.$$

On the other hand, since $T_1(0) = T_2(0, \beta') = T_3(0, \beta') = 0$, we have

$$T_1(0) - \frac{1-\gamma}{2} + a_L(\beta')\left(\frac{1-\gamma}{2} - T_2(0, \beta') - T_3(0, \beta')\right) = \big(a_L(\beta') - 1\big)\frac{1-\gamma}{2} > 0.$$

Finally, since

$$T_1(\nu) - \frac{1-\gamma}{2} + a_L(\beta')\left(\frac{1-\gamma}{2} - T_2(\nu, \beta') - T_3(\nu, \beta')\right)$$

is strictly decreasing in $\nu$ as well (for each fixed $\beta'$), we must have $0 < \nu^\star(\beta', \beta) < \nu_T$, as desired. $\qquad\square$

**Lemma C.7.** *In Proposition 5.5, fix any initialization $V_{p_0}(\hat\theta_0) \in (0, 1-\gamma)$, and define*

$$\nu^\star(\beta', \beta) \;=\; \sup\left\{\nu > 0 : \; \mathcal{N}\big(\beta', \beta, \nu, V_{p_0}(\hat\theta_0)\big) < 0\right\}.$$

*Fix $\Delta = \beta - \beta' > 0$ and write $\nu^\star(\beta', \beta) = \nu^\star(\beta', \beta' + \Delta)$. When $\beta'$ is small, $\nu^\star(\beta', \beta' + \Delta)$ is increasing in $\beta'$ and*

$$\nu^\star(\beta', \beta' + \Delta) = \frac{c(1-\gamma)^{3/2}\log\left(\frac{L}{(L!)^{1/L}}\right)}{2c_\delta\big(2^{\Delta/2} - 1\big)}\,\beta' \;+\; o(\beta') \qquad \text{as } \beta' \to 0.$$

*Proof.* Let $x_0 := V_{p_0}(\hat\theta_0)$ and $\nu_\Delta^\star(\beta') := \nu^\star(\beta', \beta' + \Delta)$ convenience. Following the notation of Theorem 5.2, we write

$$\mathcal{N}(\beta', \beta' + \Delta, \nu, x_0) = -\mathcal{E}(\beta', \beta' + \Delta, \nu, x_0) - \frac{1}{2}\big(a_L(\beta') - 1\big)(1-\gamma).$$

Moreover, $\mathcal{N}(\beta', \beta' + \Delta, \nu, x_0)$ is increasing in $\nu$, and for every $\beta' > 0$ we have $\mathcal{N}(\beta', \beta' + \Delta, 0, x_0) < 0$. Furthermore, $\nu_\Delta^\star(\beta')$ satisfies

$$\mathcal{N}\big(\beta', \beta' + \Delta, \nu_\Delta^\star(\beta'), x_0\big) = 0,$$

and note that when $\beta' = 0$, the corresponding threshold satisfies $\nu_\Delta^\star(0) = 0$.

We first compute the partial derivatives of $\mathcal{N}(\beta', \beta' + \Delta, \nu, x_0)$ with respect to $\nu$ and $\beta'$ at $(\beta', \nu) = (0, 0)$, namely $\partial_\nu \mathcal{N}(0, \Delta, 0, x_0)$ and $\partial_{\beta'} \mathcal{N}(0, \Delta, 0, x_0)$. For $\partial_\nu \mathcal{N}(0, \Delta, 0, x_0) = -\partial_\nu \mathcal{E}(0, \Delta, 0, x_0)$, note that the derivative of $\mathcal{E}(\beta', \beta' + \Delta, \nu, x_0)$ at $\nu = 0$ only depends on its linear term in $\nu$. Since

$$\frac{c_\delta \nu}{c\sqrt{1-\gamma-c_{\delta'}\nu}} \cdot \frac{1 - \left(\frac{c_\delta \nu}{2c\,(1-\gamma-c_{\delta'}\nu)^{3/2}}\right)^{L-1}}{1 - \frac{c_\delta \nu}{2c\,(1-\gamma-c_{\delta'}\nu)^{3/2}}} = \frac{c_\delta}{c\sqrt{1-\gamma}}\,\nu + o(\nu),$$

$$\frac{c_\delta \nu}{c\sqrt{2^{-\beta}(1-\gamma)} - c_{\delta'}\nu} \cdot \frac{1}{1 - \frac{c_\delta \nu}{2c\left(2^{-\beta}\left(1-\gamma-\frac{c_\delta \nu}{c\sqrt{a_0 x_0 - c_{\delta'}\nu}}\right) - c_{\delta'}\nu\right)^{3/2}} \cdot e^{-\beta/L}} = 2^{\Delta/2}\frac{c_\delta}{c\sqrt{1-\gamma}}\nu + o(\nu),$$

and

$$\left(\frac{c_\delta \nu}{2c\left(2^{-\beta}\left(1-\gamma-\frac{c_\delta \nu}{c\sqrt{a_0 x_0 - c_{\delta'}\nu}}\right) - c_{\delta'}\nu\right)^{3/2}}\right)^{L-1} L^{-\beta} \cdot \frac{c_\delta \nu}{c\sqrt{a_0 x_0 - c_{\delta'}\nu}} = O(\nu^L),$$

we obtain

$$\partial_\nu \mathcal{N}(0, \Delta, 0, x_0) = -\partial_\nu \mathcal{E}(0, \Delta, 0, x_0) = \left(2^{\Delta/2} - 1\right)\frac{c_\delta}{c\sqrt{1-\gamma}} > 0.$$

Next we compute $\partial_{\beta'}\mathcal{N}(0, \Delta, 0, x_0)$. Since $\mathcal{E}(\beta', \beta' + \Delta, \nu, x_0) \equiv 0$ at $\nu = 0$, we have $\partial_{\beta'}\mathcal{N}(0, \Delta, 0, x_0) = -\frac{1}{2}a_L'(0)(1 - \gamma)$. Recall that $a_L(\beta') = \frac{\sum_{i=1}^L i^{-\beta'}}{L^{1-\beta'}}$. Hence

$$\log a_L(\beta') = \log\left(\sum_{i=1}^L i^{-\beta'}\right) - \log L + \beta' \log L,$$

and

$$\frac{\mathrm{d}}{\mathrm{d}\beta'}\log a_L(\beta')\bigg|_{\beta'=0} = \frac{-\sum_{i=1}^L \log i}{L} + \log L = \log\left(\frac{L}{(L!)^{1/L}}\right).$$

Since $a_L(0) = \frac{\sum_{i=1}^L 1}{L} = 1$, it follows that

$$a_L'(0) = a_L(0) \cdot \frac{\mathrm{d}}{\mathrm{d}\beta'}\log a_L(\beta')\bigg|_{\beta'=0} = \log\left(\frac{L}{(L!)^{1/L}}\right) > 0.$$

Therefore,

$$\partial_{\beta'}\mathcal{N}(0, \Delta, 0, x_0) = -\frac{1}{2}a_L'(0)(1 - \gamma) = -\frac{1}{2}\log\left(\frac{L}{(L!)^{1/L}}\right)(1 - \gamma) < 0.$$

Finally, since $\nu_\Delta^\star(\beta')$ is defined implicitly by $\mathcal{N}(\beta', \beta' + \Delta, \nu_\Delta^\star(\beta'), x_0) = 0$, the implicit function theorem yields

$$\frac{\mathrm{d}}{\mathrm{d}\beta'}\nu_\Delta^\star(0) = -\frac{\partial_{\beta'}\mathcal{N}(0, \Delta, 0, x_0)}{\partial_\nu \mathcal{N}(0, \Delta, 0, x_0)} = \frac{c(1-\gamma)^{3/2}\log\left(\frac{L}{(L!)^{1/L}}\right)}{2c_\delta\left(2^{\Delta/2} - 1\right)} > 0.$$

Therefore, as $\beta' \to 0$,

$$\nu_\Delta^\star(\beta') = \nu_\Delta^\star(0) + \frac{\mathrm{d}}{\mathrm{d}\beta'}\nu_\Delta^\star(0)\,\beta' + o(\beta') = \frac{c(1-\gamma)^{3/2}\log\left(\frac{L}{(L!)^{1/L}}\right)}{2c_\delta\left(2^{\Delta/2} - 1\right)}\beta' + o(\beta').$$

This completes the proof. □

**Lemma C.8.** *In Proposition 5.5, fix any initialization* $V_{p_0}(\hat\theta_0) \in (0, 1 - \gamma)$*, and define*

$$\nu^\star(\beta', \beta) = \sup\left\{\nu > 0 : \mathcal{N}(\beta', \beta, \nu, V_{p_0}(\hat\theta_0)) < 0\right\}.$$

*Fix* $\Delta = \beta - \beta' > 0$ *and write* $\nu^\star(\beta', \beta) = \nu^\star(\beta', \beta' + \Delta)$*. Let* $\nu_T > 0$ *be the solution to*

$$\frac{c_\delta \nu}{c\sqrt{1-\gamma} - c_{\delta'}\nu} \cdot \frac{1 - \left(\frac{c_\delta \nu}{2c(1-\gamma-c_{\delta'}\nu)^{3/2}}\right)^{L-1}}{1 - \frac{c_\delta \nu}{2c(1-\gamma-c_{\delta'}\nu)^{3/2}}} = \frac{1-\gamma}{2}.$$

*Define*

$$h(\beta') := \frac{a_L(\beta')\big(a_L(\beta') - 1\big)}{a_L'(\beta')}.$$

*Then for any fixed $B > h^{-1}\big(\frac{2}{\log 2}\big) > 0$, there exists a constant $\nu_0(B) > 0$ such that whenever $\nu_T < \nu_0(B)$, we have $\nu^\star(\beta', \beta' + \Delta) < \nu_T$ for all $\beta' \in (0, B]$, and $\nu^\star(\beta', \beta' + \Delta)$ is guaranteed to be first increasing and then decreasing in $\beta'$ on $[0, B]$, with a unique optimizer on this interval. Moreover, for sufficiently large $\beta'$, the threshold $\nu^\star(\beta', \beta' + \Delta)$ satisfies $\nu^\star(\beta', \beta' + \Delta) = O(2^{-\beta'})$ and hence $\nu^\star(\beta', \beta' + \Delta) \to 0$ as $\beta' \to \infty$.*

*Proof.* Let $x_0 := V_{p_0}(\hat\theta_0)$, $\nu_\Delta^\star(\beta') := \nu^\star(\beta', \beta' + \Delta)$, and write $\mathcal{N}(\beta', \nu) := \mathcal{N}(\beta', \beta' + \Delta, \nu, x_0)$ for brevity.

First, by the proof of Lemma C.6, we directly obtain $0 < \nu_\Delta^\star(\beta') < \nu_T < \nu_0(B)$, where $\nu_0(B)$ will be chosen later. On any interval $\beta' \in [0, B]$, when $\nu_T$ is sufficiently small, all radicands appearing in $\mathcal{N}$ remain strictly positive. Thus, for $\beta' \in [0, B]$ and $0 \leq \nu \leq \nu_T$, we have the expansion

$$\mathcal{N}(\beta', \nu) = -\frac{1}{2}\big(a_L(\beta') - 1\big)(1 - \gamma) + \frac{c_\delta}{c\sqrt{1 - \gamma}}\Big(a_L(\beta')\, 2^{(\beta' + \Delta)/2} - 1\Big)\nu + R_B(\beta', \nu),$$

where there exist constants $C_0(B), C_1(B), C_2(B) > 0$ such that

$$|R_B(\beta', \nu)| \leq C_0(B)\nu^2, \qquad \big|\partial_\nu R_B(\beta', \nu)\big| \leq C_1(B)\nu, \qquad \big|\partial_{\beta'} R_B(\beta', \nu)\big| \leq C_2(B)\nu^2.$$

Moreover, note that for fixed $\Delta > 0$, we always have

$$\frac{c_\delta}{c\sqrt{1 - \gamma}}\Big(a_L(\beta')\, 2^{(\beta' + \Delta)/2} - 1\Big) > 0,$$

and on $[0, B]$ it is bounded away from zero.

Define the linear approximation of $\mathcal{N}(\beta', \nu)$ by dropping the remainder term:

$$\mathcal{N}_\ell(\beta', \nu) = -\frac{1}{2}\big(a_L(\beta') - 1\big)(1 - \gamma) + \frac{c_\delta}{c\sqrt{1 - \gamma}}\Big(a_L(\beta')\, 2^{(\beta' + \Delta)/2} - 1\Big)\nu.$$

Then the linearized root is given by

$$\nu_\Delta^\ell(\beta') := \frac{c(1 - \gamma)^{3/2}\big(a_L(\beta') - 1\big)}{2c_\delta\Big(a_L(\beta')\, 2^{(\beta' + \Delta)/2} - 1\Big)}.$$

Note that the behavior of $\nu_\Delta^\ell(\beta')$ as $\beta' \to 0$ is consistent with Lemma C.7. We now verify this. Recall that $a_L(\beta') = \frac{\sum_{i=1}^L i^{-\beta'}}{L^{1-\beta'}}$. With explicit expansions we have

$$a_L(\beta') = 1 + \log\Big(\frac{L}{(L!)^{1/L}}\Big)\beta' + o(\beta').$$

Next, since

$$2^{(\beta' + \Delta)/2} = 2^{\Delta/2}\Big(1 + \frac{\log 2}{2}\beta' + o(\beta')\Big),$$

we obtain

$$a_L(\beta')\, 2^{(\beta' + \Delta)/2} - 1 = \big(2^{\Delta/2} - 1\big) + 2^{\Delta/2}\Big(\log\Big(\frac{L}{(L!)^{1/L}}\Big) + \frac{\log 2}{2}\Big)\beta' + o(\beta').$$

Therefore, as $\beta' \to 0$,

$$\nu_\Delta^\ell(\beta') = \frac{c(1 - \gamma)^{3/2}\log\Big(\frac{L}{(L!)^{1/L}}\Big)}{2c_\delta\big(2^{\Delta/2} - 1\big)}\beta' + o(\beta'),$$

which is consistent with Lemma C.7. Therefore, when $\beta'$ is sufficiently small, the above expansion implies that $\nu_\Delta^\ell(\beta')$ is strictly increasing in $\beta'$.

We further characterize the shape of this auxiliary root. Direct differentiation shows that $\frac{\mathrm{d}}{\mathrm{d}\beta'}\nu_\Delta^\ell(\beta') = 0$ is equivalent to

$$\frac{a_L(\beta')\big(a_L(\beta') - 1\big)}{a_L'(\beta')} = \frac{2^{(\beta'+\Delta)/2} - 1}{\frac{\mathrm{d}}{\mathrm{d}\beta'}2^{(\beta'+\Delta)/2}}.$$

Let the right-hand side be denoted by $g(\beta')$. Then

$$g(\beta') := \frac{2^{(\beta'+\Delta)/2} - 1}{\frac{\mathrm{d}}{\mathrm{d}\beta'}2^{(\beta'+\Delta)/2}} = \frac{2}{\log 2}\left(1 - 2^{-(\beta'+\Delta)/2}\right).$$

This is strictly increasing in $\beta'$, and satisfies

$$g(0) = \frac{2}{\log 2}\left(1 - 2^{-\Delta/2}\right) > 0, \qquad \lim_{\beta'\to\infty} g(\beta') = \frac{2}{\log 2}.$$

Let the left-hand side be denoted by $h(\beta')$, namely

$$h(\beta') := \frac{a_L(\beta')\big(a_L(\beta') - 1\big)}{a_L'(\beta')}.$$

Lemma C.9 shows that $h(\beta')$ is also strictly increasing in $\beta'$. Moreover,

$$\lim_{\beta'\to 0} h(\beta') = 0, \qquad \lim_{\beta'\to\infty} h(\beta') = +\infty.$$

Thus $h^{-1}$ is well-defined on $(0,\infty)$. By Lemma C.10, $h$ is strictly convex in $\beta'$, while $g''(\beta') = -\frac{\log 2}{2}2^{-(\beta'+\Delta)/2} < 0$. Hence, $h(\beta') - g(\beta')$ is strictly convex. Since

$$h(0) - g(0) < 0, \qquad \frac{\mathrm{d}}{\mathrm{d}\beta'}\big(h(\beta') - g(\beta')\big)\Big|_{\beta'=0} = 1 - 2^{-\Delta/2} > 0,$$

and $\lim_{\beta'\to\infty}(h(\beta') - g(\beta')) = +\infty$, the equation $h(\beta') = g(\beta')$ admits a unique solution, denoted by $\bar{\beta}_s'$. Equivalently, $\frac{\mathrm{d}}{\mathrm{d}\beta'}\nu_\Delta^\ell(\bar{\beta}_s') = 0$. Furthermore, since $h(\bar{\beta}_s') = g(\bar{\beta}_s')$ and $g(\bar{\beta}_s') = \frac{2}{\log 2}\big(1 - 2^{-(\bar{\beta}_s'+\Delta)/2}\big) < \frac{2}{\log 2}$, the strict monotonicity of $h$ gives $\bar{\beta}_s' < h^{-1}\left(\frac{2}{\log 2}\right)$.

Choose $B > h^{-1}\left(\frac{2}{\log 2}\right)$, we have $\bar{\beta}_s' \in (0, B)$. Thus $\nu_\Delta^\ell(\beta')$ is first increasing and then decreasing on $[0, B]$, with a unique optimizer on this interval. We now show that the true threshold $\nu_\Delta^\star(\beta')$ inherits the same monotonicity pattern on $[0, B]$ when $\nu_T$ is sufficiently small.

On the one hand, since

$$\partial_\nu \mathcal{N}(\beta', \nu) = \frac{c_\delta}{c\sqrt{1-\gamma}}\left(a_L(\beta')\,2^{(\beta'+\Delta)/2} - 1\right) + \partial_\nu R_B(\beta', \nu),$$

$\big|\partial_\nu R_B(\beta', \nu)\big| \le C_1(B)\nu \le C_1(B)\nu_T$, and $\frac{c_\delta}{c\sqrt{1-\gamma}}\left(a_L(\beta')\,2^{(\beta'+\Delta)/2} - 1\right)$ admits a strictly positive lower bound on $[0, B]$, it follows that

$$\partial_\nu \mathcal{N}(\beta', \nu) = \frac{c_\delta}{c\sqrt{1-\gamma}}\left(a_L(\beta')\,2^{(\beta'+\Delta)/2} - 1\right)\left(1 + O_B(\nu_T)\right).$$

On the other hand, $\partial_{\beta'}\mathcal{N}(\beta', \nu) = \partial_{\beta'}\mathcal{N}_\ell(\beta', \nu) + \partial_{\beta'}R_B(\beta', \nu)$, so plugging in $\nu = \nu_\Delta^\star(\beta')$ gives

$$\partial_{\beta'}\mathcal{N}\big(\beta', \nu_\Delta^\star(\beta')\big) = \partial_{\beta'}\mathcal{N}_\ell\big(\beta', \nu_\Delta^\star(\beta')\big) + \partial_{\beta'}R_B\big(\beta', \nu_\Delta^\star(\beta')\big).$$

Moreover, since

$$\partial_{\beta'}\mathcal{N}_\ell(\beta', \nu) = -\frac{1}{2}a_L'(\beta')(1 - \gamma) + \frac{\mathrm{d}}{\mathrm{d}\beta'}\left[\frac{c_\delta}{c\sqrt{1-\gamma}}\left(a_L(\beta')\,2^{(\beta'+\Delta)/2} - 1\right)\right]\nu,$$

we have

$$\partial_{\beta'}\mathcal{N}_\ell\big(\beta', \nu_\Delta^\star(\beta')\big) = \partial_{\beta'}\mathcal{N}_\ell\big(\beta', \nu_\Delta^\ell(\beta')\big) + \frac{\mathrm{d}}{\mathrm{d}\beta'}\left[\frac{c_\delta}{c\sqrt{1-\gamma}}\Big(a_L(\beta')\,2^{(\beta'+\Delta)/2} - 1\Big)\right]\Big(\nu_\Delta^\star(\beta') - \nu_\Delta^\ell(\beta')\Big).$$

Hence

$$\partial_{\beta'}\mathcal{N}\big(\beta', \nu_\Delta^\star(\beta')\big) = \partial_{\beta'}\mathcal{N}_\ell\big(\beta', \nu_\Delta^\ell(\beta')\big)$$
$$+ \frac{\mathrm{d}}{\mathrm{d}\beta'}\left[\frac{c_\delta}{c\sqrt{1-\gamma}}\Big(a_L(\beta')\,2^{(\beta'+\Delta)/2} - 1\Big)\right]\Big(\nu_\Delta^\star(\beta') - \nu_\Delta^\ell(\beta')\Big)$$
$$+ \partial_{\beta'}R_B\big(\beta', \nu_\Delta^\star(\beta')\big).$$

For the remainder derivative, we use the bound $\big|\partial_{\beta'}R_B(\beta', \nu)\big| \le C_2(B)\nu^2 \le C_2(B)\nu_T^2$. Next, to control $\nu_\Delta^\star(\beta') - \nu_\Delta^\ell(\beta')$, note that $\nu_\Delta^\star(\beta')$ satisfies

$$-\frac{1}{2}\big(a_L(\beta') - 1\big)(1 - \gamma) + \frac{c_\delta}{c\sqrt{1-\gamma}}\Big(a_L(\beta')\,2^{(\beta'+\Delta)/2} - 1\Big)\nu_\Delta^\star(\beta') + R_B\big(\beta', \nu_\Delta^\star(\beta')\big) = 0,$$

while $\nu_\Delta^\ell(\beta')$ satisfies

$$-\frac{1}{2}\big(a_L(\beta') - 1\big)(1 - \gamma) + \frac{c_\delta}{c\sqrt{1-\gamma}}\Big(a_L(\beta')\,2^{(\beta'+\Delta)/2} - 1\Big)\nu_\Delta^\ell(\beta') = 0.$$

Subtracting the two equations yields

$$\nu_\Delta^\star(\beta') - \nu_\Delta^\ell(\beta') = -\frac{R_B\big(\beta', \nu_\Delta^\star(\beta')\big)}{\frac{c_\delta}{c\sqrt{1-\gamma}}\Big(a_L(\beta')\,2^{(\beta'+\Delta)/2} - 1\Big)}.$$

Since $|R_B(\beta', \nu)| \le C_0(B)\nu^2 \le C_0(B)\nu_T^2$, and the denominator admits a strictly positive lower bound on $[0, B]$, we conclude that $\nu_\Delta^\star(\beta') - \nu_\Delta^\ell(\beta') = O_B(\nu_T^2)$. Therefore, because the derivative $\frac{\mathrm{d}}{\mathrm{d}\beta'}\left[\frac{c_\delta}{c\sqrt{1-\gamma}}\Big(a_L(\beta')\,2^{(\beta'+\Delta)/2} - 1\Big)\right]$ is bounded on $[0, B]$, we obtain

$$\partial_{\beta'}\mathcal{N}\big(\beta', \nu_\Delta^\star(\beta')\big) = \partial_{\beta'}\mathcal{N}_\ell\big(\beta', \nu_\Delta^\ell(\beta')\big) + O_B(\nu_T^2),$$

uniformly for $\beta' \in [0, B]$.

Then, by the implicit function theorem,

$$\frac{\mathrm{d}}{\mathrm{d}\beta'}\nu_\Delta^\star(\beta') = -\frac{\partial_{\beta'}\mathcal{N}\big(\beta', \nu_\Delta^\star(\beta')\big)}{\partial_\nu\mathcal{N}\big(\beta', \nu_\Delta^\star(\beta')\big)} = -\frac{\partial_{\beta'}\mathcal{N}_\ell\big(\beta', \nu_\Delta^\ell(\beta')\big) + O_B(\nu_T^2)}{\frac{c_\delta}{c\sqrt{1-\gamma}}\Big(a_L(\beta')\,2^{(\beta'+\Delta)/2} - 1\Big)\big(1 + O_B(\nu_T)\big)}.$$

Equivalently,

$$\frac{\mathrm{d}}{\mathrm{d}\beta'}\nu_\Delta^\star(\beta') = -\frac{\partial_{\beta'}\mathcal{N}_\ell\big(\beta', \nu_\Delta^\ell(\beta')\big)}{\frac{c_\delta}{c\sqrt{1-\gamma}}\Big(a_L(\beta')\,2^{(\beta'+\Delta)/2} - 1\Big)} + O_B(\nu_T).$$

Therefore,

$$\frac{\mathrm{d}}{\mathrm{d}\beta'}\nu_\Delta^\star(\beta') = \frac{\mathrm{d}}{\mathrm{d}\beta'}\nu_\Delta^\ell(\beta') + O_B(\nu_T),$$

uniformly on $[0, B]$. Since $\nu_\Delta^\ell(\beta')$ is first increasing and then decreasing on $[0, B]$, with a unique optimizer on this interval, the above uniform perturbation implies that, by taking $\nu_0(B) > 0$ sufficiently small, whenever $\nu_T < \nu_0(B)$, the map $\beta' \mapsto \nu_\Delta^\star(\beta')$ is also first increasing and then decreasing on $[0, B]$, with a unique optimizer on this interval. This proves the first part of the lemma.

It remains to prove the large-$\beta'$ behavior. By the standing well-definedness requirement of $\mathcal{N}(\beta', \beta' + \Delta, \nu, x_0)$, every radicand appearing in the denominators must remain strictly positive. In particular, since $\beta = \beta' + \Delta$, we must have $2^{-(\beta'+\Delta)}(1 - \gamma) - c_{\delta'}\,\nu_\Delta^\star(\beta') > 0$. Therefore,

$$\nu_\Delta^\star(\beta') < \frac{1-\gamma}{c_{\delta'}}2^{-(\beta'+\Delta)} = O(2^{-\beta'}).$$

It follows immediately that $\nu_\Delta^\star(\beta') \to 0$ as $\beta' \to \infty$. This completes the proof. $\qquad\square$

**Lemma C.9.** *Let $L \geq 2$ be an integer and define $a_L(\beta') := \frac{1}{L} \sum_{i=1}^{L} \left(\frac{L}{i}\right)^{\beta'}$. Then the mapping*

$$\beta' \longmapsto \frac{a_L(\beta')\big(a_L(\beta') - 1\big)}{a_L'(\beta')}, \qquad \beta' > 0,$$

*is strictly increasing on $(0, \infty)$. Moreover, it satisfies*

$$\lim_{\beta' \to 0} \frac{a_L(\beta')\big(a_L(\beta') - 1\big)}{a_L'(\beta')} = 0, \qquad \lim_{\beta' \to \infty} \frac{a_L(\beta')\big(a_L(\beta') - 1\big)}{a_L'(\beta')} = +\infty.$$

*Proof.* Define

$$g(\beta') := \frac{a_L(\beta')\big(a_L(\beta') - 1\big)}{a_L'(\beta')}.$$

We first prove that $g(\beta')$ is increasing on $(0, \infty)$. For any $i \in [L]$, define $x_i := \log\left(\frac{L}{i}\right)$. Then $x_i \in [0, \log L]$, and we can rewrite

$$a_L(\beta') = \frac{1}{L} \sum_{i=1}^{L} e^{\beta' x_i}.$$

We define a random variable $X$ supported on $\{x_1, x_2, \ldots, x_L\}$ with probability mass function

$$p_i := \frac{e^{\beta' x_i}}{\sum_{j=1}^{L} e^{\beta' x_j}} = \frac{e^{\beta' x_i}}{L\, a_L(\beta')}, \qquad i = 1, \ldots, L.$$

Note that this notation $p_i$ is unrelated to the "question distribution" used elsewhere in the paper; we reuse the symbol $p$ here only to follow standard convention. With this definition, we have

$$a_L'(\beta') = \frac{1}{L} \sum_{i=1}^{L} x_i e^{\beta' x_i} = a_L(\beta')\, \mathbb{E}[X], \qquad a_L''(\beta') = \frac{1}{L} \sum_{i=1}^{L} x_i^2 e^{\beta' x_i} = a_L(\beta')\, \mathbb{E}[X^2].$$

Moreover, note that

$$\frac{\mathrm{d}}{\mathrm{d}\beta'} \mathbb{E}[X] = \frac{\mathrm{d}}{\mathrm{d}\beta'} \left(\frac{a_L'(\beta')}{a_L(\beta')}\right) = \frac{a_L''(\beta') a_L(\beta') - \big(a_L'(\beta')\big)^2}{\big(a_L(\beta')\big)^2} = \mathbb{E}[X^2] - \mathbb{E}[X]^2.$$

Therefore,

$$g(\beta') = \frac{a_L(\beta')\big(a_L(\beta') - 1\big)}{a_L(\beta')\, \mathbb{E}[X]} = \frac{a_L(\beta') - 1}{\mathbb{E}[X]}.$$

Differentiating yields

$$g'(\beta') = \frac{a_L'(\beta')\, \mathbb{E}[X] - \big(a_L(\beta') - 1\big) \frac{\mathrm{d}}{\mathrm{d}\beta'} \mathbb{E}[X]}{\mathbb{E}[X]^2} = \frac{a_L(\beta')\, \mathbb{E}[X]^2 - \big(a_L(\beta') - 1\big) \mathrm{Var}(X)}{\mathbb{E}[X]^2}.$$

Hence, to prove that $g(\beta')$ is increasing, it suffices to show that $g'(\beta') > 0$, i.e.,

$$\mathrm{Var}(X) < \frac{a_L(\beta')}{a_L(\beta') - 1}\, \mathbb{E}[X]^2.$$

By Lemma C.11 with $k = L$, the distribution of $X_L$ is the same as the distribution of $X$. Indeed, since $e^{\beta' x_i} = \left(\frac{L}{i}\right)^{\beta'} = L^{\beta'} i^{-\beta'}$, the normalizing factor $L^{\beta'}$ cancels, and hence $p_i = \frac{e^{\beta' x_i}}{\sum_{j=1}^{L} e^{\beta' x_j}} = \frac{i^{-\beta'}}{\sum_{j=1}^{L} j^{-\beta'}}$. Therefore, Lemma C.11 gives $\mathbb{E}[X^2] < 2\mathbb{E}[X]^2$. It follows that

$$\mathrm{Var}(X) = \mathbb{E}[X^2] - \mathbb{E}[X]^2 < \mathbb{E}[X]^2 < \frac{a_L(\beta')}{a_L(\beta') - 1}\, \mathbb{E}[X]^2.$$

It remains to verify the two limits. By a first-order Taylor expansion,

$$a_L(\beta') = \frac{1}{L}\sum_{i=1}^{L} e^{\beta' x_i} = 1 + \beta' \cdot \frac{1}{L}\sum_{i=1}^{L} x_i + o(\beta'), \qquad a'_L(\beta') = \frac{1}{L}\sum_{i=1}^{L} x_i e^{\beta' x_i} = \frac{1}{L}\sum_{i=1}^{L} x_i + o(1),$$

as $\beta' \to 0$. Since $\frac{1}{L}\sum_{i=1}^{L} x_i > 0$, it follows that

$$\lim_{\beta' \to 0} g(\beta') = \lim_{\beta' \to 0} \frac{a_L(\beta')\big(a_L(\beta') - 1\big)}{a'_L(\beta')} = \lim_{\beta' \to 0} \frac{\big(1 + o(1)\big)\big(\beta' \cdot \frac{1}{L}\sum_{i=1}^{L} x_i + o(\beta')\big)}{\frac{1}{L}\sum_{i=1}^{L} x_i + o(1)} = 0.$$

As $\beta' \to \infty$, the sum defining $a_L(\beta')$ is dominated by its largest term, i.e.,

$$a_L(\beta') = \frac{1}{L}\sum_{i=1}^{L} e^{\beta' x_i} \sim \frac{1}{L}e^{\beta' x_1}, \qquad a'_L(\beta') = \frac{1}{L}\sum_{i=1}^{L} x_i e^{\beta' x_i} \sim \frac{1}{L}x_1 e^{\beta' x_1},$$

and hence

$$\mathbb{E}[X] = \frac{a'_L(\beta')}{a_L(\beta')} \to x_1 = \log L, \qquad a_L(\beta') \to \infty.$$

Therefore,

$$\lim_{\beta' \to \infty} g(\beta') = \lim_{\beta' \to \infty} \frac{a_L(\beta') - 1}{\mathbb{E}[X]} = +\infty.$$

This completes the proof. $\qquad\square$

**Lemma C.10.** *Let $L \geq 2$ be an integer and define $a_L(\beta') := \frac{1}{L}\sum_{i=1}^{L} \big(\frac{L}{i}\big)^{\beta'}$. Then the mapping*

$$\beta' \longmapsto \frac{a_L(\beta')\big(a_L(\beta') - 1\big)}{a'_L(\beta')}, \qquad \beta' > 0,$$

*is strictly convex on $(0, \infty)$.*

*Proof.* As in Lemma C.9, define

$$g(\beta') := \frac{a_L(\beta')\big(a_L(\beta') - 1\big)}{a'_L(\beta')}.$$

For $i \in [L]$, define $x_i := \log\big(\frac{L}{i}\big)$. Let $X$ be the random variable supported on $\{x_1, \ldots, x_L\}$ with probability mass function

$$p_i := \frac{e^{\beta' x_i}}{\sum_{j=1}^{L} e^{\beta' x_j}} = \frac{e^{\beta' x_i}}{L\, a_L(\beta')}, \qquad i = 1, \ldots, L.$$

Then $a'_L(\beta') = a_L(\beta')\mathbb{E}[X]$ and $g(\beta') = \frac{a_L(\beta') - 1}{\mathbb{E}[X]}$. We also have

$$\frac{\mathrm{d}}{\mathrm{d}\beta'}\mathbb{E}[X] = \mathbb{E}[X^2] - \mathbb{E}[X]^2,$$

and

$$\frac{\mathrm{d}}{\mathrm{d}\beta'}\mathbb{E}[X^2] = \mathbb{E}[X^3] - \mathbb{E}[X]\mathbb{E}[X^2].$$

Differentiating $g(\beta')$ gives

$$g'(\beta') = \frac{a'_L(\beta')\mathbb{E}[X] - \big(a_L(\beta') - 1\big)\frac{\mathrm{d}}{\mathrm{d}\beta'}\mathbb{E}[X]}{\mathbb{E}[X]^2} = 1 + \big(a_L(\beta') - 1\big)\left(2 - \frac{\mathbb{E}[X^2]}{\mathbb{E}[X]^2}\right).$$

Hence

$$g''(\beta') = a_L'(\beta') \left( 2 - \frac{\mathbb{E}[X^2]}{\mathbb{E}[X]^2} \right) - \left( a_L(\beta') - 1 \right) \frac{\mathrm{d}}{\mathrm{d}\beta'} \left( \frac{\mathbb{E}[X^2]}{\mathbb{E}[X]^2} \right)$$

$$= a_L(\beta')\mathbb{E}[X] \left( 2 - \frac{\mathbb{E}[X^2]}{\mathbb{E}[X]^2} \right) - \left( a_L(\beta') - 1 \right) \frac{\mathrm{d}}{\mathrm{d}\beta'} \left( \frac{\mathbb{E}[X^2]}{\mathbb{E}[X]^2} \right).$$

Moreover,

$$\frac{\mathrm{d}}{\mathrm{d}\beta'} \left( \frac{\mathbb{E}[X^2]}{\mathbb{E}[X]^2} \right) = \frac{\mathbb{E}[X]\mathbb{E}[X^3] + \mathbb{E}[X]^2\mathbb{E}[X^2] - 2\mathbb{E}[X^2]^2}{\mathbb{E}[X]^3}.$$

Now apply Lemma C.11 with $k = L$. The distribution of $X_L$ is the same as the distribution of $X$, because $e^{\beta' x_i} = \left( \frac{L}{i} \right)^{\beta'} = L^{\beta'} i^{-\beta'}$. Thus $\mathbb{E}[X^2] < 2\mathbb{E}[X]^2$ and $2\mathbb{E}[X]\mathbb{E}[X^3] < 3\mathbb{E}[X^2]^2$. The second inequality implies $\mathbb{E}[X]\mathbb{E}[X^3] < \frac{3}{2}\mathbb{E}[X^2]^2$. Therefore,

$$\frac{\mathrm{d}}{\mathrm{d}\beta'} \left( \frac{\mathbb{E}[X^2]}{\mathbb{E}[X]^2} \right) < \frac{\frac{3}{2}\mathbb{E}[X^2]^2 + \mathbb{E}[X]^2\mathbb{E}[X^2] - 2\mathbb{E}[X^2]^2}{\mathbb{E}[X]^3} = \frac{\mathbb{E}[X^2]}{\mathbb{E}[X]} \left( 1 - \frac{\mathbb{E}[X^2]}{2\mathbb{E}[X]^2} \right).$$

Consequently,

$$g''(\beta') > a_L(\beta')\mathbb{E}[X] \left( 2 - \frac{\mathbb{E}[X^2]}{\mathbb{E}[X]^2} \right) - \left( a_L(\beta') - 1 \right) \frac{\mathbb{E}[X]}{2} \cdot \frac{\mathbb{E}[X^2]}{\mathbb{E}[X]^2} \left( 2 - \frac{\mathbb{E}[X^2]}{\mathbb{E}[X]^2} \right)$$

$$= \mathbb{E}[X] \left( 2 - \frac{\mathbb{E}[X^2]}{\mathbb{E}[X]^2} \right) \left[ a_L(\beta') - \frac{a_L(\beta') - 1}{2} \cdot \frac{\mathbb{E}[X^2]}{\mathbb{E}[X]^2} \right].$$

Since $\mathbb{E}[X^2] < 2\mathbb{E}[X]^2$, we have $2 - \frac{\mathbb{E}[X^2]}{\mathbb{E}[X]^2} > 0$. Also,

$$a_L(\beta') - \frac{a_L(\beta') - 1}{2} \cdot \frac{\mathbb{E}[X^2]}{\mathbb{E}[X]^2} > a_L(\beta') - \left( a_L(\beta') - 1 \right) = 1.$$

Finally, $\mathbb{E}[X] > 0$ for $L \geq 2$ and $\beta' > 0$. Therefore, $g''(\beta') > 0$. This completes the proof. $\qquad\square$

**Lemma C.11.** *Fix $\beta' > 0$. Let $w_i := i^{-\beta'}$ and $H_k := \sum_{i=1}^{k} w_i$ for $k \geq 1$. For each $k \geq 2$, let $X_k$ be the discrete random variable supported on*

$$\left\{ \log\left( \frac{k}{1} \right), \log\left( \frac{k}{2} \right), \dots, \log\left( \frac{k}{k} \right) \right\}$$

*with probability mass function*

$$\Pr\left( X_k = \log\left( \frac{k}{i} \right) \right) = \frac{w_i}{H_k}, \qquad i = 1, \dots, k.$$

*Then, for every $k \geq 2$, $\mathbb{E}[X_k^2] < 2\mathbb{E}[X_k]^2$, and $2\mathbb{E}[X_k]\mathbb{E}[X_k^3] < 3\mathbb{E}[X_k^2]^2$.*

*Proof.* We prove the two inequalities simultaneously by induction on $k$. First consider $k = 2$. Since

$$\Pr(X_2 = \log 2) = \frac{1}{1 + 2^{-\beta'}}, \qquad \Pr(X_2 = 0) = \frac{2^{-\beta'}}{1 + 2^{-\beta'}},$$

we have

$$\mathbb{E}[X_2] = \frac{\log 2}{1 + 2^{-\beta'}}, \qquad \mathbb{E}[X_2^2] = \frac{(\log 2)^2}{1 + 2^{-\beta'}}, \qquad \mathbb{E}[X_2^3] = \frac{(\log 2)^3}{1 + 2^{-\beta'}}.$$

Because $\frac{1}{1 + 2^{-\beta'}} > \frac{1}{2}$, we obtain $\mathbb{E}[X_2^2] < 2\mathbb{E}[X_2]^2$. Moreover,

$$2\mathbb{E}[X_2]\mathbb{E}[X_2^3] = 2 \left( \frac{1}{1 + 2^{-\beta'}} \right)^2 (\log 2)^4 < 3 \left( \frac{1}{1 + 2^{-\beta'}} \right)^2 (\log 2)^4 = 3\mathbb{E}[X_2^2]^2.$$

Thus the claim holds for $k = 2$.

Now assume that, for some $k \geq 2$, $\mathbb{E}[X_k^2] < 2\mathbb{E}[X_k]^2$ and $2\mathbb{E}[X_k]\mathbb{E}[X_k^3] < 3\mathbb{E}[X_k^2]^2$. We prove the two corresponding inequalities for $X_{k+1}$.

For $i = 1, \ldots, k$, $\log\left(\frac{k+1}{i}\right) = \log\left(\frac{k}{i}\right) + \log\left(\frac{k+1}{k}\right)$, while the point $i = k+1$ contributes value 0. Hence

$$\mathbb{E}[X_{k+1}] = \frac{H_k}{H_{k+1}}\left(\mathbb{E}[X_k] + \log\left(\frac{k+1}{k}\right)\right),$$

$$\mathbb{E}[X_{k+1}^2] = \frac{H_k}{H_{k+1}}\left(\mathbb{E}[X_k^2] + 2\log\left(\frac{k+1}{k}\right)\mathbb{E}[X_k] + \log^2\left(\frac{k+1}{k}\right)\right),$$

and

$$\mathbb{E}[X_{k+1}^3] = \frac{H_k}{H_{k+1}}\left(\mathbb{E}[X_k^3] + 3\log\left(\frac{k+1}{k}\right)\mathbb{E}[X_k^2] + 3\log^2\left(\frac{k+1}{k}\right)\mathbb{E}[X_k] + \log^3\left(\frac{k+1}{k}\right)\right).$$

We first prove $\mathbb{E}[X_{k+1}^2] < 2\mathbb{E}[X_{k+1}]^2$. By the induction hypothesis, it is enough to prove

$$2\mathbb{E}[X_k]^2 + 2\log\left(\frac{k+1}{k}\right)\mathbb{E}[X_k] + \log^2\left(\frac{k+1}{k}\right) < \frac{2H_k}{H_{k+1}}\left(\mathbb{E}[X_k] + \log\left(\frac{k+1}{k}\right)\right)^2.$$

Since $H_{k+1} = H_k + w_{k+1}$, this is equivalent to

$$\frac{w_{k+1}}{H_k} < \frac{\log\left(\frac{k+1}{k}\right)\left(2\mathbb{E}[X_k] + \log\left(\frac{k+1}{k}\right)\right)}{2\mathbb{E}[X_k]^2 + 2\log\left(\frac{k+1}{k}\right)\mathbb{E}[X_k] + \log^2\left(\frac{k+1}{k}\right)}.$$

We next prove this last inequality. By the telescoping identity $\log\left(\frac{k}{i}\right) = \sum_{j=i}^{k-1}\log\left(\frac{j+1}{j}\right)$, we get

$$H_k\mathbb{E}[X_k] = \sum_{i=1}^{k} w_i\log\left(\frac{k}{i}\right) = \sum_{j=1}^{k-1}\log\left(\frac{j+1}{j}\right)H_j.$$

For $1 \leq j \leq k-1$, $\log\left(\frac{j+1}{j}\right) \leq \frac{k}{j}\log\left(\frac{k+1}{k}\right)$. Therefore,

$$H_k\mathbb{E}[X_k] \leq k\log\left(\frac{k+1}{k}\right)\sum_{j=1}^{k-1}\frac{H_j}{j}.$$

We now show that the sequence $\left\{\frac{H_k}{kw_k}\right\}_{k \geq 1}$ is nondecreasing in $k$. To this end, note that

$$\frac{H_k}{kw_k} = \frac{1}{k}\sum_{i=1}^{k}\left(\frac{i}{k}\right)^{-\beta'}.$$

Let $f(x) := x^{-\beta'}$, which is convex on $(0, \infty)$ for $\beta' > 0$. For each $i \in \{1, \ldots, k\}$, observe that

$$\frac{i}{k} = \left(1 - \frac{i}{k}\right)\frac{i}{k+1} + \frac{i}{k}\cdot\frac{i+1}{k+1}.$$

By convexity of $f$, we obtain

$$\left(\frac{i}{k}\right)^{-\beta'} \leq \left(1 - \frac{i}{k}\right)\left(\frac{i}{k+1}\right)^{-\beta'} + \frac{i}{k}\left(\frac{i+1}{k+1}\right)^{-\beta'}.$$

Summing over $i = 1, \ldots, k$ yields

$$\sum_{i=1}^{k}\left(\frac{i}{k}\right)^{-\beta'} \leq \frac{1}{k}\sum_{i=1}^{k}(k-i)\left(\frac{i}{k+1}\right)^{-\beta'} + \frac{1}{k}\sum_{i=1}^{k}i\left(\frac{i+1}{k+1}\right)^{-\beta'}.$$

Re-indexing the second sum, we get

$$\sum_{i=1}^{k}\left(\frac{i}{k}\right)^{-\beta'} \leq \frac{1}{k}\left[\sum_{j=1}^{k}(k-j)\left(\frac{j}{k+1}\right)^{-\beta'} + \sum_{j=2}^{k+1}(j-1)\left(\frac{j}{k+1}\right)^{-\beta'}\right].$$

Since the coefficient of $\left(\frac{j}{k+1}\right)^{-\beta'}$ is $(k-j)+(j-1)=k-1$ for $j=2,\ldots,k$, while it is $k-1$ for $j=1$ and $k$ for $j=k+1$, the right-hand side equals

$$\frac{k-1}{k}\sum_{j=1}^{k}\left(\frac{j}{k+1}\right)^{-\beta'} + 1.$$

Dividing both sides by $k$ and using

$$\frac{H_{k+1}}{(k+1)w_{k+1}} = \frac{1}{k+1}\sum_{j=1}^{k+1}\left(\frac{j}{k+1}\right)^{-\beta'}, \qquad \sum_{j=1}^{k}\left(\frac{j}{k+1}\right)^{-\beta'} = (k+1)\frac{H_{k+1}}{(k+1)w_{k+1}} - 1,$$

we obtain

$$\frac{H_k}{kw_k} \leq \left(1 - \frac{1}{k^2}\right)\frac{H_{k+1}}{(k+1)w_{k+1}} + \frac{1}{k^2}.$$

Since $\frac{H_{k+1}}{(k+1)w_{k+1}} \geq 1$, it follows that

$$\frac{H_k}{kw_k} \leq \frac{H_{k+1}}{(k+1)w_{k+1}},$$

i.e., $\left\{\frac{H_k}{kw_k}\right\}_{k\geq 1}$ is nondecreasing. Hence,

$$\sum_{j=1}^{k-1}\frac{H_j}{j} = \sum_{j=1}^{k-1}\frac{H_j}{jw_j}w_j \leq \frac{H_k}{kw_k}\sum_{j=1}^{k-1}w_j = \frac{H_k H_{k-1}}{kw_k}.$$

It follows that $H_k\mathbb{E}[X_k] \leq \log\left(\frac{k+1}{k}\right)\frac{H_k H_{k-1}}{w_k}$, and hence

$$\mathbb{E}[X_k] \leq \log\left(\frac{k+1}{k}\right)\frac{H_{k-1}}{w_k}.$$

Moreover, the expression

$$\frac{\log\left(\frac{k+1}{k}\right)\left(2\mathbb{E}[X_k] + \log\left(\frac{k+1}{k}\right)\right)}{2\mathbb{E}[X_k]^2 + 2\log\left(\frac{k+1}{k}\right)\mathbb{E}[X_k] + \log^2\left(\frac{k+1}{k}\right)}$$

is decreasing as a function of $\mathbb{E}[X_k]$, because its derivative with respect to $\mathbb{E}[X_k]$ equals

$$-\frac{4\log\left(\frac{k+1}{k}\right)\mathbb{E}[X_k]\left(\mathbb{E}[X_k] + \log\left(\frac{k+1}{k}\right)\right)}{\left(2\mathbb{E}[X_k]^2 + 2\log\left(\frac{k+1}{k}\right)\mathbb{E}[X_k] + \log^2\left(\frac{k+1}{k}\right)\right)^2} < 0.$$

Therefore,

$$\frac{\log\left(\frac{k+1}{k}\right)\left(2\mathbb{E}[X_k] + \log\left(\frac{k+1}{k}\right)\right)}{2\mathbb{E}[X_k]^2 + 2\log\left(\frac{k+1}{k}\right)\mathbb{E}[X_k] + \log^2\left(\frac{k+1}{k}\right)} \geq \frac{2\frac{H_{k-1}}{w_k} + 1}{2\left(\frac{H_{k-1}}{w_k}\right)^2 + 2\frac{H_{k-1}}{w_k} + 1}.$$

On the other hand,

$$\frac{w_{k+1}}{H_k} = \frac{\frac{w_{k+1}}{w_k}}{\frac{H_{k-1}}{w_k} + 1} < \frac{1}{\frac{H_{k-1}}{w_k} + 1}.$$

Finally,

$$\frac{2\frac{H_{k-1}}{w_k} + 1}{2\left(\frac{H_{k-1}}{w_k}\right)^2 + 2\frac{H_{k-1}}{w_k} + 1} - \frac{1}{\frac{H_{k-1}}{w_k} + 1} = \frac{\frac{H_{k-1}}{w_k}}{\left(\frac{H_{k-1}}{w_k} + 1\right)\left(2\left(\frac{H_{k-1}}{w_k}\right)^2 + 2\frac{H_{k-1}}{w_k} + 1\right)} > 0.$$

Hence

$$\frac{w_{k+1}}{H_k} < \frac{\log\left(\frac{k+1}{k}\right)\left(2\mathbb{E}[X_k] + \log\left(\frac{k+1}{k}\right)\right)}{2\mathbb{E}[X_k]^2 + 2\log\left(\frac{k+1}{k}\right)\mathbb{E}[X_k] + \log^2\left(\frac{k+1}{k}\right)}.$$

This proves $\mathbb{E}[X_{k+1}^2] < 2\mathbb{E}[X_{k+1}]^2$.

It remains to prove $2\mathbb{E}[X_{k+1}]\mathbb{E}[X_{k+1}^3] < 3\mathbb{E}[X_{k+1}^2]^2$. Using the displayed formulas for the first three moments of $X_{k+1}$, it is enough to prove

$$2\left(\mathbb{E}[X_k] + \log\left(\frac{k+1}{k}\right)\right)\left(\mathbb{E}[X_k^3] + 3\log\left(\frac{k+1}{k}\right)\mathbb{E}[X_k^2] + 3\log^2\left(\frac{k+1}{k}\right)\mathbb{E}[X_k] + \log^3\left(\frac{k+1}{k}\right)\right)$$

$$< 3\left(\mathbb{E}[X_k^2] + 2\log\left(\frac{k+1}{k}\right)\mathbb{E}[X_k] + \log^2\left(\frac{k+1}{k}\right)\right)^2.$$

The difference between the right-hand side and the left-hand side equals

$$3\mathbb{E}[X_k^2]^2 - 2\mathbb{E}[X_k]\mathbb{E}[X_k^3] + 2\log\left(\frac{k+1}{k}\right)\left(3\mathbb{E}[X_k]\mathbb{E}[X_k^2] - \mathbb{E}[X_k^3]\right)$$

$$+ 6\log^2\left(\frac{k+1}{k}\right)\mathbb{E}[X_k]^2 + 4\log^3\left(\frac{k+1}{k}\right)\mathbb{E}[X_k] + \log^4\left(\frac{k+1}{k}\right).$$

By the induction hypothesis, $3\mathbb{E}[X_k^2]^2 - 2\mathbb{E}[X_k]\mathbb{E}[X_k^3] > 0$. Furthermore, since $2\mathbb{E}[X_k]\mathbb{E}[X_k^3] < 3\mathbb{E}[X_k^2]^2$ and $\mathbb{E}[X_k^2] < 2\mathbb{E}[X_k]^2$, we have

$$\mathbb{E}[X_k^3] < \frac{3\mathbb{E}[X_k^2]^2}{2\mathbb{E}[X_k]} < 3\mathbb{E}[X_k]\mathbb{E}[X_k^2].$$

Thus $3\mathbb{E}[X_k]\mathbb{E}[X_k^2] - \mathbb{E}[X_k^3] > 0$. All the remaining terms are also strictly positive. Hence the displayed difference is strictly positive, and therefore $2\mathbb{E}[X_{k+1}]\mathbb{E}[X_{k+1}^3] < 3\mathbb{E}[X_{k+1}^2]^2$. This completes the induction. $\qquad\square$

# Appendix: Experimental Details and Additional Discussions

## D. Implementation Details and Training Setup

### D.1 Experiments on Synthetic Tasks

**Graph Generation and Sample Pool.** We construct a balanced sample pool across configurations specified by the number of nodes $N$, target expected out-degree $\bar{d}$, and target distance $l$ via rejection sampling. Concretely, we first generate a directed unweighted graph $\mathcal{G}$ on $N$ labeled vertices by independently including each possible directed edge $(v_i, v_j)$ with probability $\bar{d}/(N-1)$, so that the expected out-degree is $\bar{d}$. Given $\mathcal{G}$, we then sample two distinct query vertices $v_s \neq v_t$ and compute the shortest path length from $v_s$ to $v_t$ using a standard Breadth-First Search routine on directed edges. If the resulting distance equals the target $l$ (with the convention $l = -1$ when $v_t$ is unreachable from $v_s$), we add the instance $(\mathcal{G}, v_s, v_t, l)$ to the pool associated with $(N, \bar{d}, l)$. We repeat this procedure until we either collect 2000 instances for each $(N, \bar{d}, l)$ combination or reach a preset sampling limit; if fewer than 2000 instances are found for a combination, we keep all collected instances.

In our experiments, we take $N \in \{6, 8, 10, 12, 14, 16, 18\}$, $\bar{d} \in \{2, \ldots, \lfloor N/2 \rfloor\}$, and $l \in \{-1, 1, 2, 3\}$. Each retained instance is rendered into a natural language prompt using a unified template. The prompt template is:

```
You are given a directed unweighted graph with nodes labeled 1..N.
N = ⟨N⟩
Edges are listed as ordered pairs (u,v), where each (u,v) represents a directed
edge from u to v:
⟨edge list⟩
Start s = ⟨vₛ⟩, Target t = ⟨vₜ⟩
```

```
Question:  Output the length (number of edges) of the shortest path from s to t.
If no path exists, output -1.
Answer with a single integer only.
```

The union of all instances across $(N, \bar{d}, l)$ forms our overall sample pool, from which we subsequently construct dataset splits for different experimental settings.

**Warm-up and Self-Improvement Finetuning Details.**   Our synthetic shortest path task allows control of the initialization reward $V_{p_0}(\hat{\theta}_0)$ and $V_{p_i}(\hat{\theta}_0)$ for $i \in [L]$. Note that, in our binary reward setting, $V_p(\theta)$ corresponds to the (population) Pass@1 accuracy of the model $\theta$ evaluated on questions drawn from $p$. Across experiments, we obtain different initial Pass@1 accuracies (corresponding to different values of $V_p(\theta)$) by varying the initialization model via warm-up finetuning and by varying the task difficulty through selecting different subsets of the overall sample pool.

Concretely, our warm-up datasets and the datasets used for self-improvement are sampled as disjoint subsets from the overall sample pool. Each warm-up dataset has a balanced composition across different $(N, \bar{d}, l)$ combinations, i.e., it contains equal numbers of samples for each $(N, \bar{d}, l)$. We warm up the pretrained base LLM using different warm-up datasets under different training configurations (learning rate and random seed) to obtain different initialization models.

| Hyperparameter | Value |
|---|---|
| Learning rate | $2 \times 10^{-4}$ |
| Batch size | 8 |
| LoRA rank | 16 |
| LoRA scaling | 32 |
| LoRA dropout | $5 \times 10^{-2}$ |

*Table 4.* Self-improvement finetuning hyperparameters used at each iteration on synthetic tasks.

| Figure | Initial test Pass@1 | $n$ | $m$ | $\beta'$ | $\Delta$ |
|---|---|---|---|---|---|
| Figure 3(a) | – | 5,000 | 1 | – | – |
| Figure 3(b) | 0.32 | – | 1 | – | – |
| Figure 3(c) | 0.32 | 4,000 | – | – | – |
| Figure 4(a) | – | 3,000 | 1 | – | 0.04 |
| Figure 4(b) | – | – | 1 | 0.25 | 0.04 |

*Table 5.* Experimental settings for Figures 3 and 4.

Next, for self-improvement, given an initialization model $\hat{\theta}_0$, we select an appropriate training set of size $nL$ together with a held-out test set such that the empirical Pass@1 accuracy of $\hat{\theta}_0$ matches the target value $V_{p_0}(\hat{\theta}_0)$, enabling control of the initialization performance. We fix the test set across all $L$ iterations, with a test size of 1000. To mitigate the effect of class imbalance, we enforce that, for both the per-iteration training set and the test set, the number of questions in each distance class $l \in \{-1, 1, 2, 3\}$ is balanced; moreover, within each class, we also match the number of questions that are initially answered correctly by $\hat{\theta}_0$. For experiments comparing easy-to-hard against the baseline, we require a different form of control. Specifically, the training set of size $nL$ should admit two different partitions: (i) a baseline partition whose $L$ subsets (each containing $n$ questions for one iteration) have the same (up to a small tolerance) initial Pass@1, matching $V_{p_0}(\hat{\theta}_0)$; and (ii) an easy-to-hard partition whose initial Pass@1 values across iterations satisfy Assumption 5.1. we impose these constraints jointly and solve the resulting data selection problem using a CP-SAT solver.

Given an initialization model $\hat{\theta}_0$ and a constructed dataset, we perform iterative self-improvement exactly following the setup in Sections 3 and 5.1. At each iteration, we use the same finetuning hyperparameters summarized in Table 4, and train for one epoch by default, with a minimum of 50 optimization steps for cases with too few accepted samples. Additionally, each data point in Figures 3 and 4 is obtained by averaging over five runs with different random seeds. Table 5 summarizes the experimental settings that are held fixed within each panel of Figures 3 and 4.

### D.2 Experiments on Standard Mathematical Reasoning Benchmarks

**Dataset split details.**   GSM8K (Cobbe et al., 2021) is released with 7473 training examples and 1319 test examples. Each GSM8K question is rendered into a natural language prompt using a unified template. The prompt template is:

```
Solve the following grade-school math word problem.
Give only the final answer as an integer.  Do not show steps.

Problem:  ⟨question⟩

Answer:
```

Throughout our GSM8K experiments, we fix the official test split for evaluation. For training, we fix the number of self-improvement iterations to $L = 3$, and partition the training data into three rounds such that the model's initial performance on each round matches its initial performance on the full training set. Except in the experiment that explicitly varies the question budget $n$, we set $n = 2400$.

DeepMind Mathematics (Saxton et al., 2019) contains multiple categories of mathematical reasoning problems, each further divided into fine-grained modules, with a native easy/medium/hard difficulty partition. To facilitate evaluation, we filter out all modules whose answers are not single numeric values. For the remaining modules, each question is rendered into a natural language prompt using the following unified template:

```
Solve the following school-level math problem.
Give only the final answer as a single integer, decimal, or fraction.  Do not show
steps.

Problem:  ⟨question⟩

Answer:
```

We construct a relatively simple task that yields the high initial model Pass@1 performance reported in Table 8 by selecting data from the easy splits of the nine modules on which QWEN3-8B achieves the highest accuracy. These modules are algebra_linear_1d, arithmetic_add_or_sub, arithmetic_div, arithmetic_mul, arithmetic_nearest_integer_root, measurement_conversion, numbers_gcd, numbers_place_value, and numbers_round_number. Our training and test sets contain samples drawn uniformly from these modules. We then partition the training data into three iterations such that the model's initial performance on each round matches (up to a small tolerance) its initial performance on the selected training set. For Table 8, we set $n = 400$. For the iterative self-improvement with easy-to-hard curriculum experiments, we consider the three largest categories, namely algebra, arithmetic, and numbers, and run experiments on each module in these categories. The data construction procedure is described in Section 6.2.2. For Table 2, we set $n = 1000$.

**Self-Improvement Finetuning Details.**    Similar to Appendix D.1, for both GSM8K and DeepMind Mathematics, we perform iterative self-improvement exactly following the setup in Sections 3 and 5.1. Across all experiments on standard mathematical reasoning benchmarks, we use the same self-improvement finetuning hyperparameters summarized in Table 6.

| Hyperparameter | Value |
| --- | --- |
| Learning rate | $2 \times 10^{-4}$ |
| Weight decay | $1 \times 10^{-4}$ |
| Batch size | 8 |
| LoRA rank | 16 |
| LoRA scaling | 32 |
| LoRA dropout | $5 \times 10^{-2}$ |

*Table 6.* Self-improvement finetuning hyperparameters used at each iteration on standard mathematical reasoning benchmarks.

**Additional Experimental Results.**    We include additional experimental results (Table 7, Table 8, Table 9, and Figure 5) to provide further empirical support for our main findings.

| Model | Iteration $t$ | | | | Absolute Improvement |
|---|---|---|---|---|---|
| | 0 | 1 | 2 | 3 | |
| Llama-3.2-1B-Instruct | 2.96 | 3.56 | 3.14 | 2.68 | -0.28 |
| Llama-3.2-3B-Instruct | 6.07 | 6.95 | 5.97 | 5.61 | -0.46 |
| Qwen2.5-3B-Instruct | 10.39 | 11.78 | 13.04 | 13.62 | 3.23 |
| Qwen2.5-7B-Instruct | 18.12 | 19.16 | 20.17 | 20.19 | 2.07 |
| Qwen3-8B | 27.52 | 28.66 | 29.75 | 30.67 | 3.15 |
| Qwen3-32B | 38.59 | 41.47 | 43.09 | 43.77 | 5.18 |

*Table 7.* Self-improvement trajectories of different base models on GSM8K. Iteration $t$ reports the model's Pass@1 accuracy before iteration $t$ (equivalently, after iteration $t-1$). The last column reports the absolute improvement between the final Pass@1 and the initial value. All values are percentages.

| Model | Iteration $t$ | | | | Absolute Improvement |
|---|---|---|---|---|---|
| | 0 | 1 | 2 | 3 | |
| Llama-3.2-1B-Instruct | 18.89 | 35.16 | 36.42 | 36.09 | 17.20 |
| Llama-3.2-3B-Instruct | 37.00 | 46.04 | 51.00 | 54.44 | 17.44 |
| Qwen2.5-3B-Instruct | 55.89 | 61.29 | 67.31 | 70.42 | 14.53 |
| Qwen2.5-7B-Instruct | 74.00 | 76.98 | 81.44 | 82.98 | 8.98 |
| Qwen3-8B | 85.11 | 86.65 | 90.89 | 92.78 | 7.67 |
| Qwen3-32B | 97.89 | 95.67 | 96.24 | 96.69 | -1.20 |

*Table 8.* Self-improvement trajectories of different base models on DeepMind Mathematics Dataset. Iteration $t$ reports the model's accuracy before iteration $t$ (equivalently, after iteration $t-1$). The last column reports the absolute improvement between the final value and the initial value. All values are percentages.

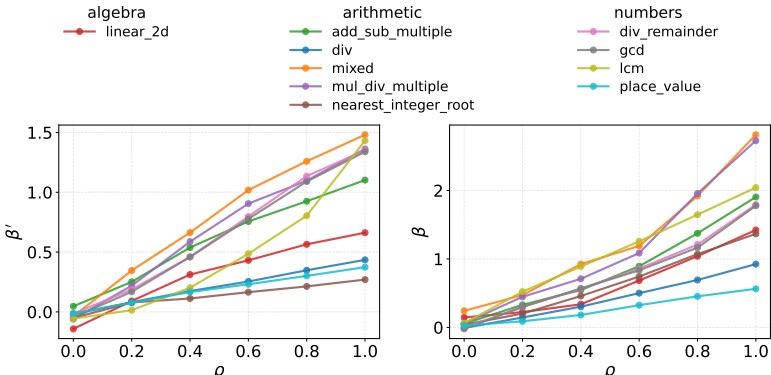

*Figure 5.* The left panel plots the estimate of $\beta'$ as a function of $\rho$, and the right panel plots the estimate of $\beta$ as a function of $\rho$, for all modules reported in Table 2. Here, the estimates of $\beta'$ and $\beta$ are computed from the initialization model using the form in Assumption 5.1.

# E. Additional Discussions

Our self-improvement framework has the potential to be extended to broader settings beyond the formulation considered in this paper. First, our reward formulation follows the clean verification setting also adopted in many prior theoretical analyses of self-improvement (e.g., Huang et al. (2025)) to isolate the core feedback mechanism. However, considering reward noise is also practically important, since in realistic LLM pipelines the reward signal can be imperfect: verifiers may make mistakes, learned reward models may be biased, and self-verification signals may depend on the current model's own capabilities. To address this issue, our framework can in principle be extended to noisy-reward settings once an explicit relationship between the observed noisy reward and the latent clean reward is specified. Nevertheless, more realistic formulations of reward noise generally depend on assumptions about the specific data distribution, verifier behavior, and model structure. We therefore view this as an important future direction.

| $n$ | Iteration $t$ | | | | Absolute Improvement |
|---|---|---|---|---|---|
| | 0 | 1 | 2 | 3 | |
| 600 | 27.52 | 21.59 | 22.30 | 23.46 | -4.06 |
| 1200 | 27.52 | 26.76 | 27.32 | 28.08 | 0.56 |
| 1800 | 27.52 | 28.40 | 29.13 | 30.25 | 2.73 |
| 2400 | 27.52 | 28.66 | 29.75 | 30.67 | 3.15 |

*(a)* Different question budgets $n$.

| $m$ | Iteration $t$ | | | | Absolute Improvement |
|---|---|---|---|---|---|
| | 0 | 1 | 2 | 3 | |
| 1 | 27.52 | 28.66 | 29.75 | 30.67 | 3.15 |
| 4 | 27.52 | 29.02 | 31.13 | 31.51 | 3.99 |
| 16 | 27.52 | 30.42 | 31.55 | 32.86 | 5.34 |

*(b)* Different per-question answer budgets $m$.

*Table 9.* Self-improvement trajectories of QWEN3-8B under different budgets on GSM8K. Iteration $t$ reports the model's Pass@1 accuracy before iteration $t$ (equivalently, after iteration $t-1$). The last column reports the absolute improvement between the final Pass@1 and the initial value. All values are percentages.

Second, our framework also has a mechanistic connection to test-time compute methods. Although test-time compute methods based purely on search or self-correction without parameter updates (Chen et al., 2025) are outside the scope of our current theory, many test-time training approaches (Sun et al., 2025; Zhang et al., 2026) improve model performance via gradient-based parameter updates, making them mechanistically close to our setting. Nevertheless, since our current finite-sample bounds rely on empirical maximum-likelihood training over reward-filtered samples, the specific bounds and feedback map do not directly apply to these methods. Accordingly, deriving analogous finite-sample penalty terms and feedback loops for the specific loss objectives and update rules of such test-time training methods is also an interesting direction for future work.

