# OpenReview forum: "A Task-centric Theory for Iterative Self-Improvement with Easy-to-Hard Curricula"
_ICML.cc/2026/Conference — ICML 2026 regular_

### Official Review · Reviewer_AEhR · 2026-02-21

**Soundness:** 3
**Presentation:** 3
**Significance:** 3
**Originality:** 3
**Overall Recommendation:** 5
**Confidence:** 2

**Summary:**

The paper introduces a theory of iterative self-improvement with easy-to-hard curricula. It provides empirical evidence using a shortest path graph task and performing iterative self-improvement.

**Compliance With Llm Reviewing Policy:**

Affirmed.

**Final Justification:**

See rebuttal acknowledgement.

**Key Questions For Authors:**

No questions.

**Limitations:**

- Only one empirical task is tested. It seems like it would be easy to to come up with other similar synthetic tasks.

**Strengths And Weaknesses:**

Unfortunately, I am rating my confidence low as this is a learning theory paper which is not my area of expertise. I cannot accurately judge the theory section of this paper, however, the empirical section of the paper is within my area. I generally understood the conclusions and what the paper was demonstrating.

Strengths:
- Some of the theory conclusions drawn around iterative self-improvement, saturation, task difficulty, and easy-to-hard curricula are interesting to study.
- The authors provide an empirical experiment to demonstrate support for some of the theory conclusions/remarks.

Weaknesses:
- A single empirical experiment task.

---

> ### Author Rebuttal · Authors · 2026-03-31
>
> We thank the reviewer for their time and feedback. We are glad that the reviewer found our theoretical findings interesting and recognized that these findings are supported by our empirical experiment.
>
> **[Weakness & Limitation: Additional experiments to further support the theory.]**
> The synthetic graph task in Section 6 enables a comprehensive test of our main theoretical results. That said, we agree that the applicability of our theory can be further demonstrated on a broader range of tasks. Below we summarize the main results of our additional experiments.
>
> Additional standard benchmarks:
> - GSM8K [1]: 8.5K grade-school math word problems.
> - DeepMind Math [2]: school-level math problems with a native easy/medium/hard difficulty split.
>
> Key results:
> - Table 1 shows that when a task is too easy or too hard relative to the model, self-improvement tends to deteriorate, whereas moderate task difficulty is more conducive to sustained iterative self-improvement.
>
> *Table 1.* Absolute Pass@1 improvement (%) $\Delta$ from the initial to the final (third) iteration of different base models on GSM8K and on the nine easiest modules of DeepMind Math.
>
> | Model | GSM8K Initial acc. | GSM8K $\Delta$ | DeepMind Math Initial acc. | DeepMind Math $\Delta$ |
> |---|---:|---:|---:|---:|
> | Llama-3.2-1B-Instruct | 3.0 | -0.3 | 18.9 | 17.2 |
> | Llama-3.2-3B-Instruct | 6.1 | -0.5 | 37.0 | 17.4 |
> | Qwen2.5-3B-Instruct | 10.4 | 3.2 | 55.9 | 14.5 |
> | Qwen2.5-7B-Instruct | 18.1 | 2.1 | 74.0 | 9.0 |
> | Qwen3-8B | 27.5 | 3.2 | 85.1 | 7.7 |
> | Qwen3-32B | 38.6 | 5.2 | 97.9 | -1.2 |
>
> - Table 2 shows that larger per-iteration question budgets $n$ and larger per-question answer budgets $m$ are both more favorable to self-improvement.
>
> *Table 2.* Absolute Pass@1 improvement (%) $\Delta$ from the initial to the final (third) iteration of Qwen3-8B under different budgets on GSM8K.
>
> | Budget | Value | $\Delta$ |
> |---|---:|---:|
> | Question budget $n$ | 600  | -4.1 |
> |  | 1200 | 0.6 |
> |  | 1800 | 2.7 |
> |  | 2400 | 3.2 |
> | Per-question answer budget $m$ | 1  | 3.2 |
> |  | 4  | 4.0 |
> |  | 16 | 5.3 |
>
> - For easy-to-hard iterative self-improvement, DeepMind Math enables us to validate our theory using its native difficulty split. We consider 3 iterations and control the adjacent task difficulty gap by varying the sampling ratios of easy/medium/hard examples. We use $\rho$ for this control: $\rho=0$ gives uniform sampling in every iteration (no difficulty separation); when $\rho=1$, the first/second/third iterations use only easy/medium/hard examples (maximal difficulty separation). For $\rho \in (0,1)$, the sampling ratios are obtained by linear interpolation.
>   - Our experiments show that, across all modules in Table 3, $\beta$ and $\beta'$ increase strictly with $\rho$ (see anonymized link https://anonymous.4open.science/r/Fig-8F84/fig.png), suggesting that these theoretical quantities serve as effective proxies for the dataset's native difficulty separation.
>   - Table 3 shows that, as the adjacent-task difficulty ratio increases, the final Pass@1 gap between easy-to-hard and the baseline typically first increases and then decreases, consistent with our theoretical prediction that moderate difficulty ratios between adjacent tasks are the most favorable for easy-to-hard scheduling.
>
> *Table 3.* Final Pass@1 gap (%) between easy-to-hard and the baseline on DeepMind Math using Qwen3-8B, under different $\rho$ values. We report all modules from algebra, arithmetic, and numbers, excluding those for which curriculum effects are not meaningfully measurable (i.e., $\beta' < 0.1$ when $\rho=1$ or gap variation < 1\%). Best values are in **bold**.
>
> | Category | Module | 0 | 0.2 | 0.4 | 0.6 | 0.8 | 1 |
> |---|---|---:|---:|---:|---:|---:|---:|
> | algebra | linear_2d | 0 | 0.04 | 0.84 | **2.52** | 1.96 | 2.04 |
> | arithmetic | add_sub_multiple | 0 | 2.76 | 7.68 | 11.60 | **17.08** | 10.64 |
> |  | div | 0 | 0.16 | 0.42 | **0.98** | 0.88 | -0.54 |
> |  | mixed | 0 | -1.06 | 1.64 | 3.40 | **4.62** | 3.20 |
> |  | mul_div_multiple | 0 | 2.60 | 4.76 | 9.56 | **12.32** | 10.48 |
> |  | nearest_integer_root | 0 | 2.38 | **2.50** | -2.72 | -4.44 | -6.64 |
> | numbers | div_remainder | 0 | 2.80 | 2.16 | 3.78 | **3.80** | 2.90 |
> |  | gcd | 0 | 0.22 | 0.32 | **0.78** | 0.04 | -0.44 |
> |  | lcm | 0 | 0.12 | 0.26 | 1.04 | 0.96 | **2.66** |
> |  | place_value | 0 | 0.02 | **0.28** | 0.02 | -0.44 | -3.12 |
>
> Overall, the above results provide clear additional evidence for the main conclusions of our theory in **Sections 4 and 5**, and are consistent with our synthetic results. Due to space constraints, we omit some additional results here. We will include all these experiments in the next version.
>
> We are happy to answer any further questions. If our responses above have addressed your concerns, we would truly appreciate a re-evaluation accordingly.
>
> [1] Training Verifiers to Solve Math Word Problems, 2021.
> [2] Analysing Mathematical Reasoning Abilities of Neural Models, 2019.

---

> > ### Author Rebuttal · Reviewer_AEhR · 2026-04-01
> >
> > I thank the authors for providing additional empirical evidence of their findings. I believe these results around iterative self-improvement are novel and interesting to the community and have been presented well. I have increased my score to "Accept" based on these strengths. I will leave my confidence here low as I still cannot judge the validity of the theory component so other reviewer's opinions should be taken into account for those findings. Best of luck to the authors on their submission.

---

### Official Review · Reviewer_zvPU · 2026-03-13

**Soundness:** 3
**Presentation:** 3
**Significance:** 3
**Originality:** 3
**Overall Recommendation:** 5
**Confidence:** 3

**Summary:**

## Summary
This paper aims to provide a task-centric theoretical foundation for iterative self-improvement and easy-to-hard (E2H) curriculum learning in Large Language Models (LLMs). By modeling the self-improvement process as maximum likelihood estimation (MLE) fine-tuning on a reward-filtered distribution, the authors derive theoretical lower bounds for the expected reward under a finite-sample setting. Furthermore, the paper quantifies the advantages of the E2H curriculum over training with a fixed distribution mixture, theoretically proving the existence of a phase transition in performance improvement given a specific sample budget. The empirical section validates the theoretical findings using the Llama-3.2-1B model on a synthetic Shortest Path prediction task.

**Compliance With Llm Reviewing Policy:**

Affirmed.

**Final Justification:**

I think the author has resolved my issue with their empirical experiment results. Thank authors for their new results and explanations. Hope author will add all these new experiment results and applications to test time training setting in revised version. Since my initial score is 5, I will keep this positive score.

**Key Questions For Authors:**

Questions:
1. Generalization to Test-Time Training: If the current "generate-and-filter" theoretical framework were extended to label-free self-correction or test-time search tasks, how would the derivation of the finite-sample penalty term $\nu$ and the acceptance rate feedback loop fundamentally change?
2. Task Separation Under Complex Distributions: The difficulty of the "Shortest Path" task in the experiments follows a very regular pattern. If applied to a dataset requiring non-linear, jumping thought processes like ARC-AGI, how do the authors suggest quantifying and ensuring that the adjacent task difficulty ratio $\beta'$ consistently remains within the theoretically expected "moderate separation" regime?
3. Quantitative Estimation of the RL Switch Point: Since the theory proves that RFT converges to a sub-optimal $x_+$, in a practical alignment pipeline, could we estimate a quantitative "switch threshold" based on the formulas in this paper? Specifically, at what sample size or performance gradient should the system stop MLE iteration and introduce reinforcement learning algorithms like PPO/GRPO to break the bottleneck?

**Limitations:**

yes

**Strengths And Weaknesses:**

## Strengths
- First-Principles Theoretical Perspective: The paper breaks away from the common "infinite data" or "continuous limit" assumptions prevalent in prior literature. By directly incorporating the finite-sample budget ($n, m$) and task difficulty into the derivation of generalization bounds, it provides a rigorous mathematical framework for the learning dynamics of LLMs that is highly relevant to engineering practice.
- Rigorous Characterization of Phase Transitions: The paper not only empirically supports the E2H generalization strategy but, more importantly, elegantly proves the existence of a "critical sample budget". This quantitative explanation of phase transitions is highly inspiring for understanding the underlying mechanisms of non-linear capability leaps as data scale increases.
- Foundational Explanatory Power for Engineering Practice: The theory clearly defines the inherent performance upper bound $x_+(1,\nu) < 1-\gamma$ of self-improvement, perfectly explaining why purely rejection sampling fine-tuning (RFT)-based mechanisms ultimately fall into an information cocoon. This provides solid mathematical backing for the widely adopted paradigm in modern RLHF pipelines (like those in openrlhf): starting with SFT/RFT for cold-start, then pivoting to advantage-based reinforcement learning.
## Weaknesses
- Limited and Singular Experimental Setting: Despite the thorough theoretical derivations, the empirical validation is limited to the synthetic "Shortest Path" task. In real-world complex reasoning scenarios (such as strong logical intelligence benchmarks like ARC-AGI, or complex code generation tasks), task difficulty is rarely linearly incremental, and distribution shifts are far more complex. Relying solely on a graph algorithm task is insufficient to demonstrate the universality of this theory in training state-of-the-art LLMs.
- Lack of Discussion on Test-Time Compute / Self-Correction: The paper's theory focuses on iterative improvement during the training phase. Given that the current frontier of the field is heavily pushing to break capability ceilings via test-time training and label-free self-correction, the theoretical framework's forward-looking value is somewhat diminished if it cannot account for the impact of test-time compute budgets (e.g., multi-path decoding attempts and verification mechanisms).
- Idealized Reward Signal Assumption: The theory heavily relies on a perfect and deterministic reward filter function. However, in actual easy-to-hard generalization, especially in label-free self-verification scenarios, the reward signal is often highly noisy. The theoretical framework lacks a discussion on tolerance to reward noise.

---

> ### Author Rebuttal · Authors · 2026-03-30
>
> We thank the reviewer for the thoughtful and positive evaluation. We are grateful that the reviewer recognizes the rigor of our mathematical framework, the empirical support for our theoretical findings, and the importance of our work to engineering practice.
>
> **[Weakness #1: Additional experiments on more reasoning tasks to further support the theory.]**
> We agree that the applicability of our theory can be further validated on a wider range of more complex reasoning tasks. To strengthen this point, we conducted additional experiments on standard math reasoning benchmarks (GSM8K [1] and DeepMind Math [2]) using a wider range of base models (Llama-3.2-{1B, 3B}-Instruct, Qwen2.5-{3B, 7B}-Instruct, and Qwen3-{8B, 32B}). Due to the character limit, **we refer the reviewer to our response to Reviewer AEhR under [Weakness & Limitation] for the full supplementary experimental results**. These results provide clear further support for the main conclusions of our theory in Sections 4-5 , and are consistent with our synthetic results.
>
> **[Weakness #2 & Question #1: Applicability of our theory to test-time compute settings.]**
> We would first like to clarify that training-time self-improvement is the main focus of our paper, as it already closely matches many practical reasoning pipelines (e.g., [3,4]). That said, our framework is also mechanistically related to some test-time compute methods. In particular, many test-time training approaches [5,6] also improve model performance via gradient-based parameter updates, making them mechanistically close to our setting. In such cases, the exact finite-sample penalty term and the feedback loop can be derived for the specific loss objective and update rule of the method. By contrast, test-time compute methods based purely on search or self-correction without parameter updates [7] are outside the scope of our current theory. We will clarify these distinctions in the next version.
>
> **[Weakness #3: Discussion of noisy reward signals.]**
> Our reward formulation follows the clean verification setting also adopted in many prior theoretical analyses of self-improvement (e.g., [8]) to isolate the core feedback mechanism. This framework can in principle be extended to noisy reward settings once an explicit relationship between the noisy reward and the latent clean reward is specified. That said, more realistic formulations of reward noise generally depend on assumptions about the specific data distribution, verifier behavior, and model structure. We therefore view this as an important future direction and will add discussion in the next version.
>
> **[Question #2: Practical guidance for easy-to-hard task scheduling.]**
> We would first like to clarify that, in our framework, difficulty is defined by the model’s expected reward on a given task, rather than by particular structural patterns of the dataset. Therefore, the framework is not restricted to tasks with regular or linear difficulty patterns. To maintain a moderate separation regime across tasks, for datasets with native difficulty splits (e.g., DeepMind Math), a practical approach is to control the sampling proportions across different difficulty levels; please see our response to Reviewer AEhR [Weakness & Limitation] for a concrete example on DeepMind Math. For datasets without an explicit difficulty partition, a practical approach is to first estimate task difficulty via expected reward, and then group tasks accordingly or control the sampling proportions across tasks to design the curriculum schedule.
>
> **[Question #3: Practical guidance for estimating the RL switch point.]**
> Our theory shows that the reward lower bound under iterative RFT converges to a suboptimal $x_+$. This suggests a practical heuristic: switch to RL once the gain from another RFT iteration becomes negligibly small. While the iteration map $F$ has an explicit form and can in principle be computed, a more practical rule is to set a small threshold $\epsilon$ and stop once the performance gain between two iterations falls below $\epsilon$. Both the curve induced by $F$ and our results in Section 6 suggest relatively fast saturation, so in practice, only a small number (3-4) of RFT iterations may already capture most of the achievable gain before switching to RL becomes preferable.
>
> We thank the reviewer again for these suggestions. We are happy to answer any further questions.
>
> [1] Training Verifiers to Solve Math Word Problems, 2021.
> [2] Analysing Mathematical Reasoning Abilities of Neural Models, 2019.
> [3] STaR: Bootstrapping Reasoning With Reasoning, 2022.
> [4] Goedel-Prover: A Frontier Model for Open-Source Automated Theorem Proving, 2025
> [5] Learning to (Learn at Test Time): RNNs with Expressive Hidden States, 2024.
> [6] Test-Time Training Done Right, 2025.
> [7] SETS: Leveraging Self-Verification and Self-Correction for Improved Test-Time Scaling, 2025.
> [8] Self-Improvement in Language Models: The Sharpening Mechanism, 2024.

---

> > ### Author Rebuttal · Reviewer_zvPU · 2026-04-04
> >
> > I think the author has resolved my issue with their empirical experiment results. Thank authors for their new results and explanations. Hope author will add all these new experiment results and applications to test time training setting in revised version. Since my initial score is 5, I will keep this positive score. Good luck.

---

### Official Review · Reviewer_7Qai · 2026-03-14

**Soundness:** 3
**Presentation:** 3
**Significance:** 3
**Originality:** 3
**Overall Recommendation:** 4
**Confidence:** 3

**Summary:**

This paper develops a finite-sample, task-centric theory for iterative self-improvement in LLM reasoning, modeling each round as maximum-likelihood training on reward-filtered self-generated data. Its main contribution is a set of lower bounds that reveal an explicit feedback mechanism: better models accept more data, which can sustain improvement but also leads to saturation. It then extends the analysis to multiple task difficulty levels and proves conditions under which an easy-to-hard curriculum achieves a better lower-bound guarantee than a fixed-mixture baseline, highlighting the roles of initialization, adjacent difficulty ratios, and sample budget. These theoretical findings are supported by Monte Carlo simulations and controlled shortest-path reasoning experiments with Llama-3.2-1B-Instruct.

**Compliance With Llm Reviewing Policy:**

Affirmed.

**Final Justification:**

The rebuttal addresses my main concerns and improves the paper’s clarity and empirical support. I would like to keep my score.

**Key Questions For Authors:**

1. Can the authors explain how one would estimate $\beta$, $\beta^\prime$, or the sign conditions on $M_i$ and $N$, in practice before deciding whether easy-to-hard should help?
2. How sensitive is the curriculum result to violations of the stable task-difficulty ordering in Assumption 5.1?

**Limitations:**

The limitations section could be improved by more explicitly discussing the gap to realistic LLM self-improvement pipelines.

**Strengths And Weaknesses:**

Strengths

1. This paper tackles a timely and important problem. Its finite-sample formulation represents a real step beyond population-level analyses, with the main theorems clearly showing how acceptance rate, question budget, and answer budget shape the guarantees.
2. The paper identifies fairly specific structural conditions for when easy-to-hard curriculum curricula should help relative to a fixed-mixture baseline, providing a potentially useful theoretical lens for understanding many recent reasoning pipelines that schedule problem difficulty across rounds.

Weaknesses

1. The iterative and curriculum guarantees  rely on a strong acceptance–performance coupling assumption (Assumption 4.4), so it is unclear whether the theory applies beyond stylized regimes with relatively uniform improvement across questions.
2. The validation is restricted to synthetic shortest-path reasoning with a small model, so the paper does not yet show that the predicted phase behavior or curriculum effects persist in more realistic LLM reasoning domains. This matters because the paper is positioned as guidance for iterative self-improvement of LLM reasoning broadly.

---

> ### Author Rebuttal · Authors · 2026-03-30
>
> We thank the reviewer for the thoughtful and constructive feedback. We appreciate the reviewer’s recognition of the timeliness and novelty of our theoretical framework for iterative self-improvement and easy-to-hard curriculum, as well as the intuitive nature of our results and their relevance to current practical reasoning pipelines. Below, we provide additional clarifications on the key points raised by the reviewer.
>
> **[Weakness #1: Justification of the acceptance–performance coupling assumption.]**
> We would like to clarify that Assumption 4.4 is not a uniform improvement assumption across questions. In our binary-reward setting, the expected reward satisfies $V_p(\theta) = \mathbb{E}_{q \sim p}[\alpha(\theta,q)]$. Thus, **Assumption 4.4 only imposes a lower-tail regularity condition** on the distribution of per-question success probabilities. It allows substantial heterogeneity across questions and only excludes a small, exceptional set of disproportionately hard questions. It is worth mentioning that our assumption is in the same spirit as classical noise/margin conditions in learning theory, e.g., the Massart/Tsbyakov noise condition.
>
> **[Weakness #2 & Limitation: Additional experiments on more reasoning tasks and stronger base models to further support the theory.]**
> The synthetic graph task in Section 6 enables a comprehensive test of our main theoretical results. That said, we agree that the applicability of our theory can be further demonstrated on a broader range of tasks and models. To strengthen this point, we conducted additional experiments on standard math reasoning benchmarks (GSM8K [1] and DeepMind Math [2]) using a wider range of base models (Llama-3.2-{1B, 3B}-Instruct, Qwen2.5-{3B, 7B}-Instruct, and Qwen3-{8B, 32B}). These results are consistent with our synthetic findings and further support the main conclusions of Sections 4-5. Due to the character limit, **we refer the reviewer to our response to Reviewer AEhR under [Weakness & Limitation] for the full results**. We will include these experiments in the next version and expand the discussion of realistic LLM self-improvement settings.
>
> **[Question #1: Practical guidance for estimating parameters/conditions for easy-to-hard curriculum.]**
> First, in practice we estimate $\beta$ and $\beta'$ by measuring accuracies of the pretrained initialization/warm-up models across different tasks, and then plugging them into the definition in Assumption 5.1. Empirically, these estimates are relatively stable, and the fact that the resulting experimental trends align well with the theoretical predictions also suggests that they are useful in practice.
> Second, the sign conditions on $M_i$ and $N$ are explicit in Definition A.2 and can in principle be checked quantitatively. Since these expressions are somewhat cumbersome to compute and interpret directly, we provide more interpretable guidance through Corollary 5.3 and Remark 5.4 for the conditions $\\{M_i<0\\}$, and through Proposition 5.5, Corollary 5.6, and Remarks 5.7.1-5.7.4 for the condition $N<0$. Accordingly, in practice we recommend using these qualitative conclusions as the main guidance.
>
> **[Question #2: Justification for focusing on the stable easy-to-hard regime.]**
> We would like to clarify that our curriculum theory is not based on an exactly fixed difficulty ordering with no variation. Assumption 5.1 already allows the relative difficulty between adjacent tasks to vary across models in the post-training neighborhood $\Theta$, controlled by the exponents $\beta,\beta’$. The width $\Delta=\beta-\beta’$ precisely captures the extent of uncertainty in the ordering. Moreover, our theory unveils the relation between such uncertainty and self-improvement: as $\Delta$ increases, the curriculum guarantee weakens continuously rather than disappearing abruptly (see Remark 5.7.2).
>
> At the same time, we agree that if task difficulties reorder substantially across models so that no bounded ordering of the form in Assumption 5.1 is valid, then the current theorem no longer applies. Studying such dynamically changing curricula is an interesting future direction, but outside the scope of this paper. Empirically, for common task families and models (like in our experiments), the relative task-difficulty ordering can be explicitly controlled to remain stable, which is why we focus on this regime here.
>
> We again thank the reviewer for these suggestions. We are happy to answer any further questions you may have. If our responses above have addressed your concerns, we would truly appreciate a re-evaluation accordingly.
>
> [1] Training Verifiers to Solve Math Word Problems, 2021.
> [2] Analysing Mathematical Reasoning Abilities of Neural Models, 2019.

---

### Decision · Program_Chairs · 2026-04-30

**Decision:**

Accept (regular)

**Comment:**

The paper makes a clear theoretical contribution to understanding LLM self-improvement and curriculum learning. While empirical breadth and some modeling assumptions could be improved, the work provides valuable insights and a foundation that future research can build upon.